# Human contributions to global soundscapes are less predictable than the acoustic rhythms of wildlife

Across the world, human (anthropophonic) sounds add to sounds of biological (biophonic) and geophysical (geophonic) origin, with human contributions including both speech and technophony (sounds of technological devices). To characterize society's contribution to the global soundscapes, we used passive acoustic recorders at 139 sites across 6 continents, sampling both urban green spaces and nearby pristine sites continuously for 3 years in a paired design. Recordings were characterized by bird species richness and by 14 complementary acoustic indices. By relating each index to seasonal, diurnal, climatic and anthropogenic factors, we show here that latitude, time of day and day of year each predict a substantial proportion of variation in key metrics of biophony—whereas anthropophony (speech and traffic) show less predictable patterns. Compared to pristine sites, the soundscape of urban green spaces is more dominated by technophony and less diverse in terms of acoustic energy across frequencies and time steps, with less instances of quiet. We conclude that the global soundscape is formed from a highly predictable rhythm in biophony, with added noise from geophony and anthropophony. At urban sites, animals experience an increasingly noisy background of sound, which poses challenges to efficient communication.

Many animals communicate by sound. Such vocal communication occurs against a backdrop of sounds of biological origin (biophony; for example, vocalizing and stridulating animals), of geophysical origin (geophony; for example, rain or wind) and of human origin (anthropophony; for example, speech and traffic)[1]. The resulting distribution of sound amplitudes across frequencies partially determines how efficiently animals can communicate with each other[1,2], thus shaping acoustic communities over evolutionary time[3–5]. In ecological time, changes in the acoustic environment can generate alterations in species behaviour, interactions and communication patterns[6].

Altogether, the structure of the acoustic environment is defined as the soundscape[7,8]. A soundscape is the collection of all sounds—biological, geophysical and anthropogenic—that occurs at a place and within a given time frame, and is perceived by living organisms, including humans[9]. Studies of soundscape ecology thus focus on variation in acoustic properties across space, time and spectral characteristics[1,2,10]. Importantly, contributions from biophony, geophony and anthropophony may differ in their patterns of variation.

The biophonic soundscape varies in both time and space, with periodic acoustic patterns across the year (seasonality) or within a day (diel patterns) defining "the rhythms of nature"[2]. Well-characterized temporal cycles of communication are found in the vocalization of birds, amphibians and insects. Many such species tend to start singing at the same time each year[11], and to sing most intensely early in the morning[12] and late in the evening. These peaks are referred to as the dawn and dusk chorus, respectively. The dawn chorus tends to be dominated by birds and amphibians, whereas the dusk chorus is dominated by insects[1]. In addition, the timing and length of both the dawn and dusk

✉e-mail: otso.t.ovaskainen@jyu.fi

| Index (class) | Description | Method | Audio sample 1 | Audio sample 2 |
|---|---|---|---|---|
| Low (A/G) | Acoustic energy 0–1 kHz. | Logarithmic average energy computed from spectrogram. | 23.500 | 13.320 |
| Middle (A/B) | Acoustic energy 1–10 kHz. | Logarithmic average energy computed from spectrogram. | 3.667 | 14.459 |
| High (B) | Acoustic energy 10–20 kHz. | Logarithmic average energy computed from spectrogram. | 5.344 | −3.738 |
| ACI (B) | Acoustic complexity index[31]. Avian vocal activity against environmental noise (specifically traffic and wind). | Summary metric of differences in intensity between adjacent time bins in a spectrogram. | 1,899.2 | 1,901.3 |
| ADI (NA) | Acoustic diversity index[32]. Diversity of acoustic energy across all frequency bands in recording. | Spectrogram divided into 1-kHz bands and Shannon entropy computed along time over all frequency bands. | 1.138 | 1.975 |
| Bio (B) | Bioacoustic index of avian abundance[29]. Acoustic energy in the range within which most bird vocalizations occur. | Area under the curve of the normalized amplitude values over the frequency range occupied by most birds (2–8 kHz). | 1.362 | 4.003 |
| NDSI (B/A) | Normalized difference soundscape Index[33]. Measure of anthropogenic disturbance on the soundscape. | Ratio of acoustic energy in frequency band 1–2 kHz to energy in frequency band 2–8 kHz. | −0.021 | 0.886 |
| H (NA) | Entropy index[34]. Overall variation in soundscape. | Product of Shannon entropies computed along time and along frequency. | 0.921 | 0.900 |
| Speech (A) | Speech detection, maximum probability. | AudioSet event class detected by YAMNet for every 1-min recording. | 0.615 | 0.133 |
| Vehicle (A) | Vehicle detection, maximum probability. | AudioSet event class detected by YAMNet for every 1-min recording. | 0.827 | 0.053 |
| Silence (B/G/A) | Silence detection, maximum probability. | AudioSet event class detected by YAMNet for every 1-min recording. | 0.212 | 0.005 |
| Wind (G) | Wind detection, maximum probability. | AudioSet event class detected by YAMNet for every 1-min recording. | 0.870 | 0.070 |
| Rain (G) | Rain detection, maximum probability. | AudioSet event class detected by YAMNet for every 1-min recording. | 0.866 | 0.024 |
| Animal (B) | Animal detection, maximum probability. | AudioSet event class detected by YAMNet for every 1-min recording. | 0.960 | 0.990 |
| BirdSpecies (B) | Number of bird species detected. | Bird species were identified using BirdNET-analyzer v.2.4; see Methods for details. | 0 | 4 |

**Fig. 1 | Acoustic indices used to track the rhythms in the global soundscape.** Top: two example spectrograms with the signatures of different sound sources highlighted and identified. Bottom: table lists and briefly defines 15 acoustic indices of the soundscape. In the table and in all subsequent figures, we sort indices by their type (how they are calculated), whereas letters clarify how they are assigned: B, biophony; G, geophony; A, anthropophony; or NA, no clear classification. The values of each index for the example audio clips are given in the last two columns. sp, species.

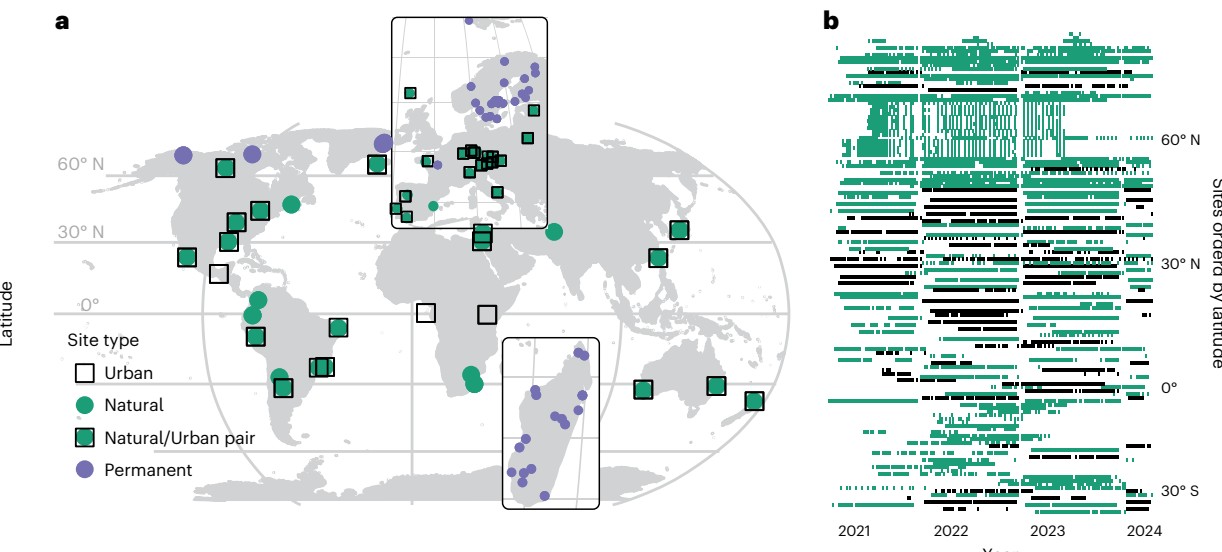

**Fig. 2 | Sampling design and coverage. a**, Across the world, we used up to five AudioMoth[31] samplers to record the local soundscape at each of 139 sites. To characterize differences between natural sites (filled green circles) and urban green spaces (open black squares), 83 of the sites were part of a paired design, with each habitat type sampled for a year within a distance of 4–50 km. To avoid the effects of specific years, the starting habitat (natural or urban green space) was randomized between sites. For part of the material (violet symbols), the habitat was fixed to natural sites alone. **b**, This resulted in multi-annual time series for individual sites (*y* axis), with the coverage of individual weeks shown by squares across time on the *x* axis. Habitats are coloured green for natural and black for urban green space.

chorus differ by latitude and time of the year[1,13]. Across seasons, each ecosystem may have a predictable soundscape phenology related to the seasonal activity of sonically active animals.

Humans contribute to the soundscape through both speech and the sounds of devices (such as traffic and machines). The resulting anthropophony may show both diel and seasonal patterns. For example, the sounds of highways are likely to be loudest during the start and end of the business and school day[1,14]. Beyond direct emission of sound, human impacts on soundscapes may also be indirect, causing changes in the vocalizing fauna through light pollution, climate change, habitat alteration or modifications to the animal community and the habitats that they rely on[15,16]. As a result, anthropogenic impacts on the global soundscape are rapidly spreading. Clearly, the impacts are likely to be greatest in urbanized areas[14,17], and local patterns and processes in soundscapes have been characterized by refs. 18,19. Overall, the urban soundscape is driven by the structure of land use and, across cities, the amount of vegetation—a metric of green infrastructure—correlates with the intensity of sound[20]. Vegetation absorbs sound energy and reduces anthropogenic sound pollution. Vegetation also attracts animals and increases the biophony. This begs the question of whether smaller green spaces may suffice to preserve key features of intact soundscapes. The relationship between green spaces and the emergent soundscape, and how the soundscapes of urban green spaces compare to nearby natural areas, are thus of key interest[1].

Large-scale acoustic surveys can now be undertaken with relative ease to generate vast datasets, offering the potential for resolving a wide set of fundamental ecological questions[21] and resolving impacts of conservation concern[22–24]. One of the key tasks for the study of soundscapes is deciding how to characterize the massive amounts of audio data[9]. The last decade has yielded a diverse set of acoustic indices (summarized by ref. 9, with a critical review in ref. 25). Overall, these indices are aimed at representing the level of complexity, diversity, energy and/or potential sound sources (biological versus non-biological). Individual indices are designed to describe different features in the distribution of sound across amplitudes, frequencies, time or all three dimensions[1,9,26] (Fig. 1). As each acoustic index measures a different aspect of the distribution of acoustic energy, broader inferences on

soundscape patterning may be more accurate when several acoustic indices are considered simultaneously and compared[1,9,26].

Here, we analyse variation in the global terrestrial soundscape in each type of index, and in bird communities scored by automated species detection from sounds (Fig. 1). Specifically, we ask the following questions: (1) How pronounced and predictable are global biophonies and anthropophonies across the latitudinal, diel and seasonal dimensions? (2) How do soundscapes differ between natural sites and urban green spaces, and what acoustic indices are these differences reflected in? We examine each of these questions using global soundscape data and a paired urban–rural experimental design. The data comprise more than a million minutes of acoustic data from 139 sites throughout 6 continents, a latitudinal gradient of 116° and a time period of 3 years (Fig. 2 and Supplementary Fig. 1).

## Results

Global acoustic recordings revealed strong seasonal and diel rhythms in the soundscape. Different acoustic indices showed different rhythms in terms of both amplitude (size of oscillations) and timing (Fig. 3 and Supplementary Figs. 2–55). Among individual acoustic indices, the predictability of patterns differed substantially (Fig. 4a; compare overall $R^2$). Differences emerged in the relative importance of diel and seasonal cycles (Fig. 4a; blue section), as compared to added signals of anthropogenic impacts (Fig. 4a; black and green sections, corresponding to positive and negative effects of the human footprint index[27], respectively) and climatic conditions (Fig. 4a; yellow section, corresponding to the joint impacts of elevation[28], average temperature[29] and precipitation[29]). Metrics directly related to animal activity proved best predictable by latitude, season and time of day. These metrics included animal activity (Fig. 4a; second blue bar from the right), which mainly consisted of sounds of wild animals (Supplementary Fig. 56), as well as the bioacoustic index of avian abundance[30] (Fig. 4a; sixth bar from the left).

Acoustic phenology was pronounced in patterns of both biophony and geophony. In both the Northern and the Southern Hemispheres, the peak of bird acoustic activity occurred during the local spring (Fig. 3). Sounds of wind were more common during the local winter than summer (Fig. 3).

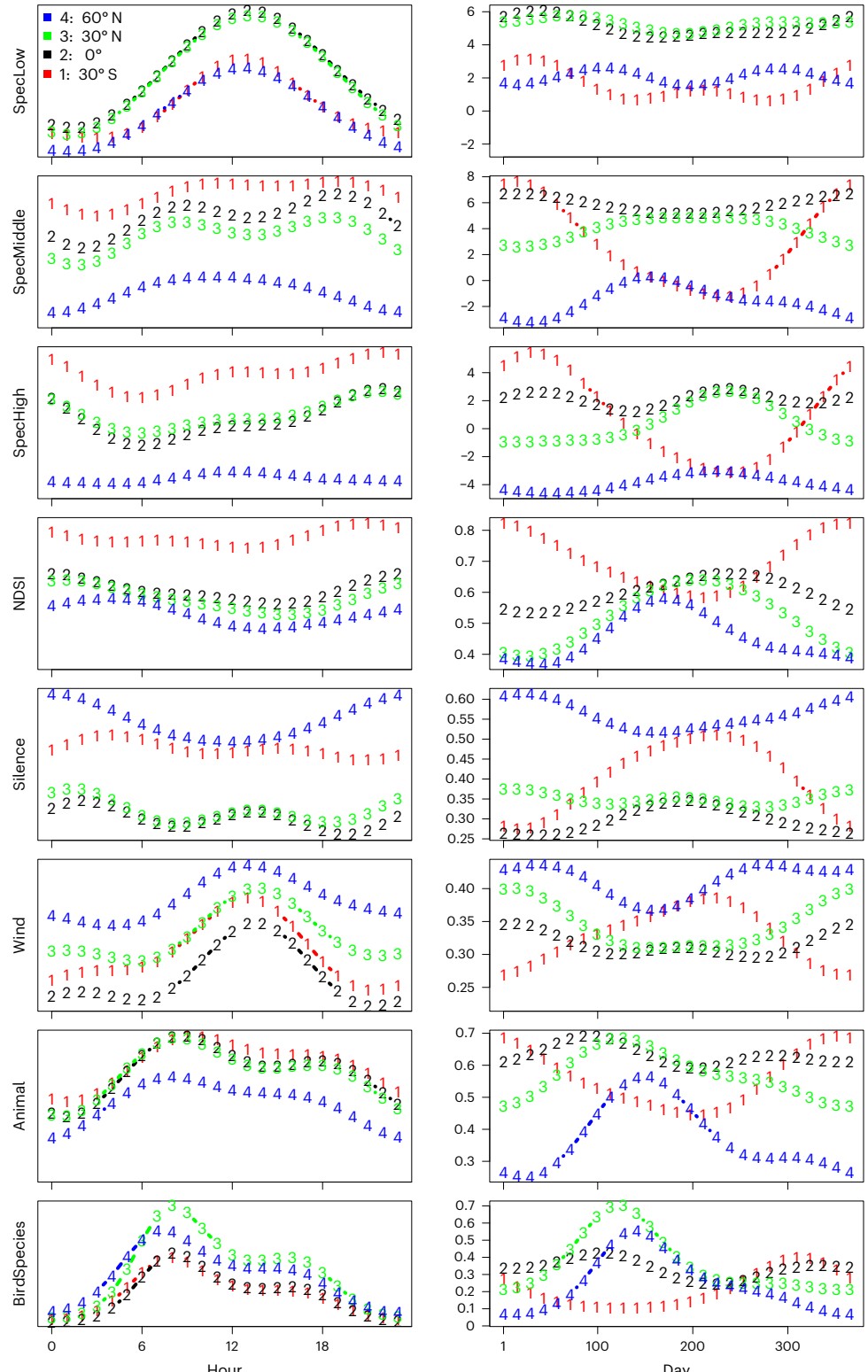

**Fig. 3 | Seasonal and diurnal variation in different acoustic indices.** Predictions for the hour of the day (left) and day of the year (right) for 30° S, 0°, 30° N and 60° N. The predictions shown are based on the global model fitted to all data and shown for those eight indices for which the seasonal and diurnal patterns explained a substantial part of the variation (Fig. 4). Here time is represented by local absolute time; for patterns with respect to time relative to sun time, see Supplementary Fig. 57. Site-specific results for all acoustic indices are shown in Supplementary Information 1. For definitions of each index, see Fig. 1.

Diel patterns were likewise pronounced. Across the day, the shapes of acoustic energy, species richness and call rates were typically bimodal. Here, we observed a larger peak in the morning and a smaller peak in the evening (Fig. 3), corresponding to the dawn and dusk chorus, respectively. The highest prevalence of quiet periods (reflected by index Silence; Methods) was observed during the night and sounds of wind were more common during the day than night (Fig. 3).

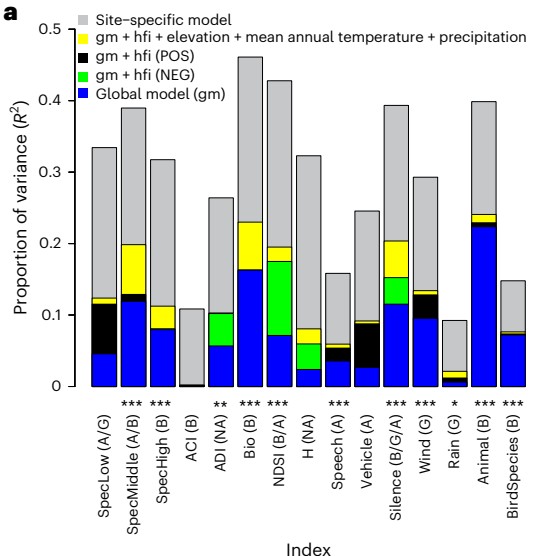

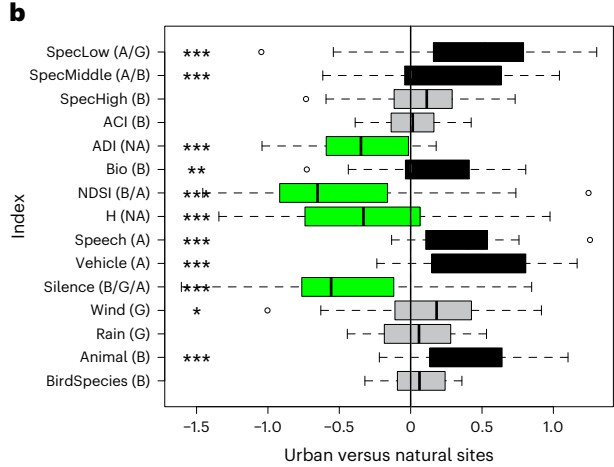

**Fig. 4 | Predictability of 15 acoustic indices in space and time and differences across pairs of natural versus urban sites. a**, Predictability of 15 acoustic indices in space and time. The total height of each bar shows the proportion of variance explained ($R^2$) by site-specific models. The blue section shows variation explained by the latitude of the site using a global model (gm). The significance of latitude was determined by a permutation test (two-sided; 139 sites; 1,000 permutations; no adjustment for multiple comparisons), with index-specific $P$ values as follows: 0.733, 0.001, 0.001, 0.693, 0.003, 0.001, 0.001, 0.359, 0.001, 0.106, 0.001, 0.001, 0.018, 0.001 and 0.001. Significance levels are \*$P \le 0.05$, \*\*$P \le 0.01$ and \*\*\*$P \le 0.001$. Black (positive effect, POS) and green (negative effect, NEG) sections show the increase in $R^2$ with the addition of the site-specific human footprint index (hfi) to the global model. The yellow sections indicate the increase in $R^2$ when climatic conditions (elevation, mean annual temperature and precipitation) are added to a model including hfi. **b**, Differences observed across

pairs of natural versus urban sites. Each box shows the distribution of empirically observed pairwise differences between the two sites within a pair—with a box for the interquartile range (50% of data), a vertical line for the median; whiskers for the minimum and maximum up to 1.5× interquartile range from the box; and individual data points for outliers beyond this range. Since the difference is calculated as urban minus natural values, a positive value indicates higher values for urban sites (significant differences shown by black boxes) whereas a negative value indicates higher values at natural sites (significant differences shown by green boxes). Index-specific $P$ values from two-sided $t$-tests across 36 paired values (without adjustments for multiple comparisons): $1.06 \times 10^{-5}$, $7.91 \times 10^{-4}$, $1.46 \times 10^{-1}$, $8.60 \times 10^{-1}$, $1.89 \times 10^{-7}$, $3.06 \times 10^{-3}$, $2.45 \times 10^{-5}$, $5.71 \times 10^{-4}$, $9.51 \times 10^{-9}$, $3.46 \times 10^{-8}$, $5.41 \times 10^{-6}$, $2.65 \times 10^{-2}$, $3.27 \times 10^{-1}$, $1.10 \times 10^{-6}$ and $6.07 \times 10^{-2}$ (with asterisks as in **a**). Index letters as in Fig. 1. Diel variation was modelled by local absolute time; for patterns with respect to sun time, see Supplementary Fig. 58.

Against the backdrop of seasonal and diurnal rhythms in the soundscape, we found variable imprints of human footprint (Fig. 4a). For some acoustic indices (notably sounds in the low and middle part of the spectrum, human speech, noise from vehicles, wind, rain and the number of animal species), we found greater values with a greater human footprint index (Fig. 4a). For other indices (ADI, NDSI, H and Silence; for definitions, see Fig. 1), we found a decrease with human footprint. Thus, sites with a greater human footprint are noisier and characterized by a less diverse soundscape. Patterns in acoustic indices differed strongly between our sampling sites in the Northern and Southern Hemispheres. Southern sites were characterized by greater levels of high- and middle-frequency animal sounds, and northern sites by greater levels of low-frequency sounds and lack of silence (Fig. 3).

To resolve differences in the soundscape between urban green spaces and nearby pristine sites, we next examined the distribution of pairwise differences in predicted values for each acoustic index. Different indices showed different patterns (Fig. 4b and Supplementary Fig. 58b), with some general denominators:

First, consistent with Fig. 4a, many indices showed greater values in urban green spaces than in the paired natural site (Fig. 4b). As expected, these indices included Speech and Vehicle. Among site pairs, the acoustic energy generated by biophony (index Bio) and energy in all spectral classes was larger in urban green spaces than at their paired natural sites. For spectral energy, the difference was largest in the low-frequency band and decreased towards higher frequencies. NDSI, H, ADI and Silence showed greater values in natural environments than in nearby urban green spaces (Fig. 4b).

Second, across site pairs, anthropogenic sounds (Vehicles and Speech) were present at most sites regardless of whether the site was

classified as more pristine or urbanized (Fig. 4 and Supplementary Figs. 2–55). Thus, at most sites, a variable but typically substantial part of the soundscape was generated by human activities (Supplementary Figs. 2–55). Anthropogenic sounds showed little seasonal patterning, but more patterning with the time of the day (Supplementary Figs. 2–55). Sounds of vehicles and human speech were concentrated to daylight hours, whereas periods of quiet were more frequent near midnight (Supplementary Figs. 2–55). Across other indices, urban green spaces generally showed greater values of anthropophony (indices SpecLow, Speech and Vehicle) than their more pristine counterparts (Fig. 4b and Supplementary Figs. 2–55).

Third, bird species richness and animal vocalization activity showed greater values in urban green spaces than at nearby natural sites, with a statistically significant difference in animal vocalization (Fig. 4b). Despite the difference in overall species richness, the sets of bird species detected in urban green spaces versus at more natural sites were partly complementary. This is revealed by the rates at which new species accumulate with increasing numbers of observations (Fig. 5c,d). With more sounds detected, we see a greater (but more variable) mean number of species detected in urban green spaces than in their natural counterparts (Fig. 5c,d; compare black and green lines). When detections are sampled randomly across urban green spaces and their paired natural sites, we score greater species richness for the same number of sound observations (Fig. 5c,d; blue lines). Thus, for a constant effort, partly different species are being detected in the two environments. A difference in species composition between the two environments was also revealed by further analyses of species trait composition (Supplementary Figure 62). Inter alia, urban sites were characterized by a greater proportion of seed-feeding species,

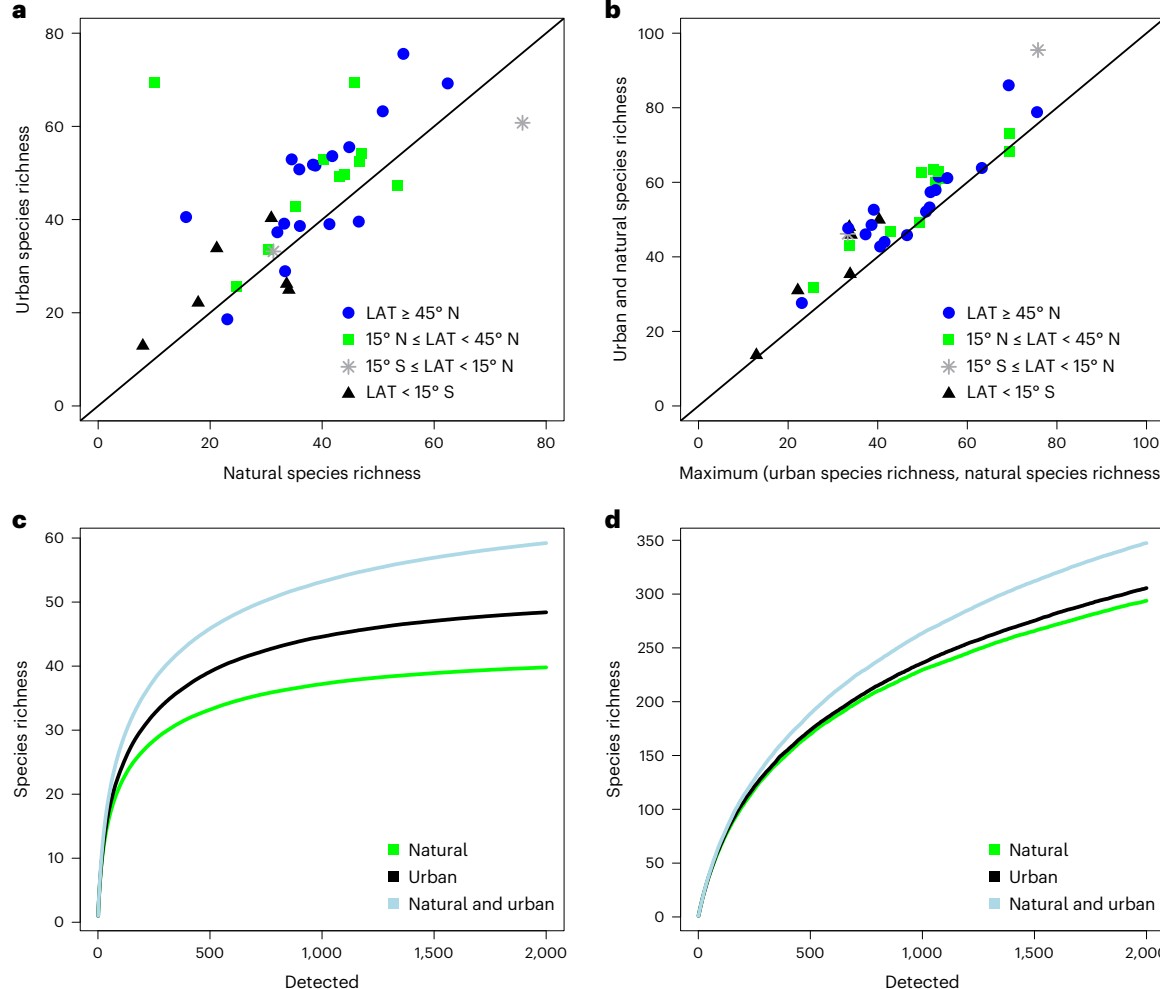

**Fig. 5 | Species richness for 36 site pairs composed of natural and urban sites.** **a,b**, Species richness for individual urban and natural sites addressing whether urban or natural sites have more species (**a**) and whether urban and natural sites have different species (**b**). **c,d**, Species richness as a function of the number of bird detections sampled separately for each site after which we averaged the resulting curves (**c**) and in samples from pooled natural and pooled urban sites (**d**). Green curves emanate from natural sites and black curves from urban green spaces. Light-blue curves are based on data combined across the natural site and its paired urban green space, and thereby represent sampling of recordings irrespective of the environment of origin. The data include bird detections (BirdNET confidence threshold 0.8) for which recordings were available for both urban and natural sites at the same day of the year and the same time of the day within each pair. The resulting data have been sampled with replacement. In **a** and **b**, species richness *S* is defined as the number of distinct species in 1,000 detections, whereas **c** and **d** show the accumulation of species richness detected up to 2,000 detections. All results are averages across 50 replicates. LAT, latitude.

by species associated with human-modified habitats, by omnivores and by species with a generalist lifestyle. Overall, bird species richness appeared to increase with latitude across site pairs, but this pattern was probably caused by a technical artefact (that is, that the accuracy of the bird detection software used varies in space; see 'Discussion').

The results shown in Figs. 3 and 4 are based on evaluating diel rhythms using local absolute time. While this time representation can arguably be considered the relevant measure of time for anthropogenic sounds, it is well known that many biophonic sounds are more closely related to the timing of dawn and dusk than to local absolute time[31]. To test the robustness of our result with the measure of time selected, we repeated our analyses using sun time instead of local absolute time. A comparison of the results (Fig. 3 versus Supplementary Fig. 57 and Fig. 4 versus Supplementary Fig. 58) shows that the latitudinal, seasonal and diel rhythms (or the lack of them) reported in Figs. 3 and 4 are robust with respect to this choice.

## Discussion

Through our systematic, world-wide and long-term acoustic survey, we characterized the drivers of the global terrestrial soundscape. We find highly predictable rhythms in multiple indices of sounds of biological origin. For animal vocalizations, we could accurately predict their presence in a soundscape on the basis of knowing merely the latitude, the date and the time of day. Against these regular rhythms in biophony, acoustic indices reflecting anthropogenic impacts proved less predictable by the same factors: rhythms in anthropophony are structured by daylight hours rather than latitude. Consistent with a discordance between biophonic and anthropophonic rhythms in the soundscape, we found the soundscapes of urban green spaces to be more dominated by technophony and less diverse in terms of sound amplitudes and frequencies than more natural, paired sites. Urban sites also showed less instances of quiet. Below, we will return to each finding in turn.

### Biophony is structured by latitude, date and time of day

Throughout the day and year, most of our passive acoustic recorders detected some sound. Moments of quiet mainly occurred close to midnight and at high latitudes during the local winter (Fig. 3 and Supplementary Figs. 13 and 14). For the biophonic part of the soundscape, our analysis of the global soundscape revealed strong rhythms

in space and time—as structured by latitude. These patterns proved best predictable in two indices: animal sounds detected (index Animal) and acoustic energy associated with bird vocalizations (index Bio[30]). Thus, based on knowing the latitude, the date and local time alone, we will be able to characterize key features of the biophonic soundscape. The patterns observed are consistent with the concept of predictable soundscape phenology[1] reflecting the seasonal activity of sonically active animals. As major animal sonic groupings (amphibians, birds and insects) have different phenology and different acoustic patterns (pulsating and melodic), multiple rhythms in the soundscape may weave into each other during the year[1]. With spring arriving at different times at different latitudes, local soundscape phenology will differ among ecosystems—but our findings suggest that they largely follow a consistent pattern. Importantly, the predictability of these patterns also remained consistent regardless of whether we used local absolute time or sun time. In other words, the exact timing will shift with the latitude and the date, but the rhythms remain equally pronounced and predictable.

The rhythms here detected at a global scale are largely consistent with previous work on local and regional soundscapes[1], suggesting that the local peaks in biophonies start before sunrise, fall during the day, then rise again before sunset. Whether nocturnal acoustic activity exceeds activity during the day depended on the index used and the latitude (Fig. 3 and Supplementary Figs. 2–55). At the level of individual acoustic indices, we note substantial differences in patterning and predictability. This is only to be expected, as these metrics were intentionally derived to capture different features of the soundscape[30,32–37]. In illustration of their complementary nature, the acoustic indices are also weakly correlated with each other (Supplementary Fig. 59). As a solution, multiple authors have emphasized the need for calculating and comparing larger suites of indices for understanding the underlying processes[38–42]. Our findings fully support this view.

In retracing the drivers of soundscape variation, we explored how they affect the numbers of sonically active species, and—for the urban–natural comparison (see below)—the types (traits) of species recorded. What they revealed were strong rhythms in the richness of species contributing sounds (Fig. 3). At present, this type of analysis can only be achieved for birds, although methods for the automated identification of vocalizations by amphibians and insects are quickly developing[43,44]. Yet, in identifying the species behind the vocalizations, a choice must be made. The assignment of a sound to a species is based on a probability that this is the right 'label'. If the label is correct, then a record of the vocalization is a true positive. By then selecting a too-stringent probability threshold before trusting the label, one increases the risk of neglecting correct assignments (causing false negatives), whereas by setting the threshold too low, one increases the risk of accepting false labels (false positives). Thus, the choice of a threshold reflects a trade-off between false positives and false negatives. For this reason, we performed the detection using two different thresholds. Notably, the shapes of both diurnal and seasonal patterns remained highly similar across threshold values (Supplementary Figs. 52–55)—even though the confidence threshold naturally affected the absolute number of trusted detections.

As a methodological caveat, bird species richness was estimated to be greater in temperate than tropical regions (Fig. 3). This pattern is probably a technical artefact, since the bird detection software used[37] is better trained to North American and European bird species, and unable to detect species from other regions with the same accuracy. Importantly, this does not invalidate patterns over seasons or times of the day. Given constant detectability of each species within a site, the rhythm remains without any systematic bias—even if its baseline level may be underestimated for tropical regions. When using the broader category of 'animals', there are more detections in the Southern Hemisphere compared to Northern Hemisphere (Fig. 3). Such a discrepancy illustrates the potential for improved use of present-day

data and the scope for future use of data generated by the ongoing sampling revolution[45]—even when analytical methods are lagging behind. Naturally, all acoustic data generated in our project (some 90 million minutes) are stored in their raw format (some 0.7 petabyte of .wav data). In the terminology of ref. 2, these recordings will become 'acoustic fossils', to be revisited once improved identification models are available.

## Anthropophony conflicts with biophony

Where biophony varied predictably with latitude, day of year and time of day—with peaks around dusk and dawn—anthropophony showed a different pattern (Supplementary Fig. 21b,d). Here, both speech and vehicle noise peaked around mid-day, whereas the main predictor of traffic noise was the human footprint index. Yet, vehicle noise increased early and late enough in the day to coincide with the dawn and dusk choruses of biophonic origin (compare Supplementary Fig. 55 to 48b). These findings were further supported by our comparisons of soundscapes between urban green spaces and more pristine sites.

Overall, urban green spaces are clearly affected by human sounds. Compared to pristine sites, the soundscape of urban green spaces proved more dominated by technophony and less diverse in terms of acoustic energy across frequencies and time steps, with less instances of quiet (Figs. 3 and 4). Most importantly, there were few places on the globe where anthropogenic sounds did not enter the local soundscape. Such anthropophony showed either little rhythm in time, or a rhythm substantially different from that generated by animals and abiotic forces (Fig. 3 and Supplementary Figs. 2–55). As human activities are centred on daylight hours, they differed from for example bird song, which is centred on dawn and dusk[46,47] (Fig. 3). Despite regional variation in the degree of urbanization, increasing human population densities will thus come with new and globally consistent challenges for animal communication. From a human perspective, urban green spaces may provide a partial respite from the sounds of machines and traffic[4]. Globally, urban green spaces are impacted by anthropogenic sound and are nowhere near quiet (Fig. 3).

The impacts of anthropogenic sounds on local wildlife are generating increasing interest and insight[48–51]. Such anthropophony will mostly contribute high-intensity, low-frequency (500 Hz to 4 kHz) sounds to the soundscape. Since many terrestrial animals (for example, birds, anurans and insects) communicate in this frequency range, anthropophony may efficiently mask animal communication. In terms of adaptations, a recent review[52] found that insects were unable to adjust any features of their acoustic signals to overcome noise. Anurans, on the other hand, were able to increase call intensities only, whereas birds were able to make a variety of adjustments including increasing dominant, minimum and/or maximum frequencies, increasing note duration and increasing the amplitude of their songs[53,54]. Several species of birds are also known to switch their singing towards quieter night-time[16,55].

One feature of the global soundscape did surprise us: urban green spaces were characterized by greater animal vocalization activity than their natural counterparts and by a higher species richness of vocalizing birds. To uncover the reasons, we validated that differences in bird species richness were not caused by the misclassification of anthropogenic sounds. To achieve this, we had an ornithologist listen through 615 sound clips from two natural and two urban sites (Supplementary Fig. 60 and Supplementary Table 1)—while annotating all sounds detected. As expected, some of the automated identifications turned out to be misclassification, but the difference between urban and natural locations persisted and cannot be attributed to any artefact (Supplementary Fig. 60 and Supplementary Table 1).

Importantly, the difference observed in the bird fauna related to species counts alone, which may partly reflect differences in detectability. With added noise, birds may increase both the frequency and amplitude of their songs[52–54]. At the same time, we found the

two environments to sustain partly different types of bird species. This was evidenced by differences in species traits between habitats (Supplementary Text 61 and Supplementary Table 2). Thus, we believe that the effect on overall diversity can be retraced to the structure of urban green spaces. They will typically consist of variable habitats intermixed, with ample edge zones[26,30]. Such environments are known to attract a mix of species of different geographical origins and habitat affinities[56–59]. In further support of this claim, we found that urban green spaces and their natural counterparts differed in bird species composition, with a more diverse set of species detected across the two environments than within either one on its own (Fig. 5). These patterns are fully consistent with the finding that birds vary widely in their sensitivity to human-dominated environments, with some being highly tolerant and other restricted to pristine environments[60].

## Conclusions

Overall, we find strong diurnal and seasonal rhythms in the global biophonic soundscape. In terms of their predictability, these rhythms are discordant with global patterns in anthrophony—and in terms of their timing, biophonic and anthropophonics rhythms are partly overlapping. This is a challenging scenario for the evolution of adaptations in animal signalling[49–51,54]. Given the potential for anthropophony to mask current animal communication and its strong contributions to global soundscapes, it is likely to exert a strongly selective force. Yet, poor predictability in space and time will compromise such selection and some taxa seem unable to adjust their acoustic signals to overcome noise[52]. The implications of our findings for conservation management, urban planning and even human–nature interactions are stark. Securing intact soundscapes through joint management is a priority for safeguarding animal communication[61]. For humans, hearing is one of our key senses[62], and the acoustic environment a key part of how we experience the world. Thus, protecting its structure is essential for upholding both ecosystem integrity and human health[1,63].

## Methods

### Data

Acoustic data were collected in the context of the project LIFEPLAN, an international initiative for characterizing biodiversity across the globe (except Antarctica). For a full description of the sampling design and the specific protocol implemented using audio recorders, see ref. 64. For the present study, we extracted acoustic data recorded between years 2021 and 2024 at 139 different sites in 6 continents. A list of the sites and their locations is provided in Supplementary Table 3. At each site, data were collected via passive acoustic monitoring using AudioMoth v.1.1 devices[65]. For any one time, there were up to five AudioMoth devices operated per site within a 1-ha area[64]. The specific sites were chosen by local teams. The natural locations were chosen to be the most natural ones present within a restricted distance from the urban one—just like the urban sites were classified as urban by regional, not global or absolute standards. Importantly, these are the urban versus natural sites accessible to the local human population, and thus represent the soundscapes that people can realistically switch between. The total number of recordings varied between the sites due to equipment malfunctioning and logistic constraints, such as site accessibility due to road damage caused by hurricanes and storms. At some sites, the sampling period included only a few months, whereas in other sites the sampling was continuous throughout the year. The data were subsampled to the level of full hours. To allow the sampling of equivalent time periods, we ensured synchronization among individual recorders. To this aim, all AudioMoth devices were synchronized weekly with coordinated universal time. From each hour recorded, a 1-min-long clip was randomly selected. Consequently, the data to be analysed consisted of up to 24 recordings per day per site. The total number of the 1-min recordings used in the present analysis was 1,484,181.

### Preprocessing

Audio data were collected with a 48-kHz sampling rate using 16 bits per sample. Data were stored in AudioMoth[65] devices as encrypted files. Thus, potential human voices present in the audio recordings were not available to anyone accessing the memory card of the physical recorder at the sampling site. Encrypted data were transferred to object storage Allas at CSC–IT Center for Science, Finland. Before analysis, the files were decrypted into standard WAV files. Owing to artefacts in the beginning of some recordings, all data were processed so that the first second of a recording was removed.

### Spectral energy

Spectrogram was computed from a 1-min recording using a 1,024-point fast Fourier transform with Hann window (21.3 ms) and 480-point time hop (10 ms) using the Python library librosa v.0.9.2.33. The result was summarized with 20 bins, each bin covering a 1-kHz frequency band. Both maximum and mean energy of each bin were used to summarize the 1-min recording. For analysis, the spectral energy was divided into three frequency bands: low (0–1 kHz), middle (1–10 kHz) and high (10–20 kHz). The final values were represented on a logarithmic scale (dB).

### Acoustic indices

Five acoustic indices were computed for each recording: the acoustic complexity index (ACI)[32], the acoustic diversity index (ADI)[33], the bioacoustic index (Bio)[30], spectral and temporal entropy (H)[35] and the normalized difference sound index (NSDI)[34]. Each index was computed using Python package Acoustic_Indices[66] with default values, except ADI for which the noise floor value was set to −26 dB meaning that all amplitude values below 5% of maximum value were considered as noise. For ACI we experimented with several values for the window length and window hop (512, 1,024 and 2,048 samples corresponding to 10, 21 and 43 ms in time), visually comparing the results obtained with different parameter settings using scatterplots. As the plots did not suggest superiority of any particular parameter setting, the final ACI results were based on a window length of 512 samples. Before calculating each index, acoustic data were filtered using a high-pass filter with a 300-Hz cutoff frequency to suppress low-frequency disturbances.

### Acoustic event detection

Recordings were processed with YAMNet[36] to detect six AudioSet event classes: speech, animal, vehicle, silence, wind and rain. Both maximum and mean probabilities of each class were calculated for every 1-min recording.

### Bird species identification

Bird species were identified using BirdNET-Analyzer v.2.4 (ref. 37). The results were restricted by the site-specific species lists. These lists were generated using script species.py of the BirdNET-Analyzer package. Two confidence thresholds (0.3 and 0.8) were used to filter the detections. All statistical analyses were performed on the basis of detections using the threshold of 0.8.

### Statistical analysis

To quantify periodic rhythms in different soundscape indices, we constructed separate generalized linear models for each acoustic index, using as predictors the day of the year and the local absolute time of the day. To arrive at the relevant periodic functions of seasonal and diel variation, we used an intercept and eight explanatory variables in each model: $\cos(2\pi \times Day)$, $\cos(4\pi \times Day)$, $\sin(2\pi \times Day)$, $\sin(4\pi \times Day)$, $\cos(2\pi \times Hour)$, $\cos(4\pi \times Hour)$, $\sin(2\pi \times Hour)$ and $\sin(4\pi \times Hour)$. Here, Day was the day of the year divided by 365 and Hour was the hour of the day divided by 24. All models were fitted separately for each site using lm in R for continuous data and glm(family=poisson) for count data. In calculating $R^2$ values as measures of predictability, we used linear models also to count data.

To evaluate the predictability of local patterns from global ones, another model was fitted where the data from all sites were present in a single global model. In this model, we modelled latitude-dependency of seasonal and diel patterns by including as predictors the interactions between second-order polynomial of latitude and the eight explanatory variables of seasonal and diel variation described above. To evaluate anthropogenic and climatic impacts, we included the main effects of four additional predictors: the human footprint index[27], elevation[28], annual mean temperature[29] and annual precipitation[29]. The significance of the latitude as a predictor was calculated using a permutation test with 1,000 permutations of latitude values over sites.

For pairs of urban green spaces and nearby natural environments (36 pairs), a linear model was constructed that included the habitat type as a factor. Here, we used the acoustic index as the response variable, whereas the explanatory variables included an indicator of whether the site was an urban green space or a natural site. As the other explanatory variables, we used the eight seasonal and diel predictors explained earlier. The model was fitted separately for each site pair and the contrast between the coefficients of urban green spaces and natural sites was calculated for every acoustic index.

Species richness for 36 site pairs composed of natural and urban sites was investigated using species accumulation curves. Data included bird detections (BirdNET confidence threshold 0.8) for which recordings were available for both urban and natural sites at the same day of the year and the same time of the day within each pair. To score the number of detections for a given number of recordings, we sampled the data with replacement for each site or combination of sites.

**Local absolute time versus sun time.** To test for differences in patterns with respect to local absolute time and versus time relative to sunrise and sunset, we fitted the local and global regression models described above to data using both time representations. To obtain the time relative to the sun cycle, the original time stamp on the recording was converted to a new time on the basis of the information of the sunrise and sunset at the recording site, obtained using R package suncalc, v.0.5.1. The hour of the day was mapped so that the sunrise corresponded to 06:00 and sunset to 18:00. For the time between the sunrise and sunset (daytime), local absolute hours were mapped linearly between 06:00 and 18:00 and for the time between the sunset and sunrise (night-time), the local absolute hours were mapped linearly between 18:00 and 06:00.

### Reporting summary

Further information on research design is available in the Nature Portfolio Reporting Summary linked to this article.

## Data availability

The data are available via Zenodo at https://doi.org/10.5281/zenodo.15369637 (ref. 67).

## Code availability

The scripts and code to reproduce the results of this paper are available via Zenodo at https://doi.org/10.5281/zenodo.15372067 (ref. 68).

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

## Acknowledgements

The contributions of M. González, Z. Colón-Piñeiro, P. Forero, K. Borja-Acosta, Y. Ochoa, L. Dary Echavarria and A. Diaz-Pulido are gratefully acknowledged, as all being part of the Lifeplan Colombia collaborative consortium. The following people and organizations provided invaluable logistical support, supervision or facilitation of sampling: P. Leitman; T. Berg; J. Silby; L. Burnside; personnel of the Reserva Hídrica Natural Municipal Los Manantiales; D. Serra (Botanical garden of Córdoba city, Argentina); Asociación Civil Los Manantiales; M. Costamagna; J. Castellanos (Baja California Sur, Mexico); ICTS-RBD; the Doñana Monitoring Team (Spain); F. Kettenhumer and E. Pröll (Kalkalpen National Park); G. Kjærstad; T. Prestø; M. Bendiksby; F. Ødegaard; K. Hårsaker (Trondheim, Norway); Joint Arctic Command (Nuuk, Greenland); Israel Ministry of Science and Technology; trainees and research assistants at Seili (UTU, Finland); Y. Miyagi, E. Katori and E. Narita (Japan); M. Urbina

and J. L. Rangel (Chiapas, Mexico); J. Tison; N. Ibagón-Escobar; Companhia das Lezírias, S.A., especially R. Alves (Lisbon, Portugal); the Tiputini Biodiversity Station field, especially C. Valle Piñuela, F. Macanilla and J. Llerena; S. Finnie and T. Vlasata (Czechia); the eLTER network (long-term ecological research); J. Kökkö and D. Sundberg (Lammi, Finland); Instituto Humboldt, Reserva Natural y Cascadas Los Tucanes (RNCLT) and Corporación Salvamontes (Colombia). Identifications of bird calls were contributed by J. Gaviraghi. Land access permissions were provided by Ngāti Whātua Ōrākei (Pourewa, New Zealand); University of Auckland (Oratia Reserve, New Zealand) and Te Kawerau-a-Maki (as having mana whenua for Oratia Reserve, New Zealand). The Guelph field site is located within the Between the Lakes Purchase (Treaty 3); the treaty lands and territory of the Mississaugas of the Credit. It is recognized that the Anishinaabe and Hodinöhsö:ni' peoples have unique, long-standing and ongoing relationships with the land and each other and that the Attawandaron people are part of the archaeological record. Naalakkersuisut, Government of Greenland, granted research and access permits to the Northeast Greenland National Park. The University of Tours granted permission to Parc de Grandmont and the City of Tours to the Forêt de Larçay. Egerton University granted permission to the urban site in Kenya. P.S., T. Roslin, B.L.F., D. Kerdraon, D.R., E.T.R., P. Lauha, P. Lehikoinen, E.P., M.L., V.-M.R. and O.O. were funded by the European Research Council (ERC) under the European Union's Horizon 2020 research and innovation programme under grant agreement no. 856506; ERC-synergy project LIFEPLAN; P. Guedes. and L.L.P. under ERC grant agreement no. 854248; and Team Czechia (PI KS) under ERC grant agreement no. 805189 and GACR 22-17593. In addition, financial support from the following sources is gratefully acknowledged: Estonian Research Council (project PRG1170); Research Council of Finland (decisions 287153, 309581, 324602, 335354, 336449 and 337026); Instituto Humboldt, Reserva Natural y Cascadas Los Tucanes (RNCLT), Metalmaderas Forero and Corporación Salvamontes (operative funding in Colombia); Swedish Research Council (2019-05191); Swedish Foundation for Strategic Environmental Research MISTRA (Project BioPath); Kew Development Funds; Consejo Nacional de Investigaciones Científicas y Técnicas (CONICET, Argentina); State of Tyrol, Department of Environmental Protection; Western Australian Department of Biodiversity, Conservation and Attractions; the Norte Portugal Regional Operational Programme (NORTE2020; project NORTE-01-0246-FEDER-000063; SITES (Swedish Infrastructure for Ecosystem Science); Auckland University of Technology, New Zealand; Government of Northwest Territories, Canada; Ministry of Agriculture, Taiwan; eLTER Plus (INFRAIA, Horizon 2020, agreement no. 871128); MUR Research Projects of National Interest (PRIN) 2022, prot. 20224MZ9HN; CNPq, grant no. 445347/2020-1; the French Polar Institute-IPEV (Program 'Interactions 1036'), Groupe de Recherche en Ecologie Arctique (GREA), ANR (ANR-21-CE02-0024 PACS); SEE-Life program of the CNRS (Ecopolaris-Interactions); the Government of Canada (Open Space grant); Wek'èezhìi Renewable Resources Board (Canada); Canadian Foundation for Innovation—Major Science Initiatives (MSI) Fund; Bavarian State Ministry of the Environment and Consumer Protection; NSERC, Canada Research Chair Program; Région Centre via the Reseau Thematique Recherche (RTR) 'EntomoCentre'; Norwegian Institute for Nature Research; Fundação para a Ciência e Tecnologia (FCT, project UIDB/00329/2020); Jane and Aatos Erkko Foundation; Research Council of Norway (Centres of Excellence Funding Scheme, 223257); Danish Environmental Protection Agency (support to the Zackenberg research station); Italian Ministry of University and Research, project CN00000033); and the University of Tours (France).

## Author contributions

Our contribution addresses global patterns, the resolution of which relies on the time, effort and data contributed by each participating author. Thus, the full project is a team effort. As all co-authors checked the paper and provided comments and corrections as needed, we here detail only the most comprehensive co-author contributions. P.S. performed the analyses. T. Roslin acquired funding, conceived the study, developed the sampling methods and wrote the first draft of the paper. B.L.F. acquired funding and participated in project coordination and data collection. B.H. participated in project coordination and data collection, and managed the data. D. Kerdraon participated in project coordination, logistics and data collection. D.R. participated in project coordination, logistics and data collection. E.T.R. participated in project coordination, logistics and data collection. P. Lauha participated in analyses. L. Griem checked bird identifications. P. Lehikoinen participated in developing the sampling methods. P. Niittynen participated in analyses. G.G.B., A. Farrell and H.M.K.R. participated in project coordination, logistics and data collection. R.S.S.-L. participated in paper conceptualization. O.O. acquired funding, conceived the study, supervised the analyses and co-wrote the paper.

## Funding

## Competing interests

The authors declare no competing interests.

## Additional information

**Correspondence and requests for materials** should be addressed to Otso Ovaskainen.

Panu Somervuo [1], Tomas Roslin [2,3], Brian L. Fisher [4,5], Bess Hardwick [1], Deirdre Kerdraon [2], Dimby Raharinjanahary[4], Eric Tsiriniaina Rajoelison[4], Patrik Lauha [1], Lukas Griem [6], Petteri Lehikoinen [7], Pekka Niittynen [8], Esko Piirainen[7], Markus Lumme[7], Ville-Matti Riihikoski[7], Orlando Acevedo-Charry [9,10], Solny A. Adalsteinsson [11], Maaz Ahmad [12], Sandra Alcobia[13,14], Jón Aldará[15], Nigel R. Andrew [16], Sten Anslan [17,18], Alexandre Antonelli [19,20,21], Julieta Soledad Arena [22], Santiago Arroyo Almeida[23], Ines Aster [24], Hannu Autto[25], Anahi Aviles Gamboa[11], Joaquín Baixeras [26], Mario Baldauf[24], Rosario Balestrieri [27], Gaia Giedre Banelyte[2], Adrian Barrett[28], Pedro Beja [29,30,31], Thomas Olof Berg[32], Benjamin Bergerot [33], Elizabeth G. Biro [11], Pedro G. Blendinger [34], Loïc Bollache[35,36], Magda Bou Dagher Kharrat [37,38], Stephane Boyer [39], Erika Bridell[40], Martyn Brotherson[41], Leslie Robert Brown [42], Hannah L. Buckley [43], Erika Buscardo [44], Nokuphila Buthelezi[45], Luciano Cagnolo [22], Alice Calvente [46], Giovanni Capobianco[47], Laura Carreón-Palau [48], Suzanne Carriere [49], Bradley S. Case [43], Jenyu Chang [50], Juan Matías Chaparro[51], Chi-Ling Chen[50], Christine Chicoine[52,53], Madeleine Christensson[54], Francisco Collado Rosique [55], William Colom Montero[40], Ricardo do Sacramento da Fonseca[56], Luís P. Da Silva [29,30], Anamaria Dal Molin [57], Tad Dallas[58], Maria Carla de Francesco[59], Jorge Arturo Del Ángel-Rodríguez [60], Ricardo Díaz-Delgado [61], Thomas Dirnböck [62], Ika Djukic [62], Philile Dladla[45], Jeremías Domínguez Masciale [22], Thiago Dorigo [32,63], Errol Douwes [45,64], Torbjørn Ekrem [65], Helena Enderskog[66], Charlotta Erefur[67], Muhammad Fahad[68], Mohsen Falahati-Anbaran[65], Arielle Farrell[2], Gabriel Ferland[69], Emanuele Ferrari [70], Axa Figueiredo [71], Fernando Forero[72], Inga Freiberga [73], Andrea Frosch-Radivo[74], Luis Alberto Ganchozo Intriago[23], Laura Garzoli [75], Paola Giacomotti[75], Andros T. Gianuca[76], Olivier Gilg [35,36], Vladimir Gilg[36], Fanney Gísladóttir[77], Ryan Glowacki[41], Brigitte Gottsberger [74], Jocelyn Gregoire[78], Elli Groner[79,80], Patrícia Guedes [29,30], Aimee Michelle Guile[81], Peter Haase [82,83], Fazal Hadi[84], Magdalena Haidegger[85], Leivur Janus Hansen [15], Lars Holst Hansen[86], Reid Harrop [87], Harald Havnås[88], David Herrera Báez[55], Chris C. Y. Ho[89], Denise Hohenbühel[74], Marketa Houska Tahadlova[73,90], Jari Hänninen [91], Linda Höglund[54], Kolbrún Í Haraldsstovu[15], Elise Imbeau[69], Jasmin Inkinen[91], Masae Iwamoto Ishihara [92], Abigail C. Jackson[93,94], Gunnar Jansson[54], Rohit Jha [95], Gerald Kager[96], Rhea Kahale [38], Oula Kalttopää[25], Elizabeth Wanjiru Karai[97], Dave Karlsson [88], Andrea Kaus-Thiel[98], Asghar Khan [99], Qaisar Khan [100], Keishi Kimoto [92,101], Shadrack Chumo Kipngetich[97], Clemens Klante [102,103,104], Leif Klemedtsson[105], Mårten Klinth [88], Janne Koskinen[106], Matti Kotakorpi[107], Agnes-Katharina Kreiling [15], Irmgard Krisai-Greilhuber [74], Erik Kristensen[108], Sebastian König [109,110], Silke Langenheder[40], Kalevi Laurila[25], Pascaline Le Gouar [33], Nicolas Lecomte [52,111,112], Erin Lecomte [52,112], Paula Moraes Leitman [32,113,114], Jorge L. León-Cortés[115], Daijiang Li [95], John Loehr [107], Carlos Lopez-Vaamonde [39,116], Mehsen Makari[38], Gabriela Giselle Mangini [34], Michael Maroschek [109,110], Vanessa A. Mata[29,30], Shunsuke Matsuoka [92], Thais Mazzafera[117], Paul G. McDonald [118], Laura Meinert[81], Mayra Meléndez-González[93,94], Angela M. Mendoza-Henao [10], Sebastien Moreau [39], Jérôme Moreau[36,119,120], Jesper Mosbacher[121], Esteban Moyer[39], Anna Mrazova [73,90], Samantha Mteshane[122], Nancy Wangari Mungai [97], Gema Muñoz Herraiz[123], Andrea Murillo-Vázquez[115], Simona Musazzi [75], Marko Mutanen [124], Jörg Müller [125,126,127], Rebeca Navarro Canales[55], Monica Ndlovu[45], Annegret Nicolai[33,128], Armin Niessner [129], Jenni Nordén [130], Paweł Nowak[131], Erin O'Connell [11], Arianna Orru[75], Thomas Pagnon [35,36], Yurani Nayive Pantoja-Diaz[10,132], Mikko Pentinsaari [133,134], Sebastian Pilloni[85], Adrian Pinder[28], Thiago A. Pinheiro [135,136], Sergei Põlme[17], Luke L. Powell [29,30,137], Gisela Pröll[62], Paola Pulido-Santacruz [138,139], Enrique Queralt[140], Mark Tristan Quilantang [89], Kirsty Quinlan[141], Ricardo Ramirez[142], Juha Rankinen[102,103], Micaela Del Valle Rasino[59], Rui Rebelo [14], Wolfram Remmers [143], Franziska Retz[125], Evelin Reyes[95], Gonzalo Rivas Torres [23], Hanna M. K. Rogers[2], Inês T. Rosário [14], Sidney Rosário Da Rosàrio da Costa[144], Tobias Rütting[105], Johannes Sahlstén[91], Carole Saliba [38], Teppo Salmirinne[145], Katerina Sam[73,90], Douglas Santos [44], Margarida Santos-Reis[14], Michel Sawan[146], Benjamin Schattanek-Wiesmair [24], Pauliina Schiestl-Aalto[147], Niels Martin Schmidt [86,148], Sebastian Seibold [109,110,149], Rupert Seidl [109,110], Linda Seifert[127], Malibongwe Sithole[42], Elise Sivault [73,90], Jessica Smart[150], Ireneusz Smerczyński[131], Ayaka Soda [151], Renata S. Sousa-Lima [135], Angela Stanisci [59], Margaret C. Stanley[152], Daleen Steenkamp[42], Elisa Stengel[125], Stefan Stoll [83,143], Willem Maartin Strauss [42], Elisabeth Stur [65], Maija Sujala [25], Janne Sundell [107], Jónína Svavarsdóttir[77], Leho Tedersoo [17,153], Saana Tepsa[106], Maor Tiko Tikochinsky[79], Esa-Pekka Tuominen[107], Stefanie Tweraser[154], Catalina Ulloa Espinosa [23,155], Joni Uusitalo[107], Mikko Vallinmäki[124], Fabrice Vannier[39], Abigail Varela[156], Emma Vatka [124], Silja Veikkolainen[25], Karl Vernes [118], Phillip C. Watts [18], Per Weslien [105], Ciara Wirth[23], Jana Helga Wisniewski [87], Amanda B. Young[93,94], Robyn Övergaard[40] & Otso Ovaskainen [1,18] ✉

[1]Organismal and Evolutionary Biology Research Programme, Faculty of Biological and Environmental Sciences, University of Helsinki, Helsinki, Finland. [2]Department of Ecology, Swedish University of Agricultural Sciences (SLU), Uppsala, Sweden. [3]Ecosystems and Environment Research Programme, Faculty of Biological and Environmental Sciences, University of Helsinki, Helsinki, Finland. [4]Madagascar Biodiversity Center, Antananarivo, Madagascar. [5]California Academy of Sciences, San Francisco, CA, USA. [6]Faculty of Agriculture/Environment/Chemistry, University of Applied Sciences HTW Dresden, Dresden, Germany. [7]Finnish Museum of Natural History, University of Helsinki, Helsinki, Finland. [8]Department of Geosciences and Geography, University of Helsinki, Helsinki, Finland. [9]School of Natural Resources and Environment, Department of Wildlife Ecology and Conservation & Florida Museum of Natural History, University of Florida, Gainesville, FL, USA. [10]Colecciones Biológicas, Instituto de Investigación de Recursos Biológicos Alexander von Humboldt, Villa de Leyva, Colombia. [11]Tyson Research Center, Washington University, St. Louis, St. Louis, MO, USA. [12]Department of Zoology, Abdul Wali Khan University Mardan, Mardan, Pakistan. [13]Companhia das Lezírias S.A., Samora Correia, Portugal. [14]Centre for Ecology, Evolution and Environmental Changes & CHANGE—Global Change and Sustainability Institute, Faculty of Sciences, University of Lisbon, Lisbon, Portugal. [15]Faroe Islands National Museum (Tjóðsavnið), Hoyvík, Faroe Islands. [16]Faculty of Science and Engineering, Southern Cross University, Lismore, New South Wales, Australia. [17]Institute of Ecology and Earth Sciences, University of Tartu, Tartu, Estonia. [18]Department of Biological and Environmental Science, University of Jyväskylä, Jyväskylä, Finland. [19]Royal Botanic Gardens, Kew, London, UK. [20]Gothenburg Global Biodiversity Centre, Department of Biological and Environmental Sciences, University of Gothenburg, Göteborg, Sweden. [21]Department of Biology, University of Oxford, Oxford, UK. [22]Instituto Multidisciplinario de Biología Vegetal, Universidad Nacional de Córdoba, Consejo Nacional de Investigaciones Científicas y Técnicas, Córdoba, Argentina. [23]Tiputini Biodiversity Station, Universidad San Francisco de Quito, Tiputini, Ecuador. [24]Department of Natural History Collections and Research Centre (SFZ), Tiroler Landesmuseen-Betriebsgesellschaft m.b.H., Hall in Tirol, Austria. [25]Kilpisjärvi Biological Station, University of Helsinki, Helsinki, Finland. [26]Cavanilles Institute of Biodiversity and Evolutionary Biology, University of Valencia, Valencia, Spain. [27]CRIMAC, Department of Integrative Marine Ecology, Stazione Zoologica Anton Dohrn, Amendolara, Italy. [28]Western Australian Department of Biodiversity, Conservation and Attractions, Perth, Western Australia, Australia. [29]CIBIO, Centro de Investigação em Biodiversidade e Recursos Genéticos, InBIO Laboratório Associado, Campus de Vairão, Universidade do Porto, Vairão, Portugal. [30]BIOPOLIS Program in Genomics, Biodiversity and Land Planning, CIBIO, Vairão, Portugal. [31]CIBIO, Centro de Investigação em Biodiversidade e Recursos Genéticos, InBIO Laboratório Associado, Instituto de Agronomia, Universidade de Lisboa, Lisboa, Portugal. [32]Fundação Antonelli Brasil, Nova Friburgo, Brazil. [33]ECOBIO-UMR 6553, CNRS, University of Rennes, Rennes, France. [34]Instituto de Ecología Regional, CONICET-UNT, Yerba Buena, Argentina. [35]Chrono-environnement, UMR 6249, Université Marie et Louis Pasteur, CNRS, Besançon, France. [36]Groupe de Recherche en Ecologie Arctique, Francheville, France. [37]European Forest Institute (EFI)—Barcelona Office Sant Pau, Barcelona, Spain. [38]Laboratoire Biodiversité et Génomique Fonctionnelle, Faculté des Sciences, Université Saint-Joseph, Campus Sciences et Technologies, Beirut, Lebanon. [39]Institut de Recherche sur la Biologie de l'Insecte, UMR 7261, Université de Tours, CNRS, Tours, France. [40]Department of Ecology and Genetics/Erken Laboratory, Uppsala University, Uppsala, Sweden. [41]Botanic Gardens and Parks Authority, Western Australian Department of Biodiversity, Conservation and Attractions, Perth, Western Australia, Australia. [42]Applied Behavioural Ecology and Ecosystem Research Unit, Department of Environmental Science, University of South Africa, Pretoria, South Africa. [43]School of Science, Auckland University of Technology, Auckland, New Zealand. [44]Department of Animal Biology, Institute of Biology, University of Campinas, Campinas, Brazil. [45]Biodiversity Management Department, eThekwini Municipality, Durban, South Africa. [46]Department of Botany and Zoology, Federal University of Rio Grande do Norte, Natal, Brazil. [47]ARDEA—Association for Research, Dissemination and Environmental Education, Napoli, Italy. [48]Centro de Investigaciones Biológicas del Noroeste, S. C. (CIBNOR), La Paz, Mexico. [49]Environment and Natural Resources (now Environment and Climate Change), Government of Northwest Territories, Yellowknife, Northwest Territories, Canada. [50]Agricultural Chemistry Division, Taiwan Agricultural Research Institute, Taichung City, Taiwan. [51]National Institute for Aerospace Technology (INTA), Madrid, Spain. [52]Université de Moncton, Moncton, New Brunswick, Canada. [53]Canada Research Chair in Polar and Boreal Ecology & Centre d'Études Nordiques, Moncton, New Brunswick, Canada. [54]Grimsö Wildlife Research Station, Department of Ecology, Swedish University of Agricultural Sciences, Riddarhyttan, Sweden. [55]Servici Devesa-Albufera (Ayuntamiento de Valencia), València, Spain. [56]Monte Pico Association, Monte Café, São Tomé and Príncipe. [57]Department of Microbiology and Parasitology, Biosciences Center, Federal University of Rio Grande do Norte, Natal, Brazil. [58]Department of Biological Sciences, University of South Carolina, Columbia, SC, USA. [59]Department of Bioscience and Territory, University of Molise, Termoli, Italy. [60]Colectivo de Académicos Sudcalifornianos A.C., La Paz, Mexico. [61]ICTS-RBD, Estación Biológica de Doñana-CSIC, Seville, Spain. [62]Environment Agency Austria, Vienna, Austria. [63]Departamento de Ecologia, Instituto de Biologia Roberto Alcantara Gomes, Universidade do Estado do Rio de Janeiro, Rio de Janeiro, Brazil. [64]School of Life Sciences, University of KwaZulu-Natal, Westville Campus, Durban, South Africa. [65]Department of Natural History, NTNU University Museum, Norwegian University of Science and Technology, Trondheim, Norway. [66]Uppsala University, Uppsala, Sweden. [67]Svartberget Research Station, Unit for Field-based Forest Research, Swedish University of Agicultural Sciences, Uppsala, Sweden. [68]International School and College of Cordoba, Batkhela Campus, Malakand, Pakistan. [69]Guelph University, Guelph, Ontario, Canada. [70]CNR—Istituto di Ricerca sulle Acque, Brugherio, Italy. [71]Department of Forestry Engineering, University of Brasília, Brasília, Brazil. [72]Reserva Natural y Cascadas Los Tucanes, Gachantivá, Colombia. [73]Biology Centre of the Czech Academy of Sciences, Institute of Entomology, Ceske Budejovice, Czech Republic. [74]Department of Botany and Biodiversity Research, University of Vienna, Vienna, Austria. [75]Water Research Institute (CNR-IRSA), National Research Council, Verbania Pallanza, Italy. [76]Department of Ecology, Federal University of Rio Grande do Norte, Natal, Brazil. [77]Agricultural University of Iceland, Hvanneyri, Iceland. [78]Environment and Climate Change Canada, Yellowknife, Northwest Territories, Canada. [79]Dead Sea and Arava Science Center (Ramon Branch), Mitzpe Ramon, Israel. [80]Ben Gurion University (Eilat Campus), Beer Sheva, Israel. [81]Wek'èezhìi Renewable Resources Board, Yellowknife, Northwest Territories, Canada. [82]Department of River Ecology and Conservation, Senckenberg Research Institute and Natural History Museum Frankfurt, Frankfurt, Germany. [83]Faculty of Biology, University of Duisburg-Essen, Essen, Germany. [84]Department of Biotechnology, University of Malakand, Malakand, Pakistan. [85]Naturpark Karwendel, Hall in Tirol, Austria. [86]Department of Ecoscience, Aarhus University, Roskilde, Denmark. [87]Collections Team, Centre for Biodiversity Genomics, Guelph, Ontario, Canada. [88]Station Linne, Färjestaden, Sweden. [89]Taxonomy Team, Centre for Biodiversity Genomics, Barcode of Life Data System, Guelph, Ontario, Canada. [90]Faculty of Science, University of South Bohemia, Ceske Budejovice, Czech Republic. [91]Archipelago Research Institute, University of Turku, Turku, Finland. [92]Ashiu Forest Research Station, Field Science Education and Research Center, Kyoto University, Kyoto, Japan. [93]Toolik Field Station, Institute of Arctic Biology, University of Alaska Fairbanks, Fairbanks, AK, USA. [94]Institute of Arctic Biology, University of Alaska Fairbanks, Fairbanks, AK, USA. [95]Department of Biological Sciences, Louisiana State University, Baton Rouge, LA, USA. [96]Bereich Naturschutz & Geoinformationstechnik, Stadt Wien, Wien, Austria. [97]Egerton University, Njoro, Kenya. [98]Nationalparkamt Hunsrück-Hochwald, Birkenfeld, Germany. [99]Department of Botany, Government Degree College Totakan, Malakand, Pakistan. [100]Department of Chemistry, University of Malakand, Malakand, Pakistan. [101]Hokkaido Forest Research Station, Field Science and Education Research Center, Kyoto University, Kyoto, Japan. [102]SITES (Swedish Infrastructure for Ecosystem Sciences), Uppsala, Sweden. [103]Sweden Water Research, Lund, Sweden. [104]Lund University, Lund, Sweden. [105]Department of Earth Sciences, University of Gothenburg, Gothenburg, Sweden. [106]Konnevesi Research Station, University of Jyväskylä, Konnevesi, Finland. [107]Lammi Biological Station, University of

Helsinki, Helsinki, Finland. [108]Unit for Field-based Forest Research, Swedish University of Agricultural Sciences, Uppsala, Sweden. [109]Ecosystem Dynamics and Forest Management Group, School of Life Sciences, Technical University of Munich, Freising, Germany. [110]Research and Monitoring, Berchtesgaden National Park, Berchtesgaden, Germany. [111]Centre d'Études Nordiques, Quebec City, Quebec, Canada. [112]Canada Research Chair in Polar and Boreal Ecology, Université de Moncton, Moncton, New Brunswick, Canada. [113]Jardim Botânico do Rio de Janeiro, Rio de Janeiro, Brazil. [114]Fundação de Amparo à Pesquisa do Estado do Rio de Janeiro, Rio de Janeiro, Brazil. [115]El Colegio de la Frontera Sur, Departamento de Conservación de la Biodiversidad, Unidad San Cristóbal de las Casas, Chiapas, Mexico. [116]INRAE, UR633, Zoologie Forestière, Orléans, France. [117]Universidade Estadual de Campinas, UNICAMP, Campinas, Brazil. [118]School of Environmental and Rural Science, University of New England, Armidale, New South Wales, Australia. [119]Biogéosciences, Université de Bourgogne, Dijon, France. [120]Centre d'Etudes Biologiques de Chizé, UMR 7372, CNRS & La Rochelle Université, Villiers-en-bois, France. [121]Norwegian Polar Institute, Tromsø, Norway. [122]WILDTRUST, Pietermaritzburg, South Africa. [123]Agricultores de la Vega SA, Valencia, Spain. [124]Ecology and Genetics Research Unit, University of Oulu, Oulu, Finland. [125]Field Station Fabrikschleichach, Department of Animal Ecology and Tropical Biology (Zoology III), Julius Maximilians University Würzburg, Rauhenebrach, Germany. [126]University of Würzburg, Würzburg, Germany. [127]Bavarian Forest National Park, Grafenau, Germany. [128]Living Lab CLEF, Plélan-le-Grand, France. [129]Panguana Foundation, Munich, Germany. [130]Norwegian Institute for Nature Research (NINA), Trondheim, Norway. [131]Białowieża Geobotanical Station, Faculty of Biology, University of Warsaw, Warsaw, Poland. [132]Programa de Ecología, Fundación Universitaria de Popayán, Popayán Cauca, Colombia. [133]Canadian National Collection of Insects (Arachnids and Nematodes), Ottawa, Ontario, Canada. [134]Centre for Biodiversity Genomics, University of Guelph, Guelph, Ontario, Canada. [135]Laboratory of Bioacoustics and EcoAcoustic Research Hub, Biosciences Center, Federal University of Rio Grande do Norte, Natal, Brazil. [136]Ecology Graduate Program, Biosciences Center, Federal University of Rio Grande do Norte, Natal, Brazil. [137]Biodiversity Initiative, Houghton, MI, USA. [138]Department of Biology, Faculty of Natural Sciences, Universidad del Rosario, Bogotá, Colombia. [139]Instituto de Investigación de Recursos Biológicos Alexander von Humboldt, Bogotá, Colombia. [140]Instituto de Biologia, Universidade Estadual de Campinas, Campinas, Brasil. [141]Biodiversity and Conservation Science, Western Australian Department of Biodiversity, Conservation and Attractions, Perth, Western Australia, Australia. [142]Tagis—Centro de Conservação das Borboletas de Portugal, Lisboa, Portugal. [143]University of Applied Sciences Trier, Environmental Campus Birkenfeld, Birkenfeld, Germany. [144]CIAT—Centro de Investigação Agronómica e Tecnologia, São Tomé, South Africa. [145]Oulanka Research Station, Kuusamo, Finland. [146]Lebanese Association for Migratory Birds, Zgharta, Lebanon. [147]Institute for Atmospheric and Earth System Research (INAR)/Physics, University of Helsinki, Helsinki, Finland. [148]Arctic Research Centre, Aarhus University, Roskilde, Denmark. [149]Forest Zoology, TUD Dresden University of Technology, Tharandt, Germany. [150]North Slave Métis Alliance, Yellowknife, Northwest Territories, Canada. [151]Department of Zoology, Graduate School of Science, Kyoto University, Kyoto, Japan. [152]Centre for Biodiversity and Biosecurity, School of Biological Sciences, University of Auckland, Auckland, New Zealand. [153]Department of Zoology, College of Science, King Saud University, Riyadh, Saudi Arabia. [154]Kalkalpen National Park, Molln, Austria. [155]Ecology and Data Science, University College, London, London, UK. [156]BirdLife International, Água Grande, South Africa. ✉e-mail: otso.t.ovaskainen@jyu.fi

# Reporting Summary

## Statistics

For all statistical analyses, confirm that the following items are present in the figure legend, table legend, main text, or Methods section.

| n/a | Confirmed | |
|---|---|---|
| ☐ | ☒ | The exact sample size (*n*) for each experimental group/condition, given as a discrete number and unit of measurement |
| ☐ | ☒ | A statement on whether measurements were taken from distinct samples or whether the same sample was measured repeatedly |
| ☐ | ☒ | The statistical test(s) used AND whether they are one- or two-sided<br>*Only common tests should be described solely by name; describe more complex techniques in the Methods section.* |
| ☐ | ☒ | A description of all covariates tested |
| ☐ | ☒ | A description of any assumptions or corrections, such as tests of normality and adjustment for multiple comparisons |
| ☐ | ☒ | A full description of the statistical parameters including central tendency (e.g. means) or other basic estimates (e.g. regression coefficient) AND variation (e.g. standard deviation) or associated estimates of uncertainty (e.g. confidence intervals) |
| ☐ | ☒ | For null hypothesis testing, the test statistic (e.g. $F$, $t$, $r$) with confidence intervals, effect sizes, degrees of freedom and $P$ value noted<br>*Give P values as exact values whenever suitable.* |
| ☒ | ☐ | For Bayesian analysis, information on the choice of priors and Markov chain Monte Carlo settings |
| ☐ | ☒ | For hierarchical and complex designs, identification of the appropriate level for tests and full reporting of outcomes |
| ☐ | ☒ | Estimates of effect sizes (e.g. Cohen's *d*, Pearson's *r*), indicating how they were calculated |

*Our web collection on statistics for biologists contains articles on many of the points above.*

## Software and code

Policy information about availability of computer code

| | |
|---|---|
| Data collection | Custom AudioMoth firmware installed on the audio recorders used to record audio: https://github.com/OpenAcousticDevices/AudioMoth-LIFEPLAN/releases/download/0.1.5/AudioMoth-LIFEPLAN-0.1.5.bin<br>Full audio recording protocol, including the two AudioMoth configurations used to record audio: https://dx.doi.org/10.17504/protocols.io.kqdg3xbp1g25/v2 |
| Data analysis | The scripts and code to reproduce the results of this manuscript are found in GitHub repository https://github.com/psomervuo/soundscape.<br>Bird species identifications: BirdNet Analyzer version 2.4. (Kahl, S., Wood, C. M., Eibl, M. & Klinck, H. BirdNET: A deep learning solution for avian diversity monitoring. Ecol Inform 61, 101236 (2021))<br>Spectrogram computing: Python library librosa version 0.9.2.33<br>Acoustic indices computing: Python package Acoustic_Indices v1.0.0 |

For manuscripts utilizing custom algorithms or software that are central to the research but not yet described in published literature, software must be made available to editors and reviewers. We strongly encourage code deposition in a community repository (e.g. GitHub). See the Nature Portfolio guidelines for submitting code & software for further information.

## Data

Policy information about availability of data

All manuscripts must include a data availability statement. This statement should provide the following information, where applicable:

- Accession codes, unique identifiers, or web links for publicly available datasets
- A description of any restrictions on data availability
- For clinical datasets or third party data, please ensure that the statement adheres to our policy

The data have been deposited to Zenodo https://doi.org/10.5281/zenodo.11516373. The repository will be opened upon acceptance of the manuscript. A link for reviewers is included in the manuscript text.

## Research involving human participants, their data, or biological material

Policy information about studies with human participants or human data. See also policy information about sex, gender (identity/presentation), and sexual orientation and race, ethnicity and racism.

| | |
|---|---|
| Reporting on sex and gender | This information has not been collected. |
| Reporting on race, ethnicity, or other socially relevant groupings | This information has not been collected. |
| Population characteristics | Research does not involve humans or human data. |
| Recruitment | Research does not involve humans or human data. |
| Ethics oversight | Research does not involve humans or human data. |

Note that full information on the approval of the study protocol must also be provided in the manuscript.

# Field-specific reporting

Please select the one below that is the best fit for your research. If you are not sure, read the appropriate sections before making your selection.

☐ Life sciences  ☐ Behavioural & social sciences  ☒ Ecological, evolutionary & environmental sciences

For a reference copy of the document with all sections, see nature.com/documents/nr-reporting-summary-flat.pdf

# Ecological, evolutionary & environmental sciences study design

All studies must disclose on these points even when the disclosure is negative.

| | |
|---|---|
| Study description | At each of 139 sites around the world, we deployed 3-5 passive audio recorders in a nested design. Some sites in the Nordic countries and Madagascar were sampled continuously, while others switched annually between a natural and urban type location as proposed by local researchers. In this study, we used a randomly selected one-minute clip from each hour recorded. We used a total of 1,484,181 clips. For these clips, we calculated a set of established acoustic indices and acoustic event classes, and identified bird species vocalising. We analysed periodic rhythms in different soundscape indices, evaluated the predictability of local patterns from global ones as well as anthropogenic and climatic impacts, and effects of natural/urban site type on acoustic index values and species richness. Explanatory variables used included day of year, time of day, latitude, human footprint index, elevation, annual mean temperature, annual precipitation and habitat type. Full study description: https://doi.org/10.1371/journal.pone.0313353 |
| Research sample | In the statistical analyses we use as the sampling unit one minute of sound. The spatiotemporal distrubution of these sampling units is described in the study description. |
| Sampling strategy | At each site, data were collected via passive acoustic monitoring (PAM) using AudioMoth v1.1 devices63. For any one time, there were up to five AudioMoth devices operated per site within a 1-ha area62. The total number of recordings varied between the sites due to equipment malfunctioning and logistic constraints, such as site accessibility due to road damage caused by hurricanes and storms. At some sites, the sampling period included only a few months, whereas in other sites the sampling was continuous throughout the year. The data were subsampled to the level of full hours. To allow the sampling of equivalent time periods, we ensured synchronisation among individual recorders. To this aim, all AudioMoth devices were synchronized weekly with Coordinated Universal Time. From each hour recorded, a one-minute-long clip was randomly selected. Consequently, the data to be analysed consisted of up to 24 recordings per day per site. The total number of the one-minute recordings used in the present analysis was 1,484,181. |
| Data collection | Sampling teams around the world placed 3-5 pre-programmed passive audio recorders at their sites. They visited them weekly or biweekly to change batteries and microSD cards. They collected metadata by scanning QR codes on the recorders and microSD cards, recording the time and location of each recorder placement and collection. |

| | |
|---|---|
| Timing and spatial scale | Audio recording began at the first site 2020-12-21 and is still ongoing. The data included in this study range from 2021-02-08 to 2024-04-18. Audio was recorded for one minute every ten minutes on site corners, and continuously for 48 hours at the start of each week and then one minute every ten minutes in the middle of the site. The recording regime was designed so that one visit per week would be enough to keep batteries running and memory cards from filling up. Our spatial scale is global, with 139 sites on six continents and a latitudinal gradient of 116 degrees. A full list of sites is in supplementary table 3. There are gaps in the recording schedule due to equipment failure and replacement delays. |
| Data exclusions | We excluded only sites that had too little sampling for statistical analysis. From each hour recorded, a one-minute-long clip was randomly selected. |
| Reproducibility | To verify species identifications, an ornithologist listened to a random sample of recordings to check the species identifications. Each site had 3-5 audio recorders whose results can be compared. |
| Randomization | We randomised sites as to starting at the Natural or Urban location. We randomised the choice of a one-minute clip from each hour of recording. |
| Blinding | Blinding was not applied because the data were analysed with automated scripts without subjective human judgment. Audio files were encrypted and not listened to or filtered by data collectors. |

Did the study involve field work?    ☒ Yes    ☐ No

## Field work, collection and transport

| | |
|---|---|
| Field conditions | Field work was carried out over several years at 139 locations on six continents around the world, and field conditions varied considerably. |
| Location | Full list of 139 study sites is in supplementary table S3. |
| Access & import/export | Data was collected and is owned by participating teams. Teams were instructed to follow their local legislation concerning audio recording and to post signs warning that audio is being recorded. |
| Disturbance | Audio recorders were attached with canvas straps to trees where available. Where trees were not available, teams installed posts in the ground. The recorders were visited once a week. |

# Reporting for specific materials, systems and methods

We require information from authors about some types of materials, experimental systems and methods used in many studies. Here, indicate whether each material, system or method listed is relevant to your study. If you are not sure if a list item applies to your research, read the appropriate section before selecting a response.

## Materials & experimental systems

| n/a | Involved in the study |
|---|---|
| ☒ | ☐ Antibodies |
| ☒ | ☐ Eukaryotic cell lines |
| ☒ | ☐ Palaeontology and archaeology |
| ☐ | ☒ Animals and other organisms |
| ☒ | ☐ Clinical data |
| ☒ | ☐ Dual use research of concern |
| ☒ | ☐ Plants |

## Methods

| n/a | Involved in the study |
|---|---|
| ☒ | ☐ ChIP-seq |
| ☒ | ☐ Flow cytometry |
| ☒ | ☐ MRI-based neuroimaging |

## Animals and other research organisms

Policy information about studies involving animals; ARRIVE guidelines recommended for reporting animal research, and Sex and Gender in Research

| | |
|---|---|
| Laboratory animals | Not used |
| Wild animals | Wild animal sounds were passively recorded. No animals were captured or manipulated. Animals identified from the recordings were birds. |
| Reporting on sex | Sex was not considered or reported on. |
| Field-collected samples | No physical samples were collected. |

| Ethics oversight | No ethical approval or guidance was required, as no animals were handled in the study. |

Note that full information on the approval of the study protocol must also be provided in the manuscript.

## Plants

| Seed stocks | Not used. |

| Novel plant genotypes | Not used. |

| Authentication | Not used. |

