## [Peer Review File · Nature Ecology & Evolution]

Less predictable global rhythms in human than wildlife contributions to soundscapes

Corresponding Author: Professor Otso Ovaskainen

Version 0:

Decision Letter:

27th August 2024

Dear Professor Ovaskainen,

Your Article, "More predictable rhythms in the natural than anthropogenic global soundscape" has now been seen by 2 reviewers. Please accept my apologies for the delay in our decision -- we waited a few extra days but were unable to obtain comments from our third reviewer. You will see from the reviewers' comments copied below, that while they find your work of considerable potential interest, they have raised substantial concerns. In light of these comments, we cannot accept the manuscript for publication, but would be very interested in considering a revised version that addresses these serious concerns.

In particular, we consider it crucial that you include additional analyses and re-write the manuscript for better clarity, as suggested by Reviewer #1. Please also report quantitative findings where possible, include relevant background literature, and explain the rationale for the study and relevance of your findings in greater detail.

If you do choose to revise your manuscript taking into account *all* reviewer and editor comments, please bear in mind that we will be reluctant to approach the reviewers again in the absence of major revisions. We hope you will find the reviewers' comments useful as you decide how to proceed.

* Include a "Response to reviewers" document detailing, point-by-point, how you addressed each referee comment. If no action was taken to address a point, you must provide a compelling argument. This response will be sent back to the referees along with the revised manuscript.

* If you have not done so already we suggest that you begin to revise your manuscript so that it conforms to our Article format instructions at <http://www.nature.com/natecolevol/info/final-submission>. Refer also to any guidelines provided in this letter.

Link Redacted

If you wish to submit a suitably revised manuscript we would hope to receive it within 6 months. If you cannot send it within this time, please let us know. We will be happy to consider your revision so long as nothing similar has been accepted for publication at Nature Ecology & Evolution or published elsewhere.

Nature Ecology & Evolution is committed to improving transparency in authorship. As part of our efforts in this direction, we are now requesting that all authors identified as 'corresponding author' on published papers create and link their Open Researcher and Contributor Identifier (ORCID) with their account on the Manuscript Tracking System (MTS), prior to acceptance. This applies to primary research papers only. ORCID helps the scientific community achieve unambiguous attribution of all scholarly contributions. You can create and link your ORCID from the home page of the MTS by clicking on 'Modify my Springer Nature account'. For more information please visit www.springernature.com/orcid.

Thank you for the opportunity to review your work.

[redacted]

Reviewer expertise:

Reviewer #1: (landscape & soundscape ecology, ecological modelling)

Reviewer #2: (ecoacoustics methods)

Reviewers' comments:

Reviewer #1 (Remarks to the Author):

I also attach a Word document version that has easier to follow formatting than this plain text version below:

The authors of the paper "More predictable rhythms in the natural than anthropogenic global soundscape" conduct an analysis of over 1.4 million audio recordings of the soundscape from 139 sites located on six continents. Recordings for each site were in a pairwise design (urban and peri-urban settings). One-minute recordings were analyzed in the context of the proposed "rhythms of nature" conceptual framework of Pijanowski et al. (2011) that describe the predictable diel and seasonal patterns of biophonies that should exist around the world. The authors use a set of standard acoustic indices (ADI, ACI, etc.), measures of sound sources that are extracted from an acoustic event detector and also apply the widely used CNN-based BirdNET tool to detect bird species. They find that for the most part, the rhythms of nature are indeed predictable in key time periods and across latitudinal gradients. Another key finding is that the human noise footprint is interfering with these natural rhythms which are key functioning features of all ecosystems. They conduct a set of spatial-temporal analyses of acoustic indices, sound source event metrics, as well as the species' habitat features. I do like the topic of the paper and I think many in ecology not familiar with the concept of the soundscape will be as well. However, there are numerous shortcomings of the paper. I spent three days reviewing this hoping to provide some insights for significant revisions and so this represents a major time commitment to make sure that if this moves toward publication, a lot of the errors and poor writing can be corrected. I hope the editors and authors find these to be helpful.

First, I found the paper not to be well written; in fact, compared to most Nature/Science/Ecology Letter/PNAS papers I read, this paper lacks the high level of density, clarity, robustness and precision of the prose I would expect from these articles. One of my best illustrations for this is the sentence in line 465 "No matter where we are and when we listen, there is sound". This is blatantly obvious to all of us in this field and does not address any of the research questions. Another is the opening of the Discussion "...how the planet is buzzing with sound". These rather "folksy comments" provide no factual content and are not very scientifically useful (actually, even a bit misleading since this paper is an analysis of the patterns of the presence of sound in the environment and not how people perceive it). Overall, the language and interpretation throughout the paper is "loose". There is what I would call a lot of hand waving without getting to robust interpretations of the analysis. The core paper is easy to follow although the supplement got difficult to navigate at times (a supplement to a supplement was confusing).

There are MANY terms (spectral, amplitude) not used very precisely either. I would highly suggest consulting the glossary of terms that are in Pijanowski and Rivas (2023; Encyclopedia of Biodiversity, Elsevier, Soundscapes and Vibrosapes, Chapter 134) for the most commonly used terms in soundscape ecology and how they should be used.

Let me state up front that this is a paper worthy of eventually being published in Nature Ecology and Evolution. The two main "take aways", that the soundscape is a predictable feature of ecosystems and that human noise is potentially affecting these rhythms is an important message to make to large communities of scholars and natural resource managers. I will for one cite it a lot once it has gone through some significant revisions. In my opinion, there is a lot more work that is needed to get the paper to a publishable form. There are areas that are quite vague and specifics of analyses and calculations are simply not described. A large body of literature (on urban soundscapes such as Lex Brown, Jian Kang and Östen Axelsson and biodiversity assessments a la conservation biology work of say, Zuzana Burivalova) are missing and the take home points are not made all that strongly. In general, there are large bodies of literature missing from the paper that would make this a very informative paper across basic and applied ecology and evolution.

The rest of my comments, which address the more general ones listed above, are summarized here:

Manuscript (i.e., Main)

Lines 329-337. It should be made clear here if the authors truly mean anthropophony or technophony as their major sound source category. The latter was proposed by Gage after the Pijanowski et al., (2011) Bioscience paper to describe sounds

sources that do not include human voices. The list of examples seems to suggest technophony but the analysis includes human voices (some of us consider human voices to be part of the biophony since we are indeed animals). If we consider human voices, to what extent do these contribute toward the disruption of the natural rhythms (perhaps some of their multivariate analyses might point to that but I suspect that this is minor). My opinion is that the authors should focus on technophony since that is the “noise” component that is being assessed with most of the analyses and aligns with the paper’s objectives.

Lines 338-347. Paragraph could be stronger and include more specifics. It is known, for example, that birds and amphibians dominate the dawn chorus and that the dusk chorus is dominated by insects. It should also be mentioned that the timing of, and the length of, both the dawn and dusk chorus differs by latitude and time of the year.

Lines 348-355. I find this paragraph to be far too general and some of the noise patterns are indeed diel. First, the sounds of highways are likely to be most loud prior to the start of the business/school day and then again most loud during the end of the business/school day. So, I’m not sure I agree with the authors. Second, each ecosystem has a predictable soundscape phenology that is related to the phenologies of major sonically active animals (here in the midlatitudes, amphibians emerge early spring, then come breeding and migratory birds, and then the insects). These major animal sonic groupings also have different acoustic patterns (pulsating, melodic). There is a lot of literature to support these kinds of patterns and they should be clearly described here. This is also the paragraph where a lot of the urban soundscapes research should be cited (Brown, Kang, Axelsson). See Chapter 13 in Pijanowski’s book (2024) for a thorough review of the work in urban soundscapes.

Lines 356-363. There is a lot of work done on the US NPS soundscapes as a natural resource (see Dumyah and Pijanowski, *Landscape Ecology* 2011, *Managing soundscapes as common pool resources*) and also the work of Kurt Fistrup and Peter Newman at Penn State University). I’m not certain this is a useful paragraph as written.

Lines 364-378. This paragraph does not precisely use the terms properly and so I was a bit confused with what they were referring to by “metrics”, “features”, “descriptors”, and “representations” which are referenced in citation #13. I could not find those specific terms there. Figure 1 is referenced and that figure (which needs A LOT of work) does not provide the reader with what these terms refer to. The authors need to clarify the use of these and perhaps provide a diagram that illustrates how these differ, especially as they relate to acoustic indices. I am also confused by the use of the term “voice abundance”. Some researchers have used the term call rates. Also, the field has not settled on how to use the number of bird species calls in any analysis. What you are specifically doing here, I think, is to use BirdNET to create a species richness value per recording and then per site.

Lines 379-383. I am almost certain I know what you mean here in your statement of the paper objectives. But these are clumsy at best. Could I suggest: “how pronounced and predictable are global biophonies across diel and seasonal patterns?” and “does noise produced by humans affect these natural rhythms?” that is then followed by “We test examine each of these questions using a global soundscape database and a paired urban-rural experimental design”.

Line 373. I’d exclude human voices from the list of anthropophony (by the way, anthropophony and anthrophony have been determined by a scholar in Latin-Greek to be synonymous).

Lines 386-394. This is the second use of the term circadian rhythms. Authors have used diurnal through to now. Other researchers use diel which is probably the best term to use of the three. I’d stay consistent with the terminology throughout the paper. As I am a bit confused by the term “descriptors”, it was difficult for me to follow this first paragraph in Results.

Lines 395-403. This is a very difficult paragraph to embrace since there is not much detail in what the authors did. The hourly patterns I think are a challenge to compare across latitudinal gradients since the timing of dawn varies by date and latitude. Using just the local clock time is misleading. Instead, the authors should have indexed the time of the recording to be something of a standard “time since civil sunrise”. Indeed, at 60° during the summer solstice, there is no period of complete darkness. If I had one serious criticism, is that they (I think) used local time across all sites which does not reflect the dawn. A second comment is that the dawn chorus at different locations and time of the year has different lengths. My extensive time at locations of the equator have taught me that the dawn and dusk periods are very short there nearly all year round.

Lines 423. I do not think that the use of the term “silence” is proper here. Silence refers to the lack of any sound. That only occurs in vacuums or anechoic chambers. Quiet is a better term and it needs to be described as very low amplitude of sound waves that are not detectable using field equipment.

Lines 434-496. What the authors have found – that is, that nearby natural areas and not different than urban areas in terms of call rates, is not that unusual. There are MANY papers that have described how birds are now calling at higher pitches, are louder or are more active at night in response to the heightened ambient sounds from humans. Can the authors place their research in this large body of literature (e.g., see the work that describes the Lombard Effect)?

Lines 454-560. The entire discussion needs to be rewritten. First, the authors are introducing (using a prominent subheading) “what does the word sound like?”. This was not one of the key research questions and I am not sure they have answered it. The section in lines 465-475 do not address this question either. The presentation is overly general and very vague (handy waving of correlation patterns). The second subheading does not seem to be fully answered either (and this is what the authors state was one of the research questions). I’m confused by several of the sentences especially the one that contains (line 502) “traditional community data...”. How do the recordings and the analysis “pinpoint the agents that are making sounds...?”. I also want to point out that the fact that the acoustic indices do not always strongly correlate is a GOOD thing. They each are measuring a different aspect of the soundscape. ADI, for example, measures the diversity of total sound amplitude across audible frequency bands split into 1 kHz bands. ACI measures the amount of modulation in time and frequency. NDSI measures the relative spectral power of sound in the frequency bands associated with biological sounds versus those produced by technophony. The role of the soundscape ecologist should be to decipher the complexities of the soundscape using these diverse tools.

Lines 510-520. One of the major issues and discussions right now is how to address the accuracy of BirdNET output. Interestingly, here the authors use the proper assessment descriptions (quantifying false negatives especially since BirdNET seems to have high values of these based on our assessments as well). First, I would not trust any output with confidence scores below 0.50 so I am not certain why they would consider anything less as these are neural net tools that learn threshold into binary output data (presence and absence). Second, one needs to develop methods that assess false

negatives (a bird called and it was not detected at all). Third, call activity levels is driven by taxonomic considerations (nocturnal nightjars call repeatedly whereas many higher-level passerines call infrequently and are not that repetitive in their call structure). I'm not sure I would use call rates in the analysis until the research community can determine how to use BirdNET data that contains a lot of this information.

Lines 554-560. This is the weakest section of the paper. The authors need to think more deeply about the implications to their results. Why should ecologists care? I can think of dozens of reasons that rise to the very top of highly crucial issues but these are not included anywhere in this paper and certainly not here. I think addressing the work in the context of the biodiversity crisis, climate change, conservation management, urban planning and even human-nature interactions are all extremely important. Indeed, this gap is evident in the introduction and abstract. Serious thought needs to be put into this part of the paper to make it worthy of publication in *Nature Ecology and Evolution*.

Materials and Methods

Lines 565-571. Paragraph needs some editing.

Lines 572-595. A figure here that depicts the experimental designs would be helpful.

Lines 618-623. FFT models are not described. What window size and type were used to process the wav file for use in the analysis? I assume 512 and Hanning?

Lines 625-631. Many of the acoustic indices used were developed using Wildlife Acoustics SM2s, and in some cases, SM4s. Some of the acoustic indices, such as ADI and ACI, are also sensitive to parameter settings and microphone sensitivities. For example, ADI was originally designed to be used for SM2s and with preamps set to the factory settings. This means that the "noise floor" of -50dBFS was ideally configured for that model. Audiomoths use an entirely different set of microphone technologies and there are a host of preamp settings. So, I am weary of folks that use default settings on these acoustic index tools. We are also finding that ACI is sensitive to FFT window sizes (makes sense when you think about about) but also the Nyquist frequency (which might make sense) for certain FFT window sizes (large ones). **THUS**, it is extremely important to (1) make sure the acoustic indices are given values that make sense (ADI does not look right to me) and (2) that a sensitivity analysis of key parameters be undertaken to make sure these are effective parameters to use). Finally, a high pass band filter applied AFTER the recordings are created does not make sense to me. I sometimes like to use the 250 Hz HPF on the device as this makes the microphone more sensitive in the biologically active areas. Using a HPF will affect ADI and NDSI, although, if these are applied across all recordings the same, then you should be OK. In short, the use of acoustic indices is somewhat careless.

Lines 646-653. The statistical analysis section is one of the most confusing parts of the paper. I have no idea what kind of model is being developed. The description of the model's explanatory variables is not described as I would. Why time is divided by 24 and 365 is not justified. I would consult a statistician for the proper way to describe each of these models. I am also very suspicious that these data follow any distribution that is part of the assumption of the models (Poisson and count data). Some of the predictors of the model should also be landscape features such as distance to road, elevation, slope, aspect, percentage of natural vegetation within fixed radii of the sensors, etc. The authors should consult Sangermano, 2022, *Landscape and Urban Planning*). Most of the data that are needed are already posted on Google Earth Engine and these can be extracted at Global Scales. In all honestly, I would use a Random Forest model which avoids the trappings of these linear models.

Lines 663-669. The results of the analysis suggest to me that perhaps roads in both the urban spaces and nearby natural environments are being detected. Indeed, highway noise can be detected about 2km away from the road and so there are few places one can be in urban and peri-urban areas without hearing road noise. Sensors should have been placed in the interior of protected areas and inside (an core area) the natural area if that is a comparative type of location desired by the authors.

Acknowledgements

863-897. This can be rewritten to be more succinct (reduce the length by about a 1/3rd).

Author contributions

I have co-authored several papers that have this many co-authors and I find the term "commented on the manuscript" not to be very useful nor adequately describe their contribution. An email that asks if they are OK with the current paper is not a contribution to the writing of the paper. Can the authors describe what is done to merit this level of contribution?

Figures

Figure 1

Lines 846-847. This is a mess. I'm confused by the three subfigures, especially B. The figure caption does not specify important parts of the diagram (why is the y-axis in B not labeled). Figure C is most informative as it tells me that the authors have not used acoustic indices properly. When any of these acoustic indices have highly skewed distributions, then the parameters that are used are not correct. This is a huge problem in this field where authors simply download the code and then use the default setting without thinking through what they are supposed to measure. Take ADI for example. ADI measures the amount of acoustic activity in each frequency band above a "noise floor". If the noise floor that is selected is too low (or if the preamps are set high on the microphone), then you will get a saturated ADI values (close to 3). Researchers need to do a sensitivity analysis of the key parameters for each of the ADI noise floor parameters; since the original ADI was created using condenser microphones and Audiomoths use an entirely different technology, values closer to 0 dBFS are probably best. Researchers need to plot out ADI values across the entire range and select the one that has a broad distribution. I suspect that similar problems are arising with ACI, H, Speech and Vehicle. I am not sure what part of Figure A is meant to describe the two spectrograms. There is a lot listed that is not clear. For the table -- Natural and Urban columns -- I assume are means?

Figure 2

These are very interesting plots. The ones on the left, however, are suspicious to me since the timing of dawn is a function of time of year and latitude. Second, the acoustic phenology curves to the right, also very interesting, lack proper context. For example, in areas of the world that experience four seasons, winter, spring, summer and fall; these are in the mid-latitudes. In the tropics, some areas have two seasons (wet and dry) and some have four (two wet and two dry). Areas at 30°N and S create, by Hadley cell circulation patterns, deserts which typically experience one very short rain event in the form of a

monsoon. So, latitude is indeed important but the phenologies should conform to these seasonal patterns that are driven by global climate regimes. Also, mountains throw a wrench into these global patterns. In the end, an interesting set of plots, but they do not tell me what I would expect. The textbook by Lomolino (Introduction to Biogeography) describes these global, biosphere patterns well and this is a good source to understand global ecosystem distributions.

Figure 3

Most useful figure in the paper. Label the x-axis in Figure 3B.

Figure 4

Very interesting plots. I wonder if the authors should consider doing a Jaccard Index of Dissimilarity to determine how similar species compositions between sites are. The urban+natural plots suggest that these are fairly different in species composition. For Figure 4A and B, is a linear model fit here or is that line just a reference point (at which case it should be removed or at least explained).

Figure S1

I would consider this as Supplement 1 and then the rest packaged as Supplement 2. The lack of any description of the dozens of (colorful) plots is frustrating.

Figure S2

I am seeing a lot of these Spearman rank correlations in papers and these authors fall into the same trap of how they are supposed to be used to interpret acoustic indices. First, if they do not correlate with one another, then the two acoustic indices can be considered to be different measures of the soundscape! Second, if they strongly correlate with one another, then you can drop those that are in future analyses that consider multivariate analysis. I'm confused by their general interpretation that a mixture of acoustic index values means that they are not informative. Third, are all of these correlations significant? Authors need to provide a level of significance and also adjust to the overall experimentwise error rate and use either the Bonferroni or Dunn-Sidak method to adjust the alpha significance level.

Figure S3

This is a good feature of the paper (to present validation of BirdNET). Earlier in the paper, the authors describe the two kinds of errors that should be assessed (False Negatives and False Positives). However, these plots only illustrate False Positives and True Positives. We are finding False Negatives to be a huge problem with BirdNET. There are many metrics that assess these model user and producer errors and these should be used to determine whether BirdNET is good. That said, the selection of 0.8 seems reasonable using this guide but it still does not provide a measure of False Negatives. It might be possible to extract that from their listener database. I suspect that False Negatives are higher in areas outside of North America and Europe.

Table S4

This is a useful table and I suggest leaving this in and distribute as an Excel spreadsheet.

Figure S5

This was unreadable for me and given that axes are not labeled, I can't comment on it.

TextS5

Should this be TextS6?

I am not certain how these habitat/trait variables are used or are then associated with each species of bird (I assume that is what is being done here). As mentioned above, many landscape variables are simply missing that I believe are strong predictors of temporal patterns of the soundscape and wonder why they are not considered.

Table S5

Should this be Table S7?

Experimentwise error rate should be adjusted in my opinion. Please consult a statistician.

Table S6

Hmmm.....these are important results that are not mentioned (seems entirely new information) in the body of the main text. It is correct that BirdNET does not do well in tropical regions but this paragraph buried deep in a supplement raises a few red flags for me. Content like this should be placed in the body of the main text and placed in the context of limitations.

Table S7

Useful table. The team name is not useful to readers. List co-author with their initials instead.

Figure S1 (Map)

Why not include this overlaid on a map of the continents? Use of ArcGIS would be helpful. No legend is provided. Sites names run over one another. Overall, a very unprofessional map.

Figure S1 (Multicolored Plots)

A set of $4 \times 27 = 108$ plots are not described at all and none have a y-axis labeled. There are no descriptions to these. Nor is there a legend unless the legend is the S1 Map color coded dots. Were these included since they are so colorful? Waste color toner and paper for me.

Conclusions

The value of this paper is the data. It is highly unique and the authors are able to address a very unique set of questions. However, it is poorly written, analyses in several areas are flawed, the *raison d'être* is not well posed, and many of the figures are confusing. A solid paper will require some significant reanalysis and also robust, concise, precise and informative prose (a complete rewrite).

Reviewer #2 (Remarks to the Author):

1. Key results: Please summarise what you consider to be the outstanding features of the work.
 - 1.1. The study described provides the largest spatial and temporal survey and analysis of the drivers of soundscape patterns to date.

1.2. To the best of my knowledge, this is the first large scale study that attempts to factor out anthropogenic, geophonic and biophonic components and investigate differences in seasonal and diel patterns between urban and natural areas, and across latitude.

2. Validity: Does the manuscript have flaws which should prohibit its publication? If so, please provide details.

- My key concerns are to ensure that conclusions are clearly substantiated. This is easily achieved through changes to the text rather than further analyses.
- There is also some need for greater precision. Main para 5. Text describes an analysis of bird species diversity, but some some proxy of richness is analysed and reported.

3. Originality and significance: If the conclusions are not original, please provide relevant references. On a more subjective note, do you feel that the results presented are of immediate interest to many people in your own discipline, and/or to people from several disciplines?

3.1. The results are of relevance to a range of areas of study:

3.1.1. The use of soundscape metrics is currently under debate in ecoacoustics; the global models created provide clear explanations for the importance of factoring latitudinal (a proxy for ecozone and acoustic community assemblage) into account.

3.1.2. The global patterns observed are significant findings for soundscape ecology and ecoacoustics and provide valuable contexts for national and regional studies.

3.1.3. The data provide valuable baseline information for future climate-focused phenological and avian migration studies

4. Data & methodology: Please comment on the validity of the approach, quality of the data and quality of presentation. Please note that we expect our reviewers to review all data, including any extended data and supplementary information. Is the reporting of data and methodology sufficiently detailed and transparent to enable reproducing the results?

4.1. Method: Rationale for study design choices are not completely clear. In particular, what does the Nested design allow you to answer that Global and National do not? Tradeoffs between spatial acuity and coverage? One sentence is sufficient.

4.2. Method: Modelling. Can you provide more details on how the global periodic models were created.

4.3. Method:

4.3.1. spectral energy. Please state rationale for / assumptions behind frequency band choices.

4.3.2. Acoustic indices. Please state rationale for 300Hz cut off. If you are analysing anthropophony, is that not useful info – e.g. engine noise etc.

4.3.3. If the choices of spectral energy contained assumptions central to your experimental aims, did you map settings of acoustic indices (BI, ADI and NDSI in particular) to these same values? Parameterising these for global models seems hard different species in different areas will have different frequency ranges.

4.3.4. Acoustic event detection. Please state whether you validate the YAMNet outputs and how.

5. Appropriate use of statistics and treatment of uncertainties: All error bars should be defined in the corresponding figure legends; please comment if that's not the case. Please include in your report a specific comment on the appropriateness of any statistical tests, and the accuracy of the description of any error bars and probability values.

5.1. Regression seems appropriate; p values and error bars are clearly labelled.

6. Conclusions: Do you find that the conclusions and data interpretation are robust, valid and reliable?

6.1. The key finding is presented as “We conclude that the global soundscape is formed from a highly predictable biophony, with increasing noise from poorly-predictable geophony and anthropophony”. You should be explicit about the basis of this claim. Are you inferring predictability from total R^2 ? Bird species richness has a lower R^2 than speech or vehicle, and you state elsewhere that the soundscape metrics show relatively weak correlations with biodiversity. As you have not tuned the metrics to known local biophonic ranges, you need to spell out how you are confident that these metrics capture biophony? This can be rectified by elucidation in the results and/or discussion.

6.2. A small but important point: The final sentence of the Summary states “humans and other animals across the world are experiencing and communicating against a drastically different sound background than ever before.” But this is not a longitudinal study – afaics there are no data from which to infer a temporal change, it is just assumed. Consider recrafting closing statement around your data? E.g. something around anthropocentric influence on the rhythms of wider biological life?

6.3. Discussion opening sentence could also be more on point! “We used systematic audio recordings around the globe to measure how the planet is buzzing with sound” – strong suggest to focus on actual key contribution e.g. “Through systematic world-wide acoustic survey we characterised the drivers of the global soundscape for the first time”

6.4. As it is so central to main claim, please clarify what you mean by anthropophonic sound – vehicles and voices? E.g. How pronounced are differences, para. 1 “Such anthropophony showed either little rhythm in time, or a rhythm substantially different from that generated by animals and abiotic forces (Fig. 2; Supplementary Fig. S1).” – can you label S1 and be clear? Vehicle and speech show strong diel patterns ... OK, they are different to wider soundscape, but they have a diel pattern.

6.5. Similarly, you write “clearly invaded by anthropogenic sounds” – “invaded” seems a strange choice for urban green spaces? It is essential to clarify what you are referring to.

Overall this is a good paper, but the final conclusions do not seem well justified. Comments line by line:

“Overall, we find strong diurnal and seasonal rhythms in the global soundscape – overlaid with a strong human signature.”

- Humans are part of the global soundscape, aren't they? Do you mean “differing biophonic and anthropophonics signatures”?

“Anthropogenic sounds are ubiquitous and widely invade the soundscapes of urban green spaces”.

- If anthropogenic sounds includes human voice “invade” seems like a strange choice for urban green spaces? “Thus, humans and other animals worldwide are now experiencing and communicating against a drastically different sound background than ever before.”
- Forgive me if I am missing something, but as above, there are no longitudinal data here, how is this statement substantiated? Something about the impact of human activity on rhythms of the rest of life seems more justified. “How this change in the global soundscape will affect animal communication and human well-being is a territory only partly explored.”
- Fine.

7. Suggested improvements: Please list additional experiments or data that could help strengthening the work in a revision.

- 7.1. Main: para 2 & 3. Do you mean “diel” rather than “circadian” – the latter being endogenous?
 - 7.1.1. Note this depends a little on whether soundscape is considered from as a subjective/ perceptual phenomena or objective. Consider qualifying and confirming perspective taken here? (see refs suggestion)
- 7.2. Main para 3. Consider referencing other disruptions to usual diel patterns such as light pollution?
- 7.3. Main. Para 4. “How green spaces can conserve acoustic environments” feels like a strange construction. “the relationship between green spaces and emergent soundscape”?
- 7.4. Throughout. Having defined soundscape as acoustic environment, suggest sticking to it throughout, rather than using “acoustic world”, “acoustic landscape” etc
- 7.5. Throughout: check speech marks.
- 7.6. Throughout: Pick one term for descriptors – e.g. “soundscape metrics” and use throughout. You use acoustic indices, soundscape indices, metrics etc
- 7.7. Methods/ design/ temporal sampling: Can you clarify whether absolute hour or time relative to sunrise were used? If the former, were results adjusted relative to dawn? Else how were diel patterns modelled through the year?
- 7.8. Figure 1. B) is pretty hard to read, could this be represented more clearly?
- 7.9. Fig1 C. Small, point, but would “metric” be a better column header than “index”? spectral energy is not really an index and you use metric in the text.
- 7.10. Fig 2. Can you clarify whether latitude in key refers to latitudinal range and how they are grouped for ease of cross-reference with S1
- 7.11. S1. Please clarify units on the y-axis. It is not clear what negative frequency means on the spectral metrics
- 7.12. Fig 1 B. Do you have anything to add on what “animal” sounds were dominated by? Given increase in urban areas, was this dogs and foxes, these are very closely related to human-influence.
- 7.13. Please label subplots in S1 for ease of cross-ref.

8. References: Does this manuscript reference previous literature appropriately? If not, what references should be included or excluded?

- 8.1. In intro paragraph of Main, consider including recent debate over validity of foundational evolutionary and ecological theories of ecoacoustics (e.g. Alcocer, I., Lima, H., Sugai, L.S.M. and Llusia, D., 2022. Acoustic indices as proxies for biodiversity: a meta-analysis. *Biological Reviews*, 97(6), pp.2209-2236.)
- 8.2. Main para 2. Consider including more recent work qualifying soundscape (see 7.1.1)
 - 8.2.1. Farina, A., Eldridge, A. and Li, P., 2021. Ecoacoustics and multispecies semiosis: Naming, semantics, semiotic characteristics, and competencies. *Biosemiotics*, 14(1), pp.141-165.
 - 8.2.2. Grinfeder, E., Lorenzi, C., Hauptert, S. and Sueur, J., 2022. What do we mean by “soundscape”? A functional description. *Frontiers in Ecology and Evolution*, 10, p.894232.
- 8.3. In methods – cite ref for Global Footprint Index

9. Clarity and context: Is the abstract clear, accessible? Are abstract, introduction and conclusions appropriate?

- 9.1. Main para 5. Needs a concise introduction to acoustic survey to link from previous paragraph. E.g. “large scale acoustic surveys can now be undertaken with relative ease, but generate vast data sets. Analyses requires automation and current research adopts soundscape metrics that provide statistical summaries of “ Etc.)
- 9.2. Lellouch is not the best ref. here. Consider Sueur, J. and Farina, A., 2015. Ecoacoustics: the ecological investigation and interpretation of environmental sound. *Biosemiotics*, 8, pp.493-502.
- 9.3. Main. Final question is not clear – “what features do these apply to” – do you mean: in which metrics are rhythms observable?
- 9.4. Discussion. Related to 6.1: You suggest that soundscape metrics do not correlate well with biodiversity, but also acknowledge the BIRDNET may not be giving accurate results for species outside US and Europe. If these metrics are not reflecting biodiversity, do you still assume they capture biophonic variation? Please elucidate a little further in discussion as it is so core to conclusions.

10. Inflammatory material: Does the manuscript contain any language that is inappropriate or potentially libelous?

No

11. EDI: Springer Nature is committed to diversity, equity and inclusion; please raise any concerns that may in your view have an impact on this commitment.

NA

12. Expertise Please indicate any particular part of the manuscript, data, or analyses that you feel is outside the scope of your expertise, or that you were unable to assess fully.

- 12.1. My expertise is in theory and methods of ecoacoustics – I am not a stats expert. The editor should ensure that someone

with requisite expertise thoroughly reviews the model construction and inference.

13. Please address any other specific question asked by the editor via email.

Version 1:

Decision Letter:

25th April 2025

Dear Otso,

Thank you for submitting your revised manuscript "Less predictable global rhythms in human than wildlife contributions to soundscapes" (NATECOLEVOL-24061733A). It has now been seen again by the original reviewers and their comments are below. The reviewers find that the paper has improved in revision, and therefore we'll be happy in principle to publish it in Nature Ecology & Evolution, contingent on final revisions to satisfy the reviewers' remaining requests and to comply with our editorial and formatting guidelines.

If you have not done so already, please ensure that you also email us completed copies of the Reporting summary and Editorial policy checklists:

Reporting summary: https://www.nature.com/documents/nr-reporting-summary.pdf

Editorial policy checklist: https://www.nature.com/documents/nr-editorial-policy-checklist.pdf

Next steps: Once you send us the Word file, we will perform detailed checks on the current version of your paper and send you a checklist of our editorial and formatting requirements in about a week. Meanwhile, you can start revising the manuscript but please do not upload any final materials to the submission system until you receive the formatting checklist from us. Once you receive the checklist, you can incorporate the (usually fairly straightforward) formatting changes in your revision. Then, you can upload the final materials to the submission system.

[redacted]

Reviewer #1 (Remarks to the Author):

see attachment

Reviewer #2 (Remarks to the Author):

The authors have made thorough revisions to the manuscript which have significantly improved the submission and does better justice to this excellent data set.

To my mind there remain a few small edits that could further improve clarity, veracity and reproducibility

Main

Para 1

"The resulting distribution of sound amplitudes across frequencies determines how efficiently animals can communicate with each other"

>> I realise you have changed this from "affects" but that is more correct. Or "partially determines" if you prefer as there are other factors – for example landscape and vegetation structure as well as weather will also impact.

Para 3.

"the dusk chorus is dominated by insects"

>> This has been suggested by the other reviewer, but is not true in many biomes of temperate northern hemisphere. Please qualify.

Results

P7. Para 1

"Overall, the urban soundscape is driven by the structure of land use, and across cities, the amount of vegetation – a metric of green infrastructure – correlates with the intensity of sound"

>> Does vegetation density correlate with biophony? Or overall soundscape intensity? Or is it inversely correlated with overall sound intensity (ie in built up areas no green = lots of traffic)? Please clarify.

P7 para 2.

>> Suggest using "quiet" rather than silence as you have throughout?

Ibid Para 5.

"Across other indices, urban green spaces proved generally noisier (index SpecLow, Speech and Vehicle) than their more pristine counterparts"

>> Do you define "noisier" somewhere? Might "higher in anthropophony" be better?

Discussion

"well-predictable" is grammatically incorrect

>> suggest strongly or highly predictable?

Conclusions

"Given the potential for anthropophony to mask current animal communication and its strong contributions to global soundscapes, it should offer a strong selective force"

>> "It is likely to exert a strongly selective force"? use imperative here feels unusual and doesn't carry what I understand to be your meaning

"Yet, poor predictability in space and time will compromise such selection, and given taxa seem constrained in adjusting their acoustic signals to overcome noise"

>> Missing clause, please complete the sentence

METHODS

Spectral energy

Thanks for putting in the FFT params, but this is an unusual way of reporting

"Spectrogram was computed from a one-minute recording using a 1024-point FFT with Hann window at 10 ms intervals using the Python library librosa"

>> For a 1024 point FFT at 48000 kHz, each window is $1024/48000$ ms = 21.333 long. 10 ms is not an exact multiple of this. For clarity and reproducibility, do you mean you used a 50% hop size (overlap) ie 512 points?

Reviewer #2 (Remarks on code availability):

I have reviewed but not run the code.

It is clearly organised, well documented and looks to include everything needed to reproduce the results of the paper.

Reviewer #1 (Remarks to the Author):

The authors of the paper “More predictable rhythms in the natural than anthropogenic global soundscape” conduct an analysis of over 1.4 million audio recordings of the soundscape from 139 sites located on six continents. Recordings for each site were in a pairwise design (urban and peri-urban settings. One-minute recordings were analyzed in the context of the proposed “rhythms of nature” conceptual framework of Pijanowski et al. (2011) that describe the predictable diel and seasonal patterns of biophonies that should exist around the world. The authors use a set of standard acoustic indices (ADI, ACI, etc.), measures of sound sources that are extracted from an acoustic event detector and also apply the widely used CNN-based BirdNET tool to detect bird species. They find that for the most part, the rhythms of nature are indeed predictable in key time periods and across latitudinal gradients. Another key finding is that the human noise footprint is interfering with these natural rhythms which are key functioning features of all ecosystems. They conduct a set of spatial-temporal analyses of acoustic indices, sound source event metrics, as well as the species’ habitat features. I do like the topic of the paper and I think many in ecology not familiar with the concept of the soundscape will be as well. However, there are numerous shortcomings of the paper. I spent three days reviewing this hoping to provide some insights for significant revisions and so this represents a major time commitment to make sure that if this moves toward publication, a lot of the errors and poor writing can be corrected. I hope the editors and authors find these to be helpful.

Reply: We are most grateful for the intensive effort invested by the Reviewer in helping us achieve a worthwhile contribution. In response, we have made every effort to make full use of each suggestion offered. To pinpoint the changes made, we will address each comment in turn while highlighting the changes made.

First, I found the paper not to be well written; in fact, compared to most Nature/Science/Ecology Letter/PNAS papers I read, this paper lacks the high level of density, clarity, robustness and precision of the prose I would expect from these articles. One of my best illustrations for this is the sentence in line 465 “No matter where we are and when we listen, there is sound”. This is blatantly obvious to all of us in this field and does not address any of the research questions. Another is the opening of the Discussion “...how the planet is buzzing with sound”. These rather “folksy comments” provide no factual content and are not very scientifically useful (actually, even a bit misleading since this paper is an analysis of the patterns of the presence of sound in the environment and not how people perceive it). Overall, the language and interpretation throughout the paper is “loose”. There is what I would call a lot of hand waving without getting to robust interpretations of the analysis. The core paper is easy to follow although the supplement got difficult to navigate at times (a supplement to a supplement was confusing).

There are MANY terms (spectral, amplitude) not used very precisely either. I would highly suggest consulting the glossary of terms that are in Pijanowski and Rivas (2023; Encyclopedia of Biodiversity, Elsevier, Soundscapes and Vibrosapes, Chapter 134) for the most commonly used terms in soundscape ecology and how they should be used.

Reply: We have now rewritten the full text, aiming for a more stringent expression. In doing so, we have made every effort to achieve both precision and accuracy in our use of the terminology (drawing on the source offered by the Reviewer) and specifying the interpretation of the analysis. The structure of the supplements has been revised.

Let me state up front that this is a paper worthy of eventually being published in Nature Ecology and Evolution. The two main “take aways”, that the soundscape is a predictable feature of ecosystems and that human noise is potentially affecting these rhythms is an important message to make to large communities of scholars and natural resource managers. I will for one cite it a lot once it has gone through some significant revisions. In my opinion, there is a lot more work that is needed to get the paper to a publishable form. There are areas that are quite vague and specifics of analyses and calculations are simply not

described. A large body of literature (on urban soundscapes such as Lex Brown, Jian Kang and Östen Axelsson and biodiversity assessments a la conservation biology work of say, Zuzana Burivalova) are missing and the take home points are not made all that strongly. In general, there are large bodies of literature missing from the paper that would make this a very informative paper across basic and applied ecology and evolution.

Reply: We are most grateful to the Reviewer for emphasizing the value of our contribution. In response, we have sharpened the take-home message, included substantial literature on urban soundscapes and added references to biodiversity assessments. The description of the analysis has been thoroughly revamped. In adding pointers to the wider literature, we must still stress that our hands are partly tied. Nature Ecology and Evolution recommends a maximum of 50 references, for which reason our original contribution included 41 references. The current count is 63, of which we feel all are needed.

The rest of my comments, which address the more general ones listed above, are summarized here:

Manuscript (i.e., Main)

Lines 329-337. It should be made clear here if the authors truly mean anthropophony or technophony as their major sound source category. The latter was proposed by Gage after the Pijanowski et al., (2011) Bioscience paper to describe sounds sources that do not include human voices. The list of examples seems to suggest technophony but the analysis includes human voices (some of us consider human voices to be part of the biophony since we are indeed animals). If we consider human voices, to what extent to these contribute toward the disruption of the natural rhythms (perhaps some of their multivariate analyses might point to that but I suspect that this is minor). My opinion is that the authors should focus on technophony since that is the “noise” component that is being assessed with most of the analyses and aligns with the paper’s objectives.

Reply: We have now tried to make a clearer distinction between different sound source categories. Indeed, one of the main objectives of the manuscript is to compare natural and urban environments using multiple acoustic features (i.e., the ones listed in Figure 1). Among the features resolved by our analysis, human speech is indeed treated a separate category (see Figures 1, 3, and 4 and Supplementary Figures). Thus, the Reviewers concern relates to what wider category the individual sources are subsequently grouped into. To avoid any unclarities, we have now explained what exact choices were made, how categories were formed and what they consequently consist of. In doing so, we have made sure to specify our conclusions with respect to what sources they relate to.

Lines 338-347. Paragraph could be stronger and include more specifics. It is known, for example, that birds and amphibians dominate the dawn chorus and that the dusk chorus is dominated by insects. It should also be mentioned that the timing of, and the length of, both the dawn and dusk chorus differs by latitude and time of the year.

Reply: We have now strengthened and elaborated on the paragraph in question. We have also emphasized that our study adds the first fully standardized quantification of the global and seasonal patterns proposed by the Reviewer – a contribution which has now been highlighted.

Lines 348-355. I find this paragraph to be far too general and some of the noise patterns are indeed diel. First, the sounds of highways are likely to be most loud prior to the start of the business/school day and then again most loud during the end of the business/school day. So, I’m not sure I agree with the authors. Second, each ecosystem has a predictable soundscape phenology that is related to the phenologies of major sonically active animals (here in the midlatitudes, amphibians emerge early spring, then come breeding and migratory birds, and then the insects). These major animal sonic groupings also have different acoustic patterns (pulsating, melodic). There is a lot of literature to support these kinds of patterns and they should

be clearly described here. This is also the paragraph where a lot of the urban soundscapes research should be cited (Brown, Kang, Axelsson). See Chapter 13 in Pijanowski's book (2024) for a thorough review of the work in urban soundscapes.

Reply: We fully agree and have now added further coverage of the considerations offered by the Reviewer. In doing so, we have greatly benefitted from the recent book by Pijanowski.

Lines 356-363. There is a lot of work done on the US NPS soundscapes as a natural resource (see Dumyahn and Pijanowski, Landscape Ecology 2011, Managing soundscapes as common pool resources) and also the work of Kurt Fistrup and Peter Newman at Penn State University). I'm not certain this is a useful paragraph as written.

Reply: In revising the manuscript, we have omitted the original paragraph and added further consideration of soundscapes as natural resources.

Lines 364-378. This paragraph does not precisely use the terms properly and so I was a bit confused with what they were referring to by "metrics", "features", "descriptors", and "representations" which are referenced in citation #13. I could not find those specific terms there. Figure 1 is referenced and that figure (which needs A LOT of work) does not provide the reader with what these terms refer to. The authors need to clarify the use of these and perhaps provide a diagram that illustrates how these differ, especially as they relate to acoustic indices. I am also confused by the use of the term "voice abundance". Some researchers have used the term call rates. Also, the field has not settled on how to use the number of bird species calls in any analysis. What you are specifically doing here, I think, is to use BirdNET to create a species richness value per recording and then per site.

Reply: The different terms were used to avoid the very manifold repetition of a single term such as "metrics". Thus, they are synonyms of each other. To avoid any confusion, we have now chosen a single term to be used throughout the text. In Figure 1 there is short description of each measure and for full description (regarding the soundscape indices) we have referenced the original papers. For BirdNET outputs, we have calculated two measures, one of which is species richness and the other call rates. These aspects have now been clearly defined in the text.

Lines 379-383. I am almost certain I know what you mean here in your statement of the paper objectives. But these are clumsy at best. Could I suggest: "how pronounced and predictable are global biophonies across diel and seasonal patterns?" and "does noise produced by humans affect these natural rhythms?" that is then followed by "We test examine each of these questions using a global soundscape database and a paired urban-rural experimental design".

Reply: We apologise for the unclear wording and have changed the wording of the paragraph.

Line 373. I'd exclude human voices from the list of anthropophony (by the way, anthropophony and anthrophony have been determined by a scholar in Latin-Greek to be synonymous).

Reply: Since we have calculated different measures separately, the grouping of them can be done in many ways. It is then a matter of choice whether to include human voices in the category of anthropophonic sounds. Importantly, this will not change our analysis, as human speech is included as a separate category, and the exact patterns detected in this category are shown in Figs. S2-S55. In principle, we are also hesitant to remove human-generated speech from the category of human-generated sounds (anthrophony). To avoid any ambiguities, we have now clarified the description of what exact original metrics are grouped in what wider categories, and discussed how human voices relate to the class of antropophony.

Lines 386-394. This is the second use of the term circadian rhythms. Authors have used diurnal through to now. Other researchers use diel which is probably the best term to use of the three. I'd stay consistent with the terminology throughout the paper. As I am a bit confused by the term "descriptors", it was difficult for me to follow this first paragraph in Results.

Reply: Thank you for pointing this out. We have changed the text to use more coherent terminology.

Lines 395-403. This is a very difficult paragraph to embrace since there is not much detail in what the authors did. The hourly patterns I think are a challenge to compare across latitudinal gradients since the timing of dawn varies by date and latitude. Using just the local clock time is misleading. Instead, the authors should have indexed the time of the recording to be something of a standard "time since civil sunrise". Indeed, at 60° during the summer solstice, there is no period of complete darkness. If I had one serious criticism, is that they (I think) used local time across all sites which does not reflect the dawn. A second comment is that the dawn chorus at different locations and time of the year has different lengths. My extensive time at locations of the equator have taught me that the dawn and dusk periods are very short there nearly all year round.

Reply: We were thinking about the varying daylight time when doing the first draft of the manuscript but then decided to use only absolute local time. However, based on the Reviewer's comment we have now repeated all analyses using also the time that reflects the available sunlight based on the latitude, longitude, and time of the year during the recording. How we did this is explained in the Materials and Methods section.

Lines 423. I do not think that the use of the term "silence" is proper here. Silence refers to the lack of any sound. That only occurs in vacuums or anechoic chambers. Quiet is a better term and it needs to be described as very low amplitude of sound waves that are not detectable using field equipment.

Reply: The word Silence refers to the name of an explicit AudioSet class used by the YAMNet classifier. To clarify its use, we have defined the term when introduced and used the word "quiet" in the main text.

Lines 434-496. What the authors have found – that is, that nearby natural areas and not different than urban areas in terms of call rates, is not that unusual. There are MANY papers that have described how birds are now calling at higher pitches, are louder or are more active at night in response to the heightened ambient sounds from humans. Can the authors place their research in this large body of literature (e.g., see the work that describes the Lombard Effect)?

Reply: We have now discussed (with references) how birds change their vocalization in urban environments, and how this may relate to the pattern found. Having said that, our data do not provide direct evidence of changes over time (since they are from 1-2 years only) and do not compare pitches or activities within species between urban and natural sites. Thus, the section puts our observations in a context but adds no direct evidence.

Lines 454-560. The entire discussion needs to be rewritten. First, the authors are introducing (using a prominent subheading) "what does the word sound like?". This was not one of the key research questions and I am not sure they have answered it. The section in lines 465-475 do not address this question either. The presentation is overly general and very vague (handy waving of correlation patterns). The second subheading does not seem to be fully answered either (and this is what the authors state was one of the research questions). I'm confused by several of the sentences especially the one that contains (line 502) "traditional community data...". How do the recordings and the analysis "pinpoint the agents that are making sounds...?". I also want to point out that the fact that the acoustic indices do not always strongly correlate is a GOOD thing. They each are measuring a different aspect of the soundscape. ADI, for example, measures the diversity of total sound amplitude across audible frequency bands split into 1 kHz bands. ACI

measures the amount of modulation in time and frequency. NDSI measures the relative spectral power of sound in the frequency bands associated with biological sounds versus those produced by technophony. The role of the soundscape ecologist should be to decipher the complexities of the soundscape using these diverse tools.

Reply: We have now rewritten the entire discussion, taking the valuable pointers of the Reviewer *ad notam*.

Lines 510-520. One of the major issues and discussions right now is how to address the accuracy of BirdNET output. Interestingly, here the authors use the proper assessment descriptions (quantifying false negatives especially since BirdNET seems to have high values of these based on our assessments as well). First, I would not trust any output with confidence scores below 0.50 so I am not certain why they would consider anything less as these are neural net tools that learn threshold into binary output data (presence and absence). Second, one needs to develop methods that assess false negatives (a bird called and it was not detected at all). Third, call activity levels is driven by taxonomic considerations (nocturnal nightjars call repeatedly whereas many higher-level passerines call infrequently and are not that repetitive in their call structure). I'm not sure I would use call rates in the analysis until the research community can determine how to use BirdNET data that contains a lot of this information.

Reply: We calculated two measures from BirdNET outputs. The first was the number of species detected (detected species richness) and the other was call abundance (i.e., how many times the detection exceeded the given threshold). The latter one is now referred to as “total call rate”, following the suggestion by the Reviewer. We find it interesting to show results of both approaches. We are not claiming that one is better than other, but aim to provide the reader with a clear view of results from both approaches. We also believe that this is an important step towards helping the research community determine how to use data from BirdNET and from other, similar and quickly-accumulating tools.

With respect to detection thresholds: there is definitely a trade-off between precision and recall. In Supplement 1 we show the results using two different thresholds 0.8 and 0.3. Indeed, the value 0.3 is very low and the results contain many false positives. However, the interesting point is that the shapes of the activity patterns observed using these two different thresholds look qualitatively similar. We are not claiming that BirdNET gives exactly true species richness – rather, it is a proxy of it. However, as BirdNET is currently the state-of-the-art method for automatically identifying bird species, this is what the current methodology is capable to perform. Importantly, what we observe *despite* the possible shortcomings of BirdNET is that its output is useful to reveal temporal patterns. These patterns emerge even against the background noise generated by the false positives and false negatives unavoidably generated by automated detection methods.

Lines 554-560. This is the weakest section of the paper. The authors need to think more deeply about the implications to their results. Why should ecologists care? I can think of dozens of reasons that rise to the very top of highly crucial issues but these are not included anywhere in this paper and certainly not here. I think addressing the work in the context of the biodiversity crisis, climate change, conservation management, urban planning and even human-nature interactions are all extremely important. Indeed, this gap is evident in the introduction and abstract. Serious thought needs to be put into this part of the paper to make it worthy of publication in Nature Ecology and Evolution.

Reply: Thank you for the comment. We have now rewritten the Conclusions to focus squarely on the implications brought forth by the Reviewer.

Materials and Methods

Lines 565-571. Paragraph needs some editing.

Reply: The paragraph has been edited for clarity.

Lines 572-595. A figure here that depicts the experimental designs would be helpful.

Reply: Fortunately, the exact design and protocols used have now been published in a recent paper (Hardwick et al. 2024. LIFEPLAN: A worldwide biodiversity sampling design. PLoS ONE 19(12), e0313353). To avoid any confusion, we now refer to this paper for the general design – whereas all the specifics of the material used in the current paper are shown in the revised figures.

Lines 618-623. FFT models are not described. What window size and type were used to process the wav file for use in the analysis? I assume 512 and Hanning?

Reply: 1024-point FFT was computed every 10ms and a Hann window was used. We have now added this to the text.

Lines 625-631. Many of the acoustic indices used were developed using Wildlife Acoustics SM2s, and in some cases, SM4s. Some of the acoustic indices, such as ADI and ACI, are also sensitive to parameter settings and microphone sensitivities. For example, ADI was originally designed to be used for SM2s and with preamps set to the factory settings. This means that the “noise floor” of -50dBFS was ideally configured for that model. Audiomoths use an entirely different set of microphone technologies and there are a host of preamp settings. So, I am weary of folks that use default settings on these acoustic index tools. We are also finding that ACI is sensitive to FFT window sizes (makes sense when you think about about) but also the Nyquist frequency (which might make sense) for certain FFT window sizes (large ones). THUS, it is extremely important to (1) make sure the acoustic indices are given values that make sense (ADI does not look right to me) and (2) that a sensitivity analysis of key parameters be undertaken to make sure these are effective parameters to use). Finally, a high pass band filter applied AFTER the recordings are created does not make sense to me. I sometimes like to use the 250 Hz HPF on the device as this makes the microphone more sensitive in the biologically active areas. Using a HPF will affect ADI and NDSI, although, if these are applied across all recordings the same, then you should be OK. In short, the use of acoustic indices is somewhat careless.

Reply: Thank you for this point. Indeed, we used ADI with default values where the noise floor was set to -50dB. We have now recalculated ADI with different values and decided to use the results for the noise floor set to -26dB. We have also experimented with multiple window lengths for ACI. Scatterplots of ACI results with different parameters did not suggest any particular value to be the best. Therefore, we decided to use the default value a 512-point window length also in the revision. For other soundscape indices, we feel that the previous values were uncompromised as they were.

Lines 646-653. The statistical analysis section is one of the most confusing parts of the paper. I have no idea what kind of model is being developed. The description of the model’s explanatory variables is not described as I would. Why time is divided by 24 and 365 is not justified. I would consult a statistician for the proper way to describe each of these models. I am also very suspicious that these data follow any distribution that is part of the assumption of the models (Poisson and count data). Some of the predictors of the model should also be landscape features such as distance to road, elevation, slope, aspect, percentage of natural vegetation within fixed radii of the sensors, etc. The authors should consult Sangermano, 2022, Landscape and Urban Planning). Most of the data that are needed are already posted on Google Earth Engine and these can be extracted at Global Scales. In all honesty, I would use a Random Forest model which avoids the trappings of these linear models.

Reply: We have now improved the descriptions of the methods. The statistical models that we apply are among the most commonly applied models in ecological statistics, i.e., generalized linear models. For the periodic functions of hour of the day and day of the year, the values 24 and 365 are the number of hours per day and number of days per year, respectively, and dividing by them is needed to obtain basis functions that are periodic over the time of the day or over the day of the year. For two different regression models,

we used linear regression for continuous-valued data and Poisson regression for count data. While count data will often show overdispersion incompatible with the Poisson model, the counts here analysed concern species richness and show no such overdispersion. To measure how much variance in the data the model describes, we used simple R^2 values. Concerning the suggestion of using predictors such as distance to road, elevation, slope, aspect, or percentage of natural vegetation within fixed radii of the sensors, we fully agree that their effects would be interesting to study. However, our study design is not targeted at capturing their effects – as we have very many temporal replicates for relatively small number of study locations, rather than a very large number of study locations. The covariates that the Reviewer mentions are constant over time and thus fixed for each study location. Thus, including them in the model would quickly make the model overparameterized. To avoid this, we included only key site-specific variables directly related to the study questions.

Lines 663-669. The results of the analysis suggest to me that perhaps roads in both the urban spaces and nearby natural environments are being detected. Indeed, highway noise can be detected about 2km away from the road and so there are few places one can be in urban and peri-urban areas without hearing road noise. Sensors should have been placed in the interior of protected areas and inside (an core area) the natural area if that is a comparative type of location desired by the authors.

Reply: The locations of recorders were set by local teams. The natural locations were chosen to be the most natural ones present within a restricted distance from the urban one – just like the urban sites were classified as urban by regional, not global or absolute standards. Importantly, these are the urban vs natural sites accessible to the local human population, and thus represent the soundscapes that people can realistically switch between.

Acknowledgements

863-897. This can be rewritten to be more succinct (reduce the length by about a 1/3rd).

Reply: The Acknowledgements section has been thoroughly revised.

Author contributions

I have co-authored several papers that have this many co-authors and I find the term “commented on the manuscript” not to be very useful nor adequately describe their contribution. An email that asks if they are OK with the current paper is not a contribution to the writing of the paper. Can the authors describe what is done to merit this level of contribution?

Reply: The list of author contributions has been fully revised to lay plain the team-wide effort and the contributions made by the coauthors.

Figures

Figure 1

Lines 846-847. This is a mess. I’m confused by the three subfigures, especially B. The figure caption does not specify important parts of the diagram (why is the y-axis in B not labeled). Figure C is most informative as it tells me that the authors have not used acoustic indices properly. When any of these acoustic indices have highly skewed distributions, then the parameters that are used are not correct. This is a huge problem in this field where authors simply download the code and then use the default setting without thinking through what they are supposed to measure. Take ADI for example. ADI measures the amount of acoustic activity in each frequency band above a “noise floor”. If the noise floor that is selected is too low (or if the preamps are set high on the microphone), then you will get a saturated ADI values (close to 3). Researchers need to do a sensitivity analysis of the key parameters for each of the ADI noise floor parameters; since the original ADI was created using condenser microphones and Audiomoths use an entirely different technology, values closer to 0 dBFS are probably best. Researchers need to plot out ADI values across the entire range and select the one that has a broad distribution. I suspect that similar problems are arising with ACI, H, Speech

and Vehicle. I am not sure what part of Figure A is meant to describe the two spectrograms. There is a lot listed that is not clear. For the table -- Natural and Urban columns -- I assume are means?

Reply: Figure 1 has now been split into two figures, Fig. 1 and Fig. 2, with the figure legends revised to clearly explain the content of each. Following the Reviewer's suggestion, Fig. 1 is now dedicated to define each metric used and what is expected to encapsulate. To illustrate the information conveyed by each, Fig. 2 has now been dedicated to showing differences in indices observed between two example spectrograms. (This was also the case in the previous illustration, but admittedly the reference to examples vs. global distributions got highly confusing, for which we apologise.) For clarity, the global distributions of values across indices has been removed.

Having said this, we dispute the Reviewer's suggestion that the previous illustration would have showed "that the authors have not used acoustic indices properly". In panel C (now omitted), the distribution of each for acoustic index was shown between minimum and maximum. For some acoustic events e.g. Speech, Vehicle, and Rain, these values are probabilities. As these events can be rare, probability values close to zero were dominating. Importantly, distribution plots in now-omitted panel C included all data. Since these data include recordings from day and night throughout a year, there are many recordings where no animal or other vocalizations are found. Regarding ADI, we recomputed all values with new parameters. In the first version of the manuscript, we used the default noise floor value. Yet, the example on the right hand side in omitted panel C showed an ADI value of 1.53, and a value close to 3. We have now recalculated ADI after adjusting the default noise floor value from the previous value of -50dB to a new value -26dB. Also for ACI, we have experimented with different window sizes, but found no major scope for optimising this parameter.

Figure 2

These are very interesting plots. The ones on the left, however, are suspicious to me since the timing of dawn is a function of time of year and latitude. Second, the acoustic phenology curves to the right, also very interesting, lack proper context. For example, in areas of the world that experience four seasons, winter, spring, summer and fall; these are in the mid-latitudes. In the tropics, some areas have two seasons (wet and dry) and some have four (two wet and two dry). Areas at 30°N and S create, by Hadley cell circulation patterns, deserts which typically experience one very short rain event in the form of a monsoon. So, latitude is indeed important but the phenologies should conform to these seasonal patterns that are driven by global climate regimes. Also, mountains throw a wrench into these global patterns. In the end, an interesting set of plots, but they do not tell me what I would expect. The textbook by Lomolino (Introduction to Biogeography) describes these global, biosphere patterns well and this is a good source to understand global ecosystem distributions.

Reply: We fully agree that there are many climatic and other factors with a potential impact on acoustic phenology. What we perhaps failed to explain properly in the original submission is that quantifying the amount and nature of such variation is indeed one of the core aims of the manuscript. As we have now clarified, the comparison between the global model and the site-specific models is explicitly targeted at addressing this question. In the site-specific models, we quantify seasonal and diurnal variation separately for each site. Hence, the site-specific models fully account for the properties of the site (e.g. whether there are two or four seasons). In contrast, the global model is based on the simplified assumption that variation among the sites in their seasonal and diurnal variation only depends on latitude (as the global model includes the interaction between the periodic functions and latitude). Now, how much more of the data can be explained by the site-specific models than the global model then quantifies how important the site-specific factors are. The Reviewer felt that the left-hand panels are suspicious since the timing of dawn is a function of time of year and latitude. As mentioned, the model fitted here allows timing to depend on latitude, but the results suggest that the effect is quite minor. Thus, we consider this as a result rather than a suspicious assumption. In more detail, we believe that the Reviewer is especially concerned about two points: The first one is that the timing of dawn and dusk varies with season and latitude. To address this, we have now reanalysed the data with respect to local sunlight conditions -- and found patterns consistent with

the original ones. These analyses, the results and our conclusions have been described in full detail in the revised manuscript. Second, the Reviewer stresses the impact of factors OTHER than time or latitude. The impact of these factors is indeed addressed, as summarised by four additional predictors: the human footprint index, elevation, annual mean temperature, annual precipitation (and most completely by the site-specific models). The results are shown in Fig 2, with the main conclusion that latitude, time of day and day of year suffice to explain major variation in key metrics of biophony but not in anthropophony. These findings have been further highlighted in the revised version of the manuscript.

Figure 3

Most useful figure in the paper. Label the x-axis in Figure 3B.

Reply: We are grateful for the Reviewer's appreciation of the figure and apologise for the missing axis label. This label has now been added.

Figure 4

Very interesting plots. I wonder if the authors should consider doing a Jaccard Index of Dissimilarity to determine how similar species compositions between sites are. The urban+natural plots suggest that these are fairly different in species composition. For Figure 4A and B, is a linear model fit here or is that line just a reference point (at which case it should be removed or at least explained).

Reply: Thank you for suggestion, the Jaccard index would indeed describe some of the differences. However, we think that in the present form the plots (C and D) give the information in a more explicit way. Plots in C and D illustrate species accumulation curves, i.e. how the species richness (y axis) grows when increasing the number of detections (x axis). In panels A and B, we have taken the species richness corresponding to a fixed number of detections (1000) and plotted those values for natural vs urban sites (panel A) and for combinations of natural and urban sites (panel B). The result is that for samples of equivalent size, more species are detected in urban sites compared to the natural site, and that the set of species observed at the two sites in a pair are partly complementary. By comparison, a simple Jaccard index would be more sensitive to the number of false positives. Because species accumulation curves are based on data that include the information how frequently each species has been detected, they are less likely to be affected by single spurious detections.

Figure S1

I would consider this as Supplement 1 and then the rest packaged as Supplement 2. The lack of any description of the dozens of (colorful) plots is frustrating.

Reply: We were sorry to hear that supplement 1 plots was uninformative. In response, and after conferring with the Editor, we have broken the figure into 55 parts and added a clarifying legend for each plot. The purpose is to show results for both site-specific models (where models were fitted using only local data) and the global model (where data from all sites were included, with latitude as an explanatory variable). The latitude is identified by the colour of each curve, with the same colour used in the map of Fig. S1.

Figure S2

I am seeing a lot of these Spearman rank correlations in papers and these authors fall into the same trap of how they are supposed to be used to interpret acoustic indices. First, if they do not correlate with one another, then the two acoustic indices can be considered to be different measures of the soundscape! Second, if they strongly correlate with one another, then you can drop those that are in future analyses that consider multivariate analysis. I'm confused by their general interpretation that a mixture of acoustic index values means that they are not informative. Third, are all of these correlations significant? Authors need to provide a level of significance and also adjust to the overall experimentwise error rate and use either the Bonferroni or Dunn-Sidak method to adjust the alpha significance level.

Reply: We agree that they all are different measures of the soundscape – and we are not claiming that they would measure the same thing. Quite the contrary: in both the original and the further-revised main text, we stress the complementary nature of these metrics. The purpose of showing the correlations as colours is to convey the idea that some measures are more similar than others. However, we are not trying to calculate any statistical significance for the difference. This is a deliberate choice, since given the (very) high number of datapoints that we have created, even tiny differences between the measures would lead to statistically significant results. Thus, what is more relevant than “how confidently we can say that there is any difference”, is “how big is the difference”, and this is what we try to illustrate. The way in which we have compared them statistically is identified in the legend to Figure 3 of the main text, i.e., we compare them on the basis of how predictable they are as a function of time of the day and day of the year (as measured by R^2). The corresponding section of the main text has now been completely rewritten.

Figure S3

This is a good feature of the paper (to present validation of BirdNET). Earlier in the paper, the authors describe the two kinds of errors that should be assessed (False Negatives and False Positives). However, these plots only illustrate False Positives and True Positives. We are finding False Negatives to be a huge problem with BirdNET. There are many metrics that assess these model user and producer errors and these should be used to determine whether BirdNET is good. That said, the selection of 0.8 seems reasonable using this guide but it still does not provide a measure of False Negatives. It might be possible to extract that from their listener database. I suspect that False Negatives are higher in areas outside of North America and Europe.

Reply: Indeed, false negatives are higher in areas outside of North America and Europe. Nonetheless, to quantify false negatives across continents would require a tremendous amount of manual work (with global bird experts listening to recordings) -- for which reason we have been forced to abstain from it in the current manuscript. Nonetheless, we stress that we are currently investing heavily in the annotation of improved sound libraries (see <https://bsg.laji.fi/> with an advertisement accessible under <https://helda.helsinki.fi/server/api/core/bitstreams/cecabf84-4f20-439d-ae6c-57b81658a44d/content>).

For some locations, it is still possible to evaluate the rate of false negatives offered by BirdNET. E.g. in Madagascar, the current BirdNET model includes only half of the bird species known to occur in the region. For the European sites, we have now used or manually curated subset of data (n=615 clips of 3 sec each) to calculate false negatives. The values are 60% with BirdNET confidence threshold 0.3 and 70% with BirdNET confidence threshold 0.8. These European values are comparable to the rates of false negatives gleanable from Fig 4 of Funosas et al. 2024 (Assessing the potential of BirdNET to infer European bird communities from large-scale ecoacoustic data. Ecological Indicators 164, 112146), and their implications are further discussed in the revised manuscript.

Table S4

This is a useful table and I suggest leaving this in and distribute as an Excel spreadsheet.

Reply: Thank you for the comment. We have now added it as a separate xlsx file.

Figure S5

This was unreadable for me and given that axes are not labeled, I can't comment on it.

Reply: We have cleaned up the figure and added explanation.

TextS5

Should this be TextS6?

I am not certain how these habitat/trait variables are used or are then associated with each species of bird (I assume that is what is being done here). As mentioned above, many landscape variables are simply

missing that I believe are strong predictors of temporal patterns of the soundscape and wonder why they are not considered.

Reply: All parts of the Supplementary material have now been renumbered for clarity, and all references from the main text to the supplements have been checked. In terms of missing landscape variables, we stress that there is, per definition, an infinite number of potential descriptors to consider. Importantly, our intent is *not* to pinpoint the exact environmental drivers of local variation in soundscapes among this set, as that would call for way more sites than here available. Instead, we use a hypothesis-driven approach: among a set of predefined metrics, we test for their added contribution beyond time of day and day of year. As key clarification, we point to our response to the Reviewer's comment on Figure 2 (above). Overall, we have now further clarified all parts of our analysis and inference, including our *a priori* choice of covariates considered.

Table S5

Should this be Table S7?

Experimentwise error rate should be adjusted in my opinion. Please consult a statistician.

Reply: Thank you for pointing this out. We have now corrected p-values due to multiple testing using Holm's method. In the revised table, we show both raw and adjusted values.

Table S6

Hmmm.....these are important results that are not mentioned (seems entirely new information) in the body of the main text. It is correct that BirdNET does not do well in tropical regions but this paragraph buried deep in a supplement raises a few red flags for me. Content like this should be placed in the body of the main text and placed in the context of limitations.

Reply: This section was originally included in the main text, but was relegated to the Supplements before our previous submission (as being a methodological consideration rather than a main result). Spurred by the Reviewer's concern, we have now reinserted it in the Discussion section of the main text.

Table S7

Useful table. The team name is not useful to readers. List co-author with their initials instead.

Reply: All relevant site information is now included as Sheet 3 in the supplementary xlsx file. The site names refer to the codes by which the data are organised and were retained as the links needed to connect site-level covariates to the raw data.

Figure S1 (Map)

Why not include this overlaid on a map of the continents? Use of ArcGIS would be helpful. No legend is provided. Sites names run over one another. Overall, a very unprofessional map.

Reply: The idea of this map was just to identify all recording sites by colour codes, to give full credit to the teams involved. We have now revised the figure for clarity and show the locations of the sites with continent borders on the map. At the same time, this map swerves as the legend for Supplementary Figures S2-S55.

Figure S1 (Multicolored Plots)

A set of $4 \times 27 = 108$ plots are not described at all and none have a y-axis labeled. There are no descriptions to these. Nor is there a legend unless the legend is the S1 Map color coded dots. Were these included since they are so colorful? Waste color toner and paper for me.

Reply: Here the idea is to show the model predictions for both site-specific models and the global model. For the rationale behind the models we refer the Reviewer to our response to the comment on Figure 2 (above) and to the revised Methods description. The colour is used for identifying the sites and their latitudes, using the same colours as introduced in the new map in Figure S1. After conferring with the Editor, we have broken the figure into 55 parts and added a clarifying legend for each plot. All axes now have labels.

Conclusions

The value of this paper is the data. It is highly unique and the authors are able to address a very unique set of questions. However, it is poorly written, analyses in several areas are flawed, the raison d'être is not well posed, and many of the figures are confusing. A solid paper will require some significant reanalysis and also robust, concise, precise and informative prose (a complete rewrite).

Reply: We are most grateful to the Reviewer for their helpful feedback, which has enabled us to reach a significant reanalysis and a complete rewrite of the paper.

Reviewer #2 (Remarks to the Author):

1. Key results: Please summarise what you consider to be the outstanding features of the work.

1.1. The study described provides the largest spatial and temporal survey and analysis of the drivers of soundscape patterns to date.

1.2. To the best of my knowledge, this is the first large scale study that attempts to factor out anthropogenic, geophonic and biophonic components and investigate differences in seasonal and diel patterns between urban and natural areas, and across latitude.

Reply: We thank the Reviewer for clearly identifying our main contributions, and have made every effort to further emphasize these main points.

2. Validity: Does the manuscript have flaws which should prohibit its publication? If so, please provide details.

- My key concerns are to ensure that conclusions are clearly substantiated. This is easily achieved through changes to the text rather than further analyses.

- There is also some need for greater precision. Main para 5. Text describes an analysis of bird species diversity, but some proxy of richness is analysed and reported.

Reply: We have now revisited every conclusion, aiming to pinpoint the exact evidence provided. In the text, we have clarified that the use of BirdNET will yield a proxy of species richness rather than true species richness. The value of this proxy, the inference drawn from it and the sources of errors involved have all been further discussed.

3. Originality and significance: If the conclusions are not original, please provide relevant references. On a more subjective note, do you feel that the results presented are of immediate interest to many people in your own discipline, and/or to people from several disciplines?

3.1. The results are of relevance to a range of areas of study:

3.1.1. The use of soundscape metrics is currently under debate in ecoacoustics; the global models created provide clear explanations for the importance of factoring latitudinal (a proxy for ecozone and acoustic community assemblage) into account.

3.1.2. The global patterns observed are significant findings for soundscape ecology and ecoacoustics and provide valuable contexts for national and regional studies.

3.1.3. The data provide valuable baseline information for future climate-focused phenological and avian migration studies

Reply: We thank the Reviewer for supporting the originality of our contribution.

4. Data & methodology: Please comment on the validity of the approach, quality of the data and quality of presentation. Please note that we expect our reviewers to review all data, including any extended data and supplementary information. Is the reporting of data and methodology sufficiently detailed and transparent to enable reproducing the results?

4.1. Method: Rationale for study design choices are not completely clear. In particular, what does the Nested design allow you to answer that Global and National do not? Tradeoffs between spatial acuity and coverage? One sentence is sufficient.

4.2. Method: Modelling. Can you provide more details on how the global periodic models were created.

Reply: The rationale for the overall design has now been published in a separate paper, to which we refer (Hardwick et al. 2024. LIFEPLAN: A worldwide biodiversity sampling design. PLoS ONE 19(12), e0313353). We have revised all figures to reveal the data structure in full detail, and added more explanation of how the global periodic models were constructed.

4.3. Method:

4.3.1. *spectral energy. Please state rationale for / assumptions behind frequency band choices.*

Reply: The choices for low, middle, and high bands are somewhat arbitrary. Below 1kHz there is usually quite much background noise and sounds of wind. Most animals and birds vocalize below 10kHz, whereas frequencies above 10kHz contain vocalisations by many insects like cicadas. To clarify our choices, we have now added justifications to the revised Methods section.

4.3.2. *Acoustic indices. Please state rationale for 300Hz cut off. If you are analysing anthropophony, is that not useful info – e.g. engine noise etc.*

Reply: The main reason to cut off very low frequencies is to get rid of any possibly non-zero DC component and low frequency disturbances. Although there can be antropophonic signals in very low frequencies, not all antropophonic sounds are there. This is also supported by our current results, where e.g. the NDSI value (which is expected to reflect the ratio between the acoustic energy in a frequency band associated with technophony to the energy in a frequency band associated with biophony) shows lower values at urban than natural sites (Figure 3B). Furthermore, we adopted a 300Hz cutoff frequency in the Butterworth highpass filter, used to dampen the lowest frequencies while not completely remove them (e.g. at 300Hz, the magnitude response is -3dB). These choices are consistent with e.g. the 500 Hz cutoff frequency used by Bradfer-Lawrence, T. et al. 2019 (Guidelines for the use of acoustic indices in environmental research. *Methods Ecol Evol.* 10: 1796–1807. <https://doi.org/10.1111/2041-210X.13254>). These considerations have now been added to the manuscript.

4.3.3. *If the choices of spectral energy contained assumptions central to your experimental aims, did you map settings of acoustic indices (BI, ADI and NDSI in particular) to these same values? Parameterising these for global models seems hard different species in different areas will have different frequency ranges.*

Reply: The choices for low, middle, and high bands were aimed at summarizing the acoustic energy with very few (here 3) components. The frequency bands were not designed to match to particular acoustic indices. However, e.g. for BI, the frequency band is roughly similar: our middle band 1-10kHz whereas in BI it is 2-8kHz.

4.3.4. *Acoustic event detection. Please state whether you validate the YAMNet outputs and how.*

Reply: We have not validated YAMNET outputs.

5. *Appropriate use of statistics and treatment of uncertainties: All error bars should be defined in the corresponding figure legends; please comment if that's not the case. Please include in your report a specific comment on the appropriateness of any statistical tests, and the accuracy of the description of any error bars and probability values.*

5.1. *Regression seems appropriate; p values and error bars are clearly labelled.*

Reply: We thank the Reviewer for checking the appropriateness of our statistical analyses.

6. *Conclusions: Do you find that the conclusions and data interpretation are robust, valid and reliable?*

6.1. *The key finding is presented as “We conclude that the global soundscape is formed from a highly predictable biophony, with increasing noise from poorly-predictable geophony and anthropophony”. You should be explicit about the basis of this claim. Are you inferring predictability from total R²? Bird species richness has a lower R² than speech or vehicle, and you state elsewhere that the soundscape metrics show relatively weak correlations with biodiversity. As you have not tuned the metrics to known local biophonic ranges, you need to spell out how you are confident that these metrics capture biophony? This can be rectified by elucidation in the results and/or discussion.*

Reply: Following the critique offered by Reviewer 1, we have now revisited and revised our conclusions.

6.2. *A small but important point: The final sentence of the Summary states “humans and other animals across the world are experiencing and communicating against a drastically different sound background than ever before.” But this is not a longitudinal study – afaics there are no data from which to infer a temporal change, it is just assumed. Consider recrafting closing statement around your data? E.g. something around anthropocentric influence on the rhythms of wider biological life?*

Reply: The relevant section has now been completely rewritten.

6.3. *Discussion opening sentence could also be more on point! ” We used systematic audio recordings around the globe to measure how the planet is buzzing with sound” – strong suggest to focus on actual key contribution e.g. “Through systematic world-wide acoustic survey we characterised the drivers of the global soundscape for the first time”*

Reply: The relevant section has now been completely rewritten (with the opening sentence suggested by the Reviewer gratefully adopted).

6.4. *As it is so central to main claim, please clarify what you mean by anthropophonic sound – vehicles and voices? E.g. How pronounced are differences, para. 1 “Such anthropophony showed either little rhythm in time, or a rhythm substantially different from that generated by animals and abiotic forces (Fig. 2; Supplementary Fig. S1).” – can you label S1 and be clear? Vehicle and speech show strong diel patterns ... OK, they are different to wider soundscape, but they have a diel pattern.*

Reply: The definition of anthropophony has now been clarified throughout the manuscript and the evidence for each statement made has been clarified.

6.5. *Similarly, you write “clearly invaded by anthropogenic sounds” – “invaded” seems a strange choice for urban green spaces? It is essential to clarify what you are referring to.*

Reply: The relevant section has now been completely rewritten.

Overall this is a good paper, but the final conclusions do not seem well justified. Comments line by line: “Overall, we find strong diurnal and seasonal rhythms in the global soundscape – overlaid with a strong human signature.”

- Humans are part of the global soundscape, aren't they? Do you mean “differing biophonic and anthropophonic signatures”?

Reply: The conclusions have been completely rewritten.

“ Anthropogenic sounds are ubiquitous and widely invade the soundscapes of urban green spaces”.

- If anthropogenic sounds includes human voice “invade” seems like a strange choice for urban green spaces?

Reply: The relevant section has been completely rewritten.

“Thus, humans and other animals worldwide are now experiencing and communicating against a drastically different sound background than ever before.”

- Forgive me if I am missing something, but as above, there are no longitudinal data here, how is this statement substantiated? Something about the impact of human activity on rhythms of the rest of life seems more justified.

Reply: The relevant section has been completely rewritten.

“How this change in the global soundscape will affect animal communication and human well-being is a territory only partly explored.”

- Fine.

7. Suggested improvements: Please list additional experiments or data that could help strengthening the work in a revision.

7.1. Main: para 2 & 3. Do you mean “diel” rather than “circadian” – the latter being endogenous?

7.1.1. Note this depends a little on whether soundscape is considered from as a subjective/ perceptual phenomena or objective. Consider qualifying and confirming perspective taken here? (see refs suggestion)

Reply: Throughout the manuscript “circadian” has been replaced by “diel”.

7.2. Main para 3. Consider referencing other disruptions to usual diel patterns such as light pollution?

Reply: This consideration has been added.

7.3. Main. Para 4. “How green spaces can conserve acoustic environments” feels like a strange construction. “the relationship between green spaces and emergent soundscape”?

Reply: The relevant section has been completely rewritten.

7.4. Throughout. Having defined soundscape as acoustic environment, suggest sticking to it throughout, rather than using “acoustic world”, “acoustic landscape” etc

Reply: We have now stuck to uniform terminology, but actually adopted “soundscapes” as the standard term used.

7.5. Throughout: check speech marks.

Reply: Checked.

7.6. Throughout: Pick one term for descriptors – e.g. “soundscape metrics” and use throughout. You use acoustic indices, soundscape indices, metrics etc

Reply: This comment echoes the concern of Reviewer 1. In response, we have chosen to use “acoustic indices” throughout the paper.

7.7. Methods/ design/ temporal sampling: Can you clarify whether absolute hour or time relative to sunrise were used? If the former, were results adjusted relative to dawn? Else how were diel patterns modelled through the year?

Reply: This comment also echoes the concern of Reviewer 1. As a response, we have now repeated all analyses using the time that reflects local sun time based on the latitude, longitude, and time of the year during the recording. How we did this is explained in the Materials and Methods section.

7.8. Figure 1. B) is pretty hard to read, could this be represented more clearly?

Reply: Figure 1 has been thoroughly revised.

7.9. Fig1 C. Small, point, but would “metric” be a better column header than “index”? spectral energy is not really an index and you use metric in the text.

Reply: As far as we understand, index is synonymous with metric. For consistency, we have stuck to “[acoustic] index” throughout the text. (See above.)

7.10. Fig 2. Can you clarify whether latitude in key refers to latitudinal range and how they are grouped for ease of cross-reference with S1

Reply: In the previous Fig 2 (new Fig 3), latitude does not represent a range but instead a single value. Based on the statistical model that we have fitted we can make predictions for any given latitude. The figure illustrates such predictions of the statistical model by showing them for four different values of latitude.

7.11. S1. Please clarify units on the y-axis. It is not clear what negative frequency means on the spectral metrics

Reply: We have now added y-axis legends. For spectral energy, the values are logarithmic, which is why they can be negative. Units are decibels with a reference value of 1. These are magnitude values, not frequencies.

7.12. Fig 1 B. Do you have anything to add on what “animal” sounds were dominated by? Given increase in urban areas, was this dogs and foxes, these are very closely related to human-influence.

Reply: YAMNET splits animal sounds into three subcategories: 1) Wild animals; 2) Livestock, farm animals, working animals; 3) Domestic animals, pets. As we now mention in the main text and show evidence for in the Supporting Information, the vast majority of our detections belong to the group of “wild animals”.

7.13. Please label subplots in S1 for ease of cross-ref.

Reply: Supplement S1 has been extensively revised and subplots labelled. After conferring with the Editor, we have broken the figure into 55 parts and added a clarifying legend for each plot.

8. References: Does this manuscript reference previous literature appropriately? If not, what references should be included or excluded?

8.1. In intro paragraph of Main, consider including recent debate over validity of foundational evolutionary and ecological theories of ecoacoustics (e.g. Alcocer, I., Lima, H., Sugai, L.S.M. and Llusia, D., 2022. Acoustic indices as proxies for biodiversity: a meta-analysis. *Biological Reviews*, 97(6), pp.2209-2236.)

8.2. Main para 2. Consider including more recent work qualifying soundscape (see 7.1.1)

8.2.1. Farina, A., Eldridge, A. and Li, P., 2021. Ecoacoustics and multispecies semiosis: Naming, semantics, semiotic characteristics, and competencies. *Biosemiotics*, 14(1), pp.141-165.

8.2.2. Grinfeder, E., Lorenzi, C., Hauptert, S. and Sueur, J., 2022. What do we mean by “soundscape”? A functional description. *Frontiers in Ecology and Evolution*, 10, p.894232.

8.3. In methods – cite ref for Global Footprint Index

Reply: We are grateful to the Reviewer for suggesting these references, which have been added to the text.

9. Clarity and context: Is the abstract clear, accessible? Are abstract, introduction and conclusions appropriate?

9.1. Main para 5. Needs a concise introduction to acoustic survey to link from previous paragraph. E.g. “large scale acoustic surveys can now be undertaken with relative ease, but generate vast data sets. Analyses requires automation and current research adopts soundscape metrics that provide statistical summaries of “ Etc.)

Reply: The relevant section has been rewritten, with input from both Reviewers adopted as offered.

9.2. Lellouch is not the best ref. here. Consider Sueur, J. and Farina, A., 2015. Ecoacoustics: the ecological investigation and interpretation of environmental sound. Biosemiotics, 8, pp.493-502.

Reply: The reference has been changed as suggested.

9.3. Main. Final question is not clear – “what features do these apply to” – do you mean: in which metrics are rhythms observable?

Reply: The relevant section has been completely rewritten.

9.4. Discussion. Related to 6.1: You suggest that soundscape metrics do not correlate well with biodiversity, but also acknowledge the BIRDNET may not be giving accurate results for species outside US and Europe. If these metrics are not reflecting biodiversity, do you still assume they capture biophonic variation? Please elucidate a little further in discussion as it is so core to conclusions.

Reply: The relevant section has been completely rewritten, and the implications of the caveat identified by the Reviewer have been discussed.

In addition to the queries and concerns listed in the review report (above), Reviewer 2 had offered a series of sharp-eyed and constructive suggestions for direct edits and clarifications in the manuscript text proper. All of these comments have now been introduced in the revised version of the manuscript.

RESPONSES TO REMAINING REVIEWERS' COMMENTS

Below, we will respond to the remaining Reviewer comments point-by-point. As this is the second revision of the paper, the letter features up to four different sets of text: original comments (from the first round); our responses to these comments; the Reviewers' responses to these responses; and our new responses to the latest set of comments. For clarity, our current responses are preceded by "**New response:**" and **formatted in bold**.

REVIEWER #1

GENERAL COMMENTS

First of all, I would like to thank the authors for a detailed response to the two reviewers (I am reviewer #1) and especially for taking the care to do a significant rewrite. Overall, I find this version to be much more readable, clear and succinct; which is what I would expect from a paper to be eventually published in top tier publication like *Nature Ecology and Evolution*.

Second, I have seen improvements in (1) clarity of the description of model that was used to generate the massive number of plots that describe the rhythms of the soundscapes; (2) all figures; (3) conclusions (why should we all care); and (4) discussion which was not strong in the first draft.

Third, I thank the authors for providing more clear statements of contributions to the entire paper.

New response: We are most pleased that the Reviewer, Professor Pijanowski, is happy with the general overhaul undertaken. We remain grateful for the massive effort that he has invested in helping us improve our contribution.

A few areas that I think still need to be improved. These include:

CORE MANUSCRIPT

Summary

I'd replace "humanity's" with "society's"..

New response: Changed as suggested.

I am a bit confused by the use of the term "poorly-predictable" geophony (does that mean it is not clearly a temporal pattern....and if so, I would find that odd since wind is a large broadband contributor to geophony and this varies fairly predictably by time of day and season).

Also, a bit confused by the "less predictable background of sound" comment.

Overall, the summary should stand on its own and right now it does not, especially the last sentence. It is close but still needs some work.

New response: In re-reading our text, we noted that the terms catching the Reviewer's eye were not strictly needed. Thus, we have simply deleted them. The last two sentences (ll. 290-292) now read: "We conclude that the global soundscape is formed from a highly predictable rhythm in biophony, with added noise from geophony and anthropophony. At urban sites, animals

experience an increasingly noisy background of sound, causing challenges to efficient communication.” These sentences are directly supported by the evidence cited above them, and the Abstract thus stands on its own.

Main

Although I see this as a first-time use, I am not fond of the terms “high” and “low” to describe data values. More precise terms of “greater” or “lesser” are more proper (I understand you are visualizing data points on a graph but these are not precise terms). Note: use of the term “higher” is fine for describing spectral frequencies, however.

New response: We have now searched the text for all instances of “high” and “low” and amended them. For “high” we have implemented fourteen replacements with “greater”, one deletion of “high” as being not strictly needed, and a few retentions in constructs like “highest prevalence”, “higher values”, “highly predictable”, “high latitudes”. The word “low” was only used in reference to sound frequencies and has thus been retained.

Somewhere in lines 350-360 I would expect a detailed description of the expected diel patterns and then the rationale. For example, I would expect biophonies to peak just starting before sunrise, fall during the day, then rise again prior to sunset. In just about all of my analyses of acoustic indices, nocturnal acoustic activity (diversity and complexity) is greater at night during the day. Most of your sun time plots show this clearly although there are a few outliers (e.g., high, northern latitudes). Indeed, these descriptions are at the heart of the paper and so I would hope that these descriptions can move to a more prominent place in the paper.

New response: We fully agree, but since the comparison to Prof. Pijanowski’s work relates to Discussion rather than current results, we have opted for inserting them in the 3rd § of the Discussion. The new sentence reads: “The rhythms here detected at a global scale are largely consistent with previous work on local and regional soundscapes¹, suggesting that the local peak in biophonies start before sunrise, fall during the day, then rise again prior to sunset. Whether nocturnal acoustic activity exceeds activity during the day depended on the index used and the latitude (Fig. 3; Supplementary Figs. S2-S55).”

In lines 361-368, the paragraph mixes the description of diel patterns with seasonal. I’d devote one paragraph to each. In my experience, the maximum diversity of biophony in the midlatitudes is around late summer, which is when all acoustically active animals occur (birds, amphibians and insects).

New response: We have now cut the § in two, as recommended by the Reviewer.

Acoustically active insects such as cicadas, katydids and crickets emerge late in the year (spring is the peak for birds obviously). I did see some discussion of these patterns in lines 431-445 but it lacked detailed sequencing as I have described here. Also note that insect sounds often exist outside the typically 2-6 kHz of most passerine bird calls.

New response: Here, we have refrained from adding further details. This choice is motivated by two key considerations: first, we are seriously restricted by limits on both manuscript length and references, and second, our current data precludes the detailed separation of more specific animal groups. Since we have included ample reference to literature detailing general rhythms and patterns in specific taxa, we have refrained from reviewing this literature in further detail.

Line 363 “Diel patterns were likewise highly pronounced” is not clear as a standalone sentence. Do you mean sounds of wind?

New response: This sentence signals a turn from seasonal to diel patterns. Thus, it does not point backwards to wind, but forwards to a new topic. To make this distinction evident, we have now inserted a paragraph break (following the above recommendation of the Reviewer).

Lines 318-341. As a soundscape ecologist and biogeographer, I was a bit surprised that the authors did not mention one of the most significant global bioclimatic patterns and how the seasonal soundscape is affected. More specifically, there is a large amount of rainfall at the equator, then the effects of the ITCZ (two rainy seasons at low latitude and one rainy season at high latitudes) kick in slightly above and below the equator, the desert belt occurs at 30 N and S with these systems being impacted by shortterm seasonal monsoon events, the mid-latitudes that have four seasons and the high latitudes dominated by cold and windy climates. Are these bioclimatic patterns (mostly created by Hadley and Ferrell Cell circulation patterns) reflected in the soundscape data seasonally? It is hard for me to tell.

New response: It is hard for us, too, to extract such information from our results. We have thus refrained from introducing further complexity in an already long text.

Lines 381-410. This is a good section of the results, clearly written and well organized.

New response: Thank you.

Discussion

I find the discussion MUCH improved, well written and easy to follow. I do not have any detailed suggestions for improving this section.

New response: Thank you.

Methods

These methods are clear and well written and thanks for a complete rewrite of this section.

New response: Thank you.

Figure Legends

For Figure 4, in the legend ad what gm, hfi, etc. mean (this is easy) as these are in the figure.

New response: We have added explanations of these in the figure legend.

Figures

Figure 1 – good

New response: Thank you.

Figure 2 – add y-axis values and what black and green mean (urban and natural I assume)

New response: The elements asked for have been added, latitude values on y-axis and explanation of black and green to the Figure caption.

Figure 3 – I would have added 1-3 more classic acoustic indices (ADI, BIO or ACI) if you could find the space.

New response: Since the panels asked for appear in the supplements, and since the current illustration already calls for substantial page space, we have refrained from adding further indices to this very Figure.

Figure 4 – add second y-axis which is percent variance explained, part A legend blocks y-axis label in B. Is green the right color here for the bars in B?

New response: The y-axis already shows the proportion of variance explained (1 = 100%). The Figure has been cleaned up to avoid blocking of y-axis labels in B. The meaning of green in panel B has been clarified in the text.

Figure 5 – is this plot correct? This shows species richness greatest at high, northern latitudes (blue)?

New response: The figure has been checked and is correct. Although some of the highest species richness points are indeed blue, the figure does not actually show species richness being generally greatest at high, northern latitudes – since if this was the case, the blue points would dominate in the upper right corner. But because the results are based on BirdNET detections, it misses species detections in southern sites (this feature of BirdNET software has been addressed in the main text). The main point in this figure is to compare species richness between urban green and natural sites.

Although BirdNET species list includes more northern species and does not cover all southern species, the comparison between urban green and natural sites are done within a site pair.

SUPPLEMENTALS

First, although I do appreciate the lengthy descriptions for the figures in the supplement, they are essentially a cut and paste effort. The plots are rich in information and in some cases I do wonder how well the authors have described these temporal trends in the core of the manuscript. In addition, the supplement contains several sets of plots and tables. Thus, as a person who is an expert in this field, I would tend to do deep dives on supplemental material like this. I'd like to suggest a few ideas to help researchers like me. This would include a high-level description of the sets of figures. I think this is easy, just add a narrative ahead of the table of contents that describes the sets of figures. I would expect a supplemental narrative of around 1-2 pages. Again, it is mostly to navigate through a massive number of figures. Perhaps consider a break/title page between the key sections (e.g., for time that is clock hour vs. sun hour).

New response: We greatly appreciate the Reviewer's efforts in helping us make the Supplements accessible to the reader. Given how much information they contain – and given how much effort we have invested in generating the nearly 70-page supplements – we naturally want to make full use of them. As proposed by the Reviewer, we have now inserted a high-level description of the sets of figures ahead of the table of contents that describes the sets of figures. We have also added page breaks and headings between key sections.

Second, I wish there was something in this paper that does a more analytical assessment of whether the temporal patterns follow expectations of the authors or not. Each acoustic index seems to behave differently. Does that mean that there is more to the trends of acoustical patterns in the soundscapes other than, viz., dawn and dusk chorusing for the biological sound sources? Indeed, when you examine the correlation plot (Figure S59), many are not correlated (e.g., Wind and SpecLow) while others are (e.g., H and ADI). What specifically can one glean from these patterns?

New response: This is indeed a key challenge, given the number of indices proposed and their different information content. In our contribution, we have tried to make higher-level sense of this variation by i) focusing the main text on differences between the general categories of biophony, geophony and anthrophony, and ii) pointing to the differences in information content AND the (greatly variable)

correlation structure among indices, highlighting that they will be complementary to each other (which is a point earlier stressed by the Reviewer). In doing so, we feel strongly that embarking on a multidimensional analysis of individual metrics (either through joint modelling or a detailed inspection metric-by-metric) would constitute a both monumental and complex exercise way beyond our current intents and well beyond the concise format sought by nature Ecology and Evolution.

Third, there appear to be figures in the paper that are also in the supplement (e.g., S57 and S58) and would recommend pulling those from the supplement and there are some (e.g., S27 and S28) duplicates (there might be others).

New response: We have now carefully checked our material for overlap between the main text and the supplements. For Fig. S57 and S58, we have further clarified that these figures are based on a definition of time different from that used in the main text: In Figs. 3 and 4 of the main text, we show predictions for models using absolute hour, in Fig. S57 and S58, we show predictions from models using time relative to sun rise and sun set. The replication of otherwise identical figures is specifically aimed at evaluating how much alternative definition of time affects our inference (finding that it does not). For S27 and S28, the difference between them is that one shows species richness and the other abundance.

Fourth, the titles for many figures here have “th .8” which looks odd to me.

New response: the “th .8” refers to the confidence threshold used in scoring the metrics. This is explained in the figure legend.

Finally, the most amount of work done by the authors to address one of my concerns is described in detail in Fig S60 (p.62 of the supplemental). This is great work by the ornithologist. However, the way that the data are presented are not clear. I would like to suggest a few things that would help introduce more rigorous statistics. First, the scores should be sorted into four bins: true positive, false positive, true negatives (which might not be easy) and false negatives (very important statistic). Can you calculate specificity and sensitivity? These values are interpretable and useful. Second, some of these bins may also have sub-bin categories. For example, I find that BirdNET gets confused if two birds call at the same time and so I get a false negative (does not detect bird 1) and a true positive (detects bird 2 correctly). I also often get double false negatives (both birds are not detected). Can these be reported? Adding more rigor to these CNN-based tools I think is warranted in global biodiversity assessments and patterns. I'm hoping that the data are ready at hand to do this.

New response: We have now calculated sensitivity and specificity for two cases 1) one bird species present and 2) multiple bird species present. As expected by reviewer, the case with multiple bird species is more difficult for BirdNET. The results are included in Supplement. We would also like to point out that we have included the reference to the following paper in our Supplement that might be of interest: Funosas, D. et al. Assessing the potential of BirdNET to infer European bird communities from large-scale ecoacoustic data. *Ecol Indic* 164, 112146 (2024).

Reviewer #1 (Remarks to the Author): (with Reviewer #1 Responses)

The authors of the paper “More predictable rhythms in the natural than anthropogenic global soundscape” conduct an analysis of over 1.4 million audio recordings of the soundscape from 139 sites located on six continents. Recordings for each site were in a pairwise design (urban and peri-urban settings. One-minute recordings were analyzed in the context of the proposed “rhythms of nature” conceptual framework of Pijanowski et al. (2011) that describe the predictable diel and seasonal patterns of biophonies that should exist around the world. The authors use a set of standard acoustic indices (ADI, ACI, etc.), measures of sound sources that are extracted from an acoustic event detector and also apply the widely used CNN-based

BirdNET tool to detect bird species. They find that for the most part, the rhythms of nature are indeed predictable in key time periods and across latitudinal gradients. Another key finding is that the human noise footprint is interfering with these natural rhythms which are key functioning features of all ecosystems. They conduct a set of spatial-temporal analyses of acoustic indices, sound source event metrics, as well as the species' habitat features. I do like the topic of the paper and I think many in ecology not familiar with the concept of the soundscape will be as well.

However, there are numerous shortcomings of the paper. I spent three days reviewing this hoping to provide some insights for significant revisions and so this represents a major time commitment to make sure that if this moves toward publication, a lot of the errors and poor writing can be corrected. I hope the editors and authors find these to be helpful.

Reply: We are most grateful for the intensive effort invested by the Reviewer in helping us achieve a worthwhile contribution. In response, we have made every effort to make full use of each suggestion offered. To pinpoint the changes made, we will address each comment in turn while highlighting the changes made.

Reviewer #1 Response. Thank you. It is greatly improved manuscript.

New response: Thank you.

First, I found the paper not to be well written; in fact, compared to most Nature/Science/Ecology Letter/PNAS papers I read, this paper lacks the high level of density, clarity, robustness and precision of the prose I would expect from these articles. One of my best illustrations for this is the sentence in line 465 "No matter where we are and when we listen, there is sound". This is blatantly obvious to all of us in this field and does not address any of the research questions. Another is the opening of the Discussion "...how the planet is buzzing with sound". These rather "folksy comments" provide no factual content and are not very scientifically useful (actually, even a bit misleading since this paper is an analysis of the patterns of the presence of sound in the environment and not how people perceive it). Overall, the language and interpretation throughout the paper is "loose". There is what I would call a lot of hand waving without getting to robust interpretations of the analysis. The core paper is easy to follow although the supplement got difficult to navigate at times (a supplement to a supplement was confusing).

There are MANY terms (spectral, amplitude) not used very precisely either. I would highly suggest consulting the glossary of terms that are in Pijanowski and Rivas (2023; Encyclopedia of Biodiversity, Elsevier, Soundscapes and Vibrosapes, Chapter 134) for the most commonly used terms in soundscape ecology and how they should be used.

Reply: We have now rewritten the full text, aiming for a more stringent expression. In doing so, we have made every effort to achieve both precision and accuracy in our use of the terminology (drawing on the source offered by the Reviewer) and specifying the interpretation of the analysis. The structure of the supplements has been revised.

Reviewer #1 Response. It is greatly improved manuscript.

New response: We are most grateful to the Reviewer, Professor Pijanowski, for helping us improve our contribution.

Let me state up front that this is a paper worthy of eventually being published in Nature Ecology and Evolution. The two main "take aways", that the soundscape is a predictable feature of ecosystems and that human noise is potentially affecting these rhythms is an important message to make to large communities of scholars and natural resource managers. I will for one cite it a lot once it has gone through some significant

revisions. In my opinion, there is a lot more work that is needed to get the paper to a publishable form. There are areas that are quite vague and specifics of analyses and calculations are simply not described. A large body of literature (on urban soundscapes such as Lex Brown, Jian Kang and Östen Axelsson and biodiversity assessments a la conservation biology work of say, Zuzana Burivalova) are missing and the take home points are not made all that strongly. In general, there are large bodies of literature missing from the paper that would make this a very informative paper across basic and applied ecology and evolution.

Reply: We are most grateful to the Reviewer for emphasizing the value of our contribution. In response, we have sharpened the take-home message, included substantial literature on urban soundscapes and added references to biodiversity assessments. The description of the analysis has been thoroughly revamped. In adding pointers to the wider literature, we must still stress that our hands are partly tied. Nature Ecology and Evolution recommends a maximum of 50 references, for which reason our original contribution included 41 references. The current count is 63, of which we feel all are needed.

Reviewer #1 Response. One reference by Pijanowski et al. is a duplicate so you have 62.

New response: We are very grateful to the Reviewer for noticing this error in our reference library. We have removed the duplicate reference and corrected the reference numbers accordingly.

The rest of my comments, which address the more general ones listed above, are summarized here:

Manuscript (i.e., Main)

Lines 329-337. It should be made clear here if the authors truly mean anthropophony or technophony as their major sound source category. The latter was proposed by Gage after the Pijanowski et al., (2011) Bioscience paper to describe sounds sources that do not include human voices. The list of examples seems to suggest technophony but the analysis includes human voices (some of us consider human voices to be part of the biophony since we are indeed animals). If we consider human voices, to what extent to these contribute toward the disruption of the natural rhythms (perhaps some of their multivariate analyses might point to that but I suspect that this is minor). My opinion is that the authors should focus on technophony since that is the “noise” component that is being assessed with most of the analyses and aligns with the paper’s objectives.

Reply: We have now tried to make a clearer distinction between different sound source categories. Indeed, one of the main objectives of the manuscript is to compare natural and urban environments using multiple acoustic features (i.e., the ones listed in Figure 1). Among the features resolved by our analysis, human speech is indeed treated a separate category (see Figures 1, 3, and 4 and Supplementary Figures). Thus, the Reviewers concern relates to what wider category the individual sources are subsequently grouped into. To avoid any unclarities, we have now explained what exact choices were made, how categories were formed and what they consequently consist of. In doing so, we have made sure to specify our conclusions with respect to what sources they relate to.

Reviewer #1 Response. This was successfully completed to my satisfaction.

New response: Excellent.

Lines 338-347. Paragraph could be stronger and include more specifics. It is known, for example, that birds and amphibians dominate the dawn chorus and that the dusk chorus is dominated by insects. It should also be mentioned that the timing of, and the length of, both the dawn and dusk chorus differs by latitude and time of the year.

Reply: We have now strengthened and elaborated on the paragraph in question. We have also emphasized that our study adds the first fully standardized quantification of the global and seasonal patterns proposed by the Reviewer – a contribution which has now been highlighted.

Reviewer #1 Response. Well done.

New response: Thank you.

Lines 348-355. I find this paragraph to be far too general and some of the noise patterns are indeed diel. First, the sounds of highways are likely to be most loud prior to the start of the business/school day and then again most loud during the end of the business/school day. So, I'm not sure I agree with the authors. Second, each ecosystem has a predictable soundscape phenology that is related to the phenologies of major sonically active animals (here in the midlatitudes, amphibians emerge early spring, then come breeding and migratory birds, and then the insects). These major animal sonic groupings also have different acoustic patterns (pulsating, melodic). There is a lot of literature to support these kinds of patterns and they should be clearly described here. This is also the paragraph where a lot of the urban soundscapes research should be cited (Brown, Kang, Axelsson). See Chapter 13 in Pijanowski's book (2024) for a thorough review of the work in urban soundscapes.

Reply: We fully agree and have now added further coverage of the considerations offered by the Reviewer. In doing so, we have greatly benefitted from the recent book by Pijanowski.

Reviewer #1 Response. Pleased that helped.

New response: Thank you.

Lines 356-363. There is a lot of work done on the US NPS soundscapes as a natural resource (see Dumyahn and Pijanowski, Landscape Ecology 2011, Managing soundscapes as common pool resources) and also the work of Kurt Fristrup and Peter Newman at Penn State University). I'm not certain this is a useful paragraph as written.

Reply: In revising the manuscript, we have omitted the original paragraph and added further consideration of soundscapes as natural resources.

Reviewer #1 Response. Well done.

New response: Thank you.

Lines 364-378. This paragraph does not precisely use the terms properly and so I was a bit confused with what they were referring to by "metrics", "features", "descriptors", and "representations" which are referenced in citation #13. Figure 1 is referenced and that figure (which needs A LOT of work) does not provide the reader with what these terms refer to. The authors need to clarify the use of these and perhaps provide a diagram that illustrates how these differ, especially as they relate to acoustic indices. I am also confused by the use of the term "voice abundance". Some researchers have used the term call rates. Also, the field has not settled on how to use the number of bird species calls in any analysis. What you are specifically doing here, I think, is to use BirdNET to create a species richness value per recording and then per site.

Reply: The different terms were used to avoid the very manifold repetition of a single term such as "metrics". Thus, they are synonyms of each other. To avoid any confusion, we have now chosen a single term to be used throughout the text. In Figure 1 there is short description of each measure and for full description (regarding the soundscape indices) we have referenced the original papers. For BirdNET outputs, we have calculated two measures, one of which is species richness and the other call rates.

These aspects have now been clearly defined in the text.

Reviewer #1 Response. Well done.

New response: Thank you.

Lines 379-383. I am almost certain I know what you mean here in your statement of the paper objectives. But these are clumsy at best. Could I suggest: “how pronounced and predictable are global biophonies across diel and seasonal patterns?” and “does noise produced by humans affect these natural rhythms?” that is then followed by “We test examine each of these questions using a global soundscape database and a paired urban-rural experimental design”.

Reply: We apologise for the unclear wording and have changed the wording of the paragraph.

Reviewer #1 Response. Much better. Well done.

New response: Thank you.

Line 373. I'd exclude human voices from the list of anthropophony (by the way, anthropophony and anthrophony have been determined by a scholar in Latin-Greek to be synonymous).

Reply: Since we have calculated different measures separately, the grouping of them can be done in many ways. It is then a matter of choice whether to include human voices in the category of anthropophonic sounds. Importantly, this will not change our analysis, as human speech is included as a separate category, and the exact patterns detected in this category are shown in Figs. S2-S55. In principle, we are also hesitant to remove human-generated speech from the category of humangenerated sounds (anthrophony). To avoid any ambiguities, we have now clarified the description of what exact original metrics are grouped in what wider categories, and discussed how human voices relate to the class of antropophony.

Reviewer #1 Response. Understood and acceptable answer.

New response: Thank you.

Lines 386-394. This is the second use of the term circadian rhythms. Authors have used diurnal through to now. Other researchers use diel which is probably the best term to use of the three. I'd stay consistent with the terminology throughout the paper. As I am a bit confused by the term “descriptors”, it was difficult for me to follow this first paragraph in Results.

Reply: Thank you for pointing this out. We have changed the text to use more coherent terminology.

Reviewer #1 Response. Thank you.

New response: Our pleasure.

Lines 395-403. This is a very difficult paragraph to embrace since there is not much detail in what the authors did. The hourly patterns I think are a challenge to compare across latitudinal gradients since the timing of dawn varies by date and latitude. Using just the local clock time is misleading. Instead, the authors should have indexed the time of the recording to be something of a standard “time since civil sunrise”. Indeed, at 60° during the summer solstice, there is no period of complete darkness. If I had one serious criticism, is that they (I think) used local time across all sites which does not reflect the dawn. A second comment is that the dawn chorus at different locations and time of the year has different lengths. My extensive time at locations of the equator have taught me that the dawn and dusk periods are very short there nearly all year round.

Reply: We were thinking about the varying daylight time when doing the first draft of the manuscript but then decided to use only absolute local time. However, based on the Reviewer's comment we have now repeated all analyses using also the time that reflects the available sunlight based on the latitude, longitude, and time of the year during the recording. How we did this is explained in the Materials and Methods section.

Reviewer #1 Response. I know this was probably a lot of work but I think was essential for this to better reflect that acoustic communication aligns with daylight circadian rhythms.

New response: This validation was important, as it addressed a common question emerging in the readers' minds.

Lines 423. I do not think that the use of the term "silence" is proper here. Silence refers to the lack of any sound. That only occurs in vacuums or anechoic chambers. Quiet is a better term and it needs to be described as very low amplitude of sound waves that are not detectable using field equipment.

Reply: The word Silence refers to the name of an explicit AudioSet class used by the YAMNet classifier. To clarify its use, we have defined the term when introduced and used the word "quiet" in the main text.

Reviewer #1 Response. Understood and a good response.

New response: Thank you.

Lines 434-496. What the authors have found – that is, that nearby natural areas and not different than urban areas in terms of call rates, is not that unusual. There are MANY papers that have described how birds are now calling at higher pitches, are louder or are more active at night in response to the heightened ambient sounds from humans. Can the authors place their research in this large body of literature (e.g., see the work that describes the Lombard Effect)?

Reply: We have now discussed (with references) how birds change their vocalization in urban environments, and how this may relate to the pattern found. Having said that, our data do not provide direct evidence of changes over time (since they are from 1-2 years only) and do not compare pitches or activities within species between urban and natural sites. Thus, the section puts our observations in a context but adds no direct evidence.

Reviewer #1 Response. Most scholars in bioacoustics know these patterns and would expect some interpretations around these known patterns.

New response: We take it that the issue was addressed to the Reviewer's satisfaction.

Lines 454-560. The entire discussion needs to be rewritten. First, the authors are introducing (using a prominent subheading) "what does the word sound like?". This was not one of the key research questions and I am not sure they have answered it. The section in lines 465-475 do not address this question either. The presentation is overly general and very vague (handy waving of correlation patterns). The second subheading does not seem to be fully answered either (and this is what the authors state was one of the research questions). I'm confused by several of the sentences especially the one that contains (line 502) "traditional community data...". How do the recordings and the analysis "pinpoint the agents that are making sounds...?". I also want to point out that the fact that the acoustic indices do not always strongly correlate is a GOOD thing. They each are measuring a different aspect of the soundscape. ADI, for example, measures the diversity of total sound amplitude across audible frequency bands split into 1 kHz bands. ACI measures the amount of modulation in time and frequency. NDSI measures the relative spectral power of sound in the frequency bands associated with biological sounds versus those produced by technophony. The

role of the soundscape ecologist should be to decipher the complexities of the soundscape using these diverse tools.

Reply: We have now rewritten the entire discussion, taking the valuable pointers of the Reviewer *ad notam*.

Reviewer #1 Response. It reads well and this was worth the effort and time.

New response: We agree that this was time well invested.

Lines 510-520. One of the major issues and discussions right now is how to address the accuracy of BirdNET output. Interestingly, here the authors use the proper assessment descriptions (quantifying false negatives especially since BirdNET seems to have high values of these based on our assessments as well).

First, I would not trust any output with confidence scores below 0.50 so I am not certain why they would consider anything less as these are neural net tools that learn threshold into binary output data (presence and absence). Second, one needs to develop methods that assess false negatives (a bird called and it was not detected at all). Third, call activity levels is driven by taxonomic considerations (nocturnal nightjars call repeatedly whereas many higher-level passerines call infrequently and are not that repetitive in their call structure). I'm not sure I would use call rates in the analysis until the research community can determine how to use BirdNET data that contains a lot of this information.

Reply: We calculated two measures from BirdNET outputs. The first was the number of species detected (detected species richness) and the other was call abundance (i.e., how many times the detection exceeded the given threshold). The latter one is now referred to as “total call rate”, following the suggestion by the Reviewer. We find it interesting to show results of both approaches. We are not claiming that one is better than other, but aim to provide the reader with a clear view of results from both approaches. We also believe that this is an important step towards helping the research community determine how to use data from BirdNET and from other, similar and quickly-accumulating tools.

With respect to detection thresholds: there is definitely a trade-off between precision and recall. In Supplement 1 we show the results using two different thresholds 0.8 and 0.3. Indeed, the value 0.3 is very low and the results contain many false positives. However, the interesting point is that the shapes of the activity patterns observed using these two different thresholds look qualitatively similar. We are not claiming that BirdNET gives exactly true species richness – rather, it is a proxy of it. However, as BirdNET is currently the state-of-the-art method for automatically identifying bird species, this is what the current methodology is capable to perform. Importantly, what we observe *despite* the possible shortcomings of BirdNET is that its output is useful to reveal temporal patterns. These patterns emerge even against the background noise generated by the false positives and false negatives unavoidably generated by automated detection methods.

Reviewer #1 Response. This was a huge amount of work but necessary to lend credibility to the BirdNET results. Adds a lot of value to what researchers need to do to include results of these tools, especially at a global scale.

New response: Thank you.

Lines 554-560. This is the weakest section of the paper. The authors need to think more deeply about the implications to their results. Why should ecologists care? I can think of dozens of reasons that rise to the very top of highly crucial issues but these are not included anywhere in this paper and certainly not here. I think addressing the work in the context of the biodiversity crisis, climate change, conservation management, urban planning and even human-nature interactions are all extremely important. Indeed, this

gap is evident in the introduction and abstract. Serious thought needs to be put into this part of the paper to make it worthy of publication in Nature Ecology and Evolution.

Reply: Thank you for the comment. We have now rewritten the Conclusions to focus squarely on the implications brought forth by the Reviewer.

Reviewer #1 Response. Reads well and hits the key elements of a conclusions.

New response: Excellent, thank you!

Materials and Methods

Lines 565-571. Paragraph needs some editing.

Reply: The paragraph has been edited for clarity.

Reviewer #1 Response. Agreed.

New response: Thank you.

Lines 572-595. A figure here that depicts the experimental designs would be helpful.

Reply: Fortunately, the exact design and protocols used have now been published in a recent paper (Hardwick et al. 2024. LIFEPLAN: A worldwide biodiversity sampling design. PloS ONE 19(12), e0313353). To avoid any confusion, we now refer to this paper for the general design – whereas all the specifics of the material used in the current paper are shown in the revised figures.

Reviewer #1 Response. Thank you for that additional information.

New response: Our pleasure.

Lines 618-623. FFT models are not described. What window size and type were used to process the wav file for use in the analysis? I assume 512 and Hanning?

Reply: 1024-point FFT was computed every 10ms and a Hann window was used. We have now added this to the text.

Reviewer #1 Response. Thank you for that additional information as these are important considerations for how acoustic indices are calculated with non-SongMeter devices.

New response: Thank you.

Lines 625-631. Many of the acoustic indices used were developed using Wildlife Acoustics SM2s, and in some cases, SM4s. Some of the acoustic indices, such as ADI and ACI, are also sensitive to parameter settings and microphone sensitivities. For example, ADI was originally designed to be used for SM2s and with preamps set to the factory settings. This means that the “noise floor” of -50dBFS was ideally configured for that model. Audiomoths use an entirely different set of microphone technologies and there are a host of preamp settings. So, I am weary of folks that use default settings on these acoustic index tools. We are also finding that ACI is sensitive to FFT window sizes (makes sense when you think about about) but also the Nyquist frequency (which might make sense) for certain FFT window sizes (large ones). THUS, it is extremely important to (1) make sure the acoustic indices are given values that make sense (ADI does not look right to me) and (2) that a sensitivity analysis of key parameters be undertaken to make sure these are effective parameters to use). Finally, a high pass band filter applied AFTER the recordings are created does not make sense to me. I sometimes like to use the 250 Hz HPF on the device as this makes the microphone more sensitive in the biologically active areas. Using a HPF will affect ADI and NDSI, although, if these are applied

across all recordings the same, then you should be OK. In short, the use of acoustic indices is somewhat careless.

Reply: Thank you for this point. Indeed, we used ADI with default values where the noise floor was set to -50dB. We have now recalculated ADI with different values and decided to use the results for the noise floor set to -26dB. We have also experimented with multiple window lengths for ACI. Scatterplots of ACI results with different parameters did not suggest any particular value to be the best. Therefore, we decided to use the default value a 512-point window length also in the revision. For other soundscape indices, we feel that the previous values were uncompromised as they were.

Reviewer #1 Response. Thank you for addressing this. -26 dB seems a bit high but I have not done any optimization of ADI with audiomoths.

New response: Thank you.

Lines 646-653. The statistical analysis section is one of the most confusing parts of the paper. I have no idea what kind of model is being developed. The description of the model's explanatory variables is not described as I would. Why time is divided by 24 and 365 is not justified. I would consult a statistician for the proper way to describe each of these models. I am also very suspicious that these data follow any distribution that is part of the assumption of the models (Poisson and count data). Some of the predictors of the model should also be landscape features such as distance to road, elevation, slope, aspect, percentage of natural vegetation within fixed radii of the sensors, etc. The authors should consult Sangermano, 2022, Landscape and Urban Planning). Most of the data that are needed are already posted on Google Earth Engine and these can be extracted at Global Scales. In all honestly, I would use a Random Forest model which avoids the trappings of these linear models.

Reply: We have now improved the descriptions of the methods. The statistical models that we apply are among the most commonly applied models in ecological statistics, i.e., generalized linear models. For the periodic functions of hour of the day and day of the year, the values 24 and 365 are the number of hours per day and number of days per year, respectively, and dividing by them is needed to obtain basis functions that are periodic over the time of the day or over the day of the year. For two different regression models, we used linear regression for continuous-valued data and Poisson regression for count data. While count data will often show overdispersion incompatible with the Poisson model, the counts here analysed concern species richness and show no such overdispersion. To measure how much variance in the data the model describes, we used simple R^2 values. Concerning the suggestion of using predictors such as distance to road, elevation, slope, aspect, or percentage of natural vegetation within fixed radii of the sensors, we fully agree that their effects would be interesting to study. However, our study design is not targeted at capturing their effects – as we have very many temporal replicates for relatively small number of study locations, rather than a very large number of study locations. The covariates that the Reviewer mentions are constant over time and thus fixed for each study location. Thus, including them in the model would quickly make the model overparameterized. To avoid this, we included only key site-specific variables directly related to the study questions.

Reviewer #1 Response. Thank you rewriting this section. It is now clear and succinct.

New response: Thank you.

Lines 663-669. The results of the analysis suggest to me that perhaps roads in both the urban spaces and nearby natural environments are being detected. Indeed, highway noise can be detected about 2km away from the road and so there are few places one can be in urban and peri-urban areas without hearing road

noise. Sensors should have been placed in the interior of protected areas and inside (an core area) the natural area if that is a comparative type of location desired by the authors.

Reply: The locations of recorders were set by local teams. The natural locations were chosen to be the most natural ones present within a restricted distance from the urban one – just like the urban sites were classified as urban by regional, not global or absolute standards. Importantly, these are the urban vs natural sites accessible to the local human population, and thus represent the soundscapes that people can realistically switch between.

Reviewer #1 Response. This might be added to the supplement as general rules for sensor deployment.

New response: This consideration has now been added to the Supplements.

Acknowledgements

863-897. This can be rewritten to be more succinct (reduce the length by about a 1/3rd).

Reply: The Acknowledgements section has been thoroughly revised.

Reviewer #1 Response. Well done.

New response: Thank you

Author contributions

I have co-authored several papers that have this many co-authors and I find the term “commented on the manuscript” not to be very useful nor adequately describe their contribution. An email that asks if they are OK with the current paper is not a contribution to the writing of the paper. Can the authors describe what is done to merit this level of contribution?

Reviewer #1 Response. Much better, thank you.

New response: Thank you.

Reply: The list of author contributions has been fully revised to lay plain the team-wide effort and the contributions made by the coauthors.

Figures

Figure 1

Lines 846-847. This is a mess. I’m confused by the three subfigures, especially B. The figure caption does not specify important parts of the diagram (why is the y-axis in B not labeled). Figure C is most informative as it tells me that the authors have not used acoustic indices properly. When any of these acoustic indices have highly skewed distributions, then the parameters that are used are not correct. This is a huge problem in this field where authors simply download the code and then use the default setting without thinking through what they are supposed to measure. Take ADI for example. ADI measures the amount of acoustic activity in each frequency band above a “noise floor”. If the noise floor that is selected is too low (or if the preamps are set high on the microphone), then you will get a saturated ADI values (close to 3). Researchers need to do a sensitivity analysis of the key parameters for each of the ADI noise floor parameters; since the original ADI was created using condenser microphones and Audiomoths use an entirely different technology, values closer to 0 dBFS are probably best. Researchers need to plot out ADI values across the entire range and select the one that has a broad distribution. I suspect that similar problems are arising with ACI, H, Speech and Vehicle. I am not sure what part of Figure A is meant to describe the two spectrograms. There is a lot listed that is not clear. For the table -- Natural and Urban columns -- I assume are means?

Reply: Figure 1 has now been split into two figures, Fig. 1 and Fig. 2, with the figure legends revised to clearly explain the content of each. Following the Reviewer's suggestion, Fig. 1 is now dedicated to define each metric used and what is expected to encapsulate. To illustrate the information conveyed by each, Fig. 2 has now been dedicated to showing differences in indices observed between two example spectrograms. (This was also the case in the previous illustration, but admittedly the reference to examples vs. global distributions got highly confusing, for which we apologise.) For clarity, the global distributions of values across indices has been removed.

Having said this, we dispute the Reviewer's suggestion that the previous illustration would have showed "that the authors have not used acoustic indices properly". In panel C (now omitted), the distribution of each for acoustic index was shown between minimum and maximum. For some acoustic events e.g. Speech, Vehicle, and Rain, these values are probabilities. As these events can be rare, probability values close to zero were dominating. Importantly, distribution plots in now-omitted panel C included all data. Since these data include recordings from day and night throughout a year, there are many recordings where no animal or other vocalizations are found. Regarding ADI, we recomputed all values with new parameters. In the first version of the manuscript, we used the default noise floor value. Yet, the example on the right hand side in omitted panel C showed an ADI value of 1.53, and a value close to 3. We have now recalculated ADI after adjusting the default noise floor value from the previous value of -50dB to a new value -26dB. Also for ACI, we have experimented with different window sizes, but found no major scope for optimising this parameter.

Reviewer #1 Response. Much better, thank you.

New response: Thank you.

Figure 2

These are very interesting plots. The ones on the left, however, are suspicious to me since the timing of dawn is a function of time of year and latitude. Second, the acoustic phenology curves to the right, also very interesting, lack proper context. For example, in areas of the world that experience four seasons, winter, spring, summer and fall; these are in the mid-latitudes. In the tropics, some areas have two seasons (wet and dry) and some have four (two wet and two dry). Areas at 30°N and S create, by Hadley cell circulation patterns, deserts which typically experience one very short rain event in the form of a monsoon. So, latitude is indeed important but the phenologies should conform to these seasonal patterns that are driven by global climate regimes. Also, mountains throw a wrench into these global patterns. In the end, an interesting set of plots, but they do not tell me what I would expect. The textbook by Lomolino (Introduction to Biogeography) describes these global, biosphere patterns well and this is a good source to understand global ecosystem distributions.

Reply: We fully agree that there are many climatic and other factors with a potential impact on acoustic phenology. What we perhaps failed to explain properly in the original submission is that quantifying the amount and nature of such variation is indeed one of the core aims of the manuscript. As we have now clarified, the comparison between the global model and the site-specific models is explicitly targeted at addressing this question. In the site-specific models, we quantify seasonal and diurnal variation separately for each site. Hence, the site-specific models fully account for the properties of the site (e.g. whether there are two or four seasons). In contrast, the global model is based on the simplified assumption that variation among the sites in their seasonal and diurnal variation only depends on latitude (as the global model includes the interaction between the periodic functions and latitude). Now, how much more of the data can be explained by the site-specific models than the global model then quantifies how important the site-specific factors are. The Reviewer felt that the left-hand panels are suspicious since the timing of dawn

is a function of time of year and latitude. As mentioned, the model fitted here allows timing to depend on latitude, but the results suggest that the effect is quite minor. Thus, we consider this as a result rather than a suspicious assumption. In more detail, we believe that the Reviewer is especially concerned about two points: The first one is that the timing of dawn and dusk varies with season and latitude. To address this, we have now reanalysed the data with respect to local sunlight conditions -- and found patterns consistent with the original ones. These analyses, the results and our conclusions have been described in full detail in the revised manuscript. Second, the Reviewer stresses the impact of factors OTHER than time or latitude. The impact of these factors is indeed addressed, as summarised by four additional predictors: the human footprint index, elevation, annual mean temperature, annual precipitation (and most completely by the site-specific models). The results are shown in Fig 2, with the main conclusion that latitude, time of day and day of year suffice to explain major variation in key metrics of biophony but not in anthropophony. These findings have been further highlighted in the revised version of the manuscript.

Reviewer #1 Response. Much better, thank you.

New response: Thank you.

Figure 3

Most useful figure in the paper. Label the x-axis in Figure 3B.

Reply: We are grateful for the Reviewer's appreciation of the figure and apologise for the missing axis label. This label has now been added.

Reviewer #1 Response. Much better, thank you.

New response: Thank you.

Figure 4

Very interesting plots. I wonder if the authors should consider doing a Jaccard Index of Dissimilarity to determine how similar species compositions between sites are. The urban+natural plots suggest that these are fairly different in species composition. For Figure 4A and B, is a linear model fit here or is that line just a reference point (at which case it should be removed or at least explained).

Reply: Thank you for suggestion, the Jaccard index would indeed describe some of the differences. However, we think that in the present form the plots (C and D) give the information in a more explicit way. Plots in C and D illustrate species accumulation curves, i.e. how the species richness (y axis) grows when increasing the number of detections (x axis). In panels A and B, we have taken the species richness corresponding to a fixed number of detections (1000) and plotted those values for natural vs urban sites (panel A) and for combinations of natural and urban sites (panel B). The result is that for samples of equivalent size, more species are detected in urban sites compared to the natural site, and that the set of species observed at the two sites in a pair are partly complementary. By comparison, a simple Jaccard index would be more sensitive to the number of false positives. Because species accumulation curves are based on data that include the information how frequently each species has been detected, they are less likely to be affected by single spurious detections.

Reviewer #1 Response. Much better, thank you.

New response: Thank you.

Figure S1

I would consider this as Supplement 1 and then the rest packaged as Supplement 2. The lack of any description of the dozens of (colorful) plots is frustrating.

Reply: We were sorry to hear that supplement 1 plots was uninformative. In response, and after conferring with the Editor, we have broken the figure into 55 parts and added a clarifying legend for each plot. The purpose is to show results for both site-specific models (where models were fitted using only local data) and the global model (where data from all sites were included, with latitude as an explanatory variable). The latitude is identified by the colour of each curve, with the same colour used in the map of Fig. S1.

Figure S2

I am seeing a lot of these Spearman rank correlations in papers and these authors fall into the same trap of how they are supposed to be used to interpret acoustic indices. First, if they do not correlate with one another, then the two acoustic indices can be considered to be different measures of the soundscape! Second, if they strongly correlate with one another, then you can drop those that are in future analyses that consider multivariate analysis. I'm confused by their general interpretation that a mixture of acoustic index values means that they are not informative. Third, are all of these correlations significant? Authors need to provide a level of significance and also adjust to the overall experimentwise error rate and use either the Bonferroni or Dunn-Sidak method to adjust the alpha significance level.

Reply: We agree that they all are different measures of the soundscape – and we are not claiming that they would measure the same thing. Quite the contrary: in both the original and the further-revised main text, we stress the complementary nature of these metrics. The purpose of showing the correlations as colours is to convey the idea that some measures are more similar than others. However, we are not trying to calculate any statistical significance for the difference. This is a deliberate choice, since given the (very) high number of datapoints that we have created, even tiny differences between the measures would lead to statistically significant results. Thus, what is more relevant than “how confidently we can say that there is any difference”, is “how big is the difference”, and this is what we try to illustrate. The way in which we have compared them statistically is identified in the legend to Figure 3 of the main text, i.e., we compare them on the basis of how predictable they are as a function of time of the day and day of the year (as measured by R^2). The corresponding section of the main text has now been completely rewritten.

Reviewer #1 Response. *I'm fine with this approach.*

New response: **Thank you.**

Figure S3

This is a good feature of the paper (to present validation of BirdNET). Earlier in the paper, the authors describe the two kinds of errors that should be assessed (False Negatives and False Positives). However, these plots only illustrate False Positives and True Positives. We are finding False Negatives to be a huge problem with BirdNET. There are many metrics that assess these model user and producer errors and these should be used to determine whether BirdNET is good. That said, the selection of 0.8 seems reasonable using this guide but it still does not provide a measure of False Negatives. It might be possible to extract that from their listener database. I suspect that False Negatives are higher in areas outside of North America and Europe.

Reply: Indeed, false negatives are higher in areas outside of North America and Europe. Nonetheless, to quantify false negatives across continents would require a tremendous amount of manual work (with global bird experts listening to recordings) -- for which reason we have been forced to abstain from it in the current manuscript. Nonetheless, we stress that we are currently investing heavily in the annotation of

improved sound libraries (see <https://bsg.laji.fi/> with an advertisement accessible under <https://helda.helsinki.fi/server/api/core/bitstreams/cecabf84-4f20-439d-ae6c-57b81658a44d/content>).

For some locations, it is still possible to evaluate the rate of false negatives offered by BirdNET. E.g. in Madagascar, the current BirdNET model includes only half of the bird species known to occur in the region. For the European sites, we have now used or manually curated subset of data (n=615 clips of 3 sec each) to calculate false negatives. The values are 60% with BirdNET confidence threshold 0.3 and 70% with BirdNET confidence threshold 0.8. These European values are comparable to the rates of false negatives gleanable from Fig 4 of Funosas et al. 2024 (Assessing the potential of BirdNET to infer European bird communities from large-scale ecoacoustic data. Ecological Indicators 164, 112146), and their implications are further discussed in the revised manuscript.

Reviewer #1 Response. See my comments on this as there may be a little more that can be done.

New response: We agree that more could be done, but not without substantial effort. As outlined in our response above, the added value achievable should be gauged against work already published by Funosa et al. 2024.

Table S4

This is a useful table and I suggest leaving this in and distribute as an Excel spreadsheet.

Reply: Thank you for the comment. We have now added it as a separate xlsx file.

Reviewer #1 Response. Looks great, thanks. See Comments document for other small suggested changes to the figures in the Supplement.

New response: We have not received any additional Comments document.

NEW NOTE: The comments below include no new responses from the Reviewer, and we thus take it that they have been settled.

Figure S5

This was unreadable for me and given that axes are not labeled, I can't comment on it.

Reply: We have cleaned up the figure and added explanation.

TextS5

Should this be TextS6?

I am not certain how these habitat/trait variables are used or are then associated with each species of bird (I assume that is what is being done here). As mentioned above, many landscape variables are simply missing that I believe are strong predictors of temporal patterns of the soundscape and wonder why they are not considered.

Reply: All parts of the Supplementary material have now been renumbered for clarity, and all references from the main text to the supplements have been checked. In terms of missing landscape variables, we stress that there is, per definition, an infinite number of potential descriptors to consider. Importantly, our intent is *not* to pinpoint the exact environmental drivers of local variation in soundscapes among this set, as that would call for way more sites than here available. Instead, we use a hypothesis-driven approach:

among a set of predefined metrics, we test for their added contribution beyond time of day and day of year. As key clarification, we point to our response to the Reviewer's comment on Figure 2 (above). Overall, we have now further clarified all parts of our analysis and inference, including our *a priori* choice of covariates considered.

Table S5

Should this be Table S7?

Experimentwise error rate should be adjusted in my opinion. Please consult a statistician.

Reply: Thank you for pointing this out. We have now corrected p-values due to multiple testing using Holm's method. In the revised table, we show both raw and adjusted values.

Table S6

Hmmm.....these are important results that are not mentioned (seems entirely new information) in the body of the main text. It is correct that BirdNET does not do well in tropical regions but this paragraph buried deep in a supplement raises a few red flags for me. Content like this should be placed in the body of the main text and placed in the context of limitations.

Reply: This section was originally included in the main text, but was relegated to the Supplements before our previous submission (as being a methodological consideration rather than a main result). Spurred by the Reviewer's concern, we have now reinserted it in the Discussion section of the main text.

Table S7

Useful table. The team name is not useful to readers. List co-author with their initials instead.

Reply: All relevant site information is now included as Sheet 3 in the supplementary xlsx file. The site names refer to the codes by which the data are organised and were retained as the links needed to connect site-level covariates to the raw data.

Figure S1 (Map)

Why not include this overlaid on a map of the continents? Use of ArcGIS would be helpful. No legend is provided. Sites names run over one another. Overall, a very unprofessional map.

Reply: The idea of this map was just to identify all recording sites by colour codes, to give full credit to the teams involved. We have now revised the figure for clarity and show the locations of the sites with continent borders on the map. At the same time, this map swerves as the legend for Supplementary Figures S2-S55.

Figure S1 (Multicolored Plots)

A set of $4 \times 27 = 108$ plots are not described at all and none have a y-axis labeled. There are no descriptions to these. Nor is there a legend unless the legend is the S1 Map color coded dots. Were these included since they are so colorful? Waste color toner and paper for me.

Reply: Here the idea is to show the model predictions for both site-specific models and the global model. For the rationale behind the models we refer the Reviewer to our response to the comment on Figure 2 (above) and to the revised Methods description. The colour is used for identifying the sites and their latitudes, using the same colours as introduced in the new map in Figure S1. After conferring with the Editor, we have broken the figure into 55 parts and added a clarifying legend for each plot. All axes now have labels.

Conclusions

The value of this paper is the data. It is highly unique and the authors are able to address a very unique set of questions. However, it is poorly written, analyses in several areas are flawed, the raison d'être is not well posed, and many of the figures are confusing. A solid paper will require some significant reanalysis and also robust, concise, precise and informative prose (a complete rewrite).

Reply: We are most grateful to the Reviewer for their helpful feedback, which has enabled us to reach a significant reanalysis and a complete rewrite of the paper.

Review by Bryan C. Pijanowski

REVIEWER #2

Remarks to the Author

The authors have made thorough revisions to the manuscript which have significantly improved the submission and does better justice to this excellent data set.

To my mind there remain a few small edits that could further improve clarity, veracity and reproducibility

New response: We are most grateful to the Reviewer for their kind comments provided along the way. We have implemented all edits suggested, with point-by-point responses below.

Main

Para 1

“The resulting distribution of sound amplitudes across frequencies determines how efficiently animals can communicate with each other”

>> I realise you have changed this from “affects” but that is more correct. Or “partially determines” if you prefer as there are other factors – for example landscape and vegetation structure as well as weather will also impact.

New response: We have now changed the text to "partially determines".

Para 3.

“the dusk chorus is dominated by insects”

>> This has been suggested by the other reviewer, but is not true in many biomes of temperate northern hemisphere. Please qualify.

New response: Since we back up this sentence with a reference to a paper by reviewer #1, we hope that it will provide sufficient context for the reader to critically validate our claim.

Results

P7. Para 1

“Overall, the urban soundscape is driven by the structure of land use, and across cities, the amount of vegetation – a metric of green infrastructure – correlates with the intensity of sound”

>> Does vegetation density correlate with biophony? Or overall soundscape intensity? Or is it inversely correlated with overall sound intensity (ie in built up areas no green = lots of traffic)? Please clarify.

New response: To clarify this issue, we have now added a qualifying statement indicating that vegetation absorbs sounds. In this way vegetation density inversely correlates with antropophonic sounds. For biophony, however, the situation is different, since vegetation may attract more animals.

P7 para 2.

>> Suggest using “quiet” rather than silence as you have throughout?

New response: The apparent inconsistency in terminology derives from the fact that here, Silence refers to a specific output class provided of YAMNet, not to the general phenomenon of

“quiet”. 'Silence' is one sound category of google's AudioSet. To clarify the issue, we have now capitalised “S” in “Silence” when referring to this specific variable.

Ibid Para 5.

"Across other indices, urban green spaces proved generally noisier (index SpecLow, Speech and Vehicle) than their more pristine counterparts"

>> Do you define “noisier” somewhere? Might “higher in anthropophony” be better?

New response: Thank you for pointing this out. We have now replaced the word "noisier" by "higher in antropophony" as suggested by the Reviewer.

Discussion

"well-predictable" is grammatically incorrect

>> suggest strongly or highly predictable?

New response: Thank you for this suggestion, we have changed "well-predictable" to "highly predictable".

Conclusions

"Given the potential for anthropophony to mask current animal communication and its strong contributions to global soundscapes, it should offer a strong selective force"

>> “It is likely to exert a strongly selective force”? use imperative here feels unusual and doesn’t carry what I understand to be your meaning

New response: We have now changed the text as suggested.

"Yet, poor predictability in space and time will compromise such selection, and given taxa seem constrained in adjusting their acoustic signals to overcome noise"

>> Missing clause, please complete the sentence

New response: Thank you, we have now corrected the sentence. The new sentence reads “Yet, poor predictability in space and time will compromise such selection, and some taxa seem unable to adjust their acoustic signals to overcome noise”.

METHODS

Spectral energy

Thanks for putting in the FFT params, but this is an unusual way of reporting

"Spectrogram was computed from a one-minute recording using a 1024-point FFT with Hann window at 10 ms intervals using the Python library librosa"

>> For a 1024 point FFT at 48000 kHz, each window is 1024/48000 ms = 21.333 long. 10 ms is not an exact multiple of this. For clarity and reproducibility, do you mean you used a 50% hop size (overlap) ie 512 points?

New response: For clarity, we have now specified the explanation of time lengths (window size and hop length) and now report them both in terms of time points and milliseconds.

Reviewer #2 (Remarks on code availability):

I have reviewed but not run the code.

It is clearly organised, well documented and looks to include everything needed to reproduce the results of the paper.

New response: Thank you.